# A New Non-Extensive Equation of State for the Fluid Phases of Argon, Including the Metastable States, from the Melting Line to 2300 K and 50 GPa

**Frédéric Aitken \*** , **André Denat and Ferdinand Volino**

University Grenoble Alpes, CNRS, Grenoble INP, G2Elab, F-38000 Grenoble, France
* Correspondence: frederic.aitken@g2elab.grenoble-inp.fr

**Abstract:** A new equation of state for argon was developed with the view of extending the range of validity of the equation of state previously proposed by Tegeler et al. and obtaining a better physical description of the experimental thermodynamic data for the whole fluid region (single-phase, metastable, and saturation states). As proposed by Tegeler et al., this equation is also based on a functional form of the residual part of the reduced Helmholtz free energy. However, in this work, the fundamental equation for Helmholtz free energy was derived from the measured quantities $C_V(\rho, T)$ and $P(\rho, T)$. The empirical description of the isochoric heat capacity $C_V(\rho, T)$ was based on an original empirical description explicitly containing the metastable states. The thermodynamic properties (internal energy, entropy, and free energy) were then obtained by combining the integration of $C_V(\rho, T)$. The arbitrary functions introduced by the integration process were deduced from a comparison between calculated and experimental pressure $P(\rho, T)$ data. The new formulation is valid for the whole fluid region from the melting line to 2300 K and for pressures up to 50 GPa. It also predicts the existence of a maximum of the isochoric heat capacity $C_V$ along isochors, as experimentally observed in several other fluids. For many applications, an approximate form of the equation of state for the liquid phase may be sufficient. A Tait–Tammann equation is therefore proposed between the triple-point temperature and 148 K.

**Keywords:** argon; data evaluation; equation of state; fundamental equation; property tables; thermal and caloric properties; vapor–liquid coexistence curve; spinodal; metastable state; Tait–Tammann equation of state

## Contents

**Abbreviations**

Symbol description

| | |
|---|---|
| $c$ | sound speed |
| $C_P$ | isobaric heat capacity |
| $C_V$ | isochoric heat capacity |
| $F$ | Helmholtz free energy |
| $G$ | Gibbs energy |
| $H$ | enthalpy |
| $i, j, k$ | serial numbers |
| $M$ | molar mass |
| $\mathscr{N}_a$ | Avogadro number |
| $P$ | pressure |
| $R_A$ | specific gas constant |
| $S$ | entropy |
| $T$ | thermodynamic temperature |
| $U$ | internal energy |
| $V$ | specific volume ($V = 1/\rho$) |
| $x$ | dimensionless parameter ($x = T_{\mathrm{div}}/T$) |
| $y$ | dimensionless parameter ($y = x^{-1} = T/T_{\mathrm{div}}$) |
| $Z$ | compressibility factor |
| Greek | |
| $\partial$ | partial differential |
| $\delta_T$ | isothermal throttling coefficient |
| $\rho$ | density |
| $\Gamma$ | incomplete gamma function |
| Superscripts | |
| o | ideal gas property |
| r, * | residual terms |
| ~ | dimensionless quantity |
| ^ | dimensionless quantity (for energy only) using $\frac{3}{2}R_A T_c$ as a reference |

Subscripts

| | |
|---|---|
| c | at the critical point |
| calc | calculated |
| exp | experimental |
| sat | denotes states at saturation |
| sp | denotes spinodal states |
| t | at the triple point |
| σl | saturated liquid state |
| σv | saturated vapor state |
| 0 | terms that do not contribute to $C_V$ |

**Physical Constants for Argon**

| | |
|---|---|
| $M$ | molar mass |
| $M = 39.948$ g mol$^{-1}$ | |
| $R$ | universal gas constant |
| $R = 8.31451$ J mol$^{-1}$ K$^{-1}$ | |
| $R_A$ | specific gas constant |
| $R_A = 0.2081333$ kJ kg$^{-1}$ K$^{-1}$ | |
| $T_c$ | critical temperature |
| $T_c = 150.687$ K | |
| $P_c$ | critical pressure |
| $P_c = 4.863$ MPa | |
| $\rho_c$ | critical density |
| $\rho_c = 0.535599$ g cm$^{-3}$ | |
| $T_t$ | triple-point temperature |
| $T_t = 83.8058$ K | |
| $P_t$ | triple-point pressure |
| $P_t = 68.891$ kPa | |
| $\rho_{t,Gaz}$ | triple-point gas density |
| $\rho_{t,Gaz} = 0.0040546$ g cm$^{-3}$ | |
| $\rho_{t,Liq}$ | triple-point liquid density |
| $\rho_{t,Liq} = 1.41680$ g cm$^{-3}$ | |
| $\rho_{t,Sol}$ | triple-point solid density |
| $\rho_{t,Sol} = 1.6239$ g cm$^{-3}$ | |

## 1. Introduction

Argon is a noble gas, and on earth, its isotopic composition is 99.6% $^{40}$Ar, 0.34% $^{36}$Ar, and 0.06% $^{38}$Ar. Argon is very stable and chemically inert under most conditions. Due to those properties and its low cost, argon is largely used in scientific and industrial applications. For instance, in high-temperature industrial processes, an argon atmosphere can prevent material burning, material oxidation, material defects during the growth of crystals, etc. Due to its molecular simplicity (monoatomic and quasi-spherical geometry), argon is also considered a reference fluid with well-known properties, i.e., its triple-point temperature (83.8058 K) is a defining fixed point in the International Temperature Scale of 1990 [1]. Its simple fluid characteristics allow, for example, to understand the fundamental mechanisms of interaction between ions and neutral species and thus gain a deeper insight into ion transport regimes (e.g., [2]). The widespread use of argon requires accurate knowledge of its thermodynamic properties in the largest possible temperature and pressure ranges, i.e., covering both stable and metastable states. Numerous empirical equations of state can be found in the literature, but most of them cover only small parts of the fluid region. For example, Shamsundar et al. [3] have shown that the development of cubic-like equations of state provides very accurate thermodynamic properties of liquids on the coexistence curve and in the metastable (superheated) state. However, this approach has flaws on the vapor side. A very detailed overview of argon's experimental thermodynamics and the most important equations of state published prior to 1999 can be found in [4], so we will not delve into it here. In [4], Tegeler et al. also describe a new equation of state for argon that covers a very wide range of the fluid phase and will serve as the reference equation of state for this study.

The development of the equation of state generally starts by an empirical description of Helmholtz free energy $F$ (i.e., an arbitrary set of mathematical functions is a priori chosen) with two independent variables: density $\rho$ and temperature $T$. All thermodynamic properties of a pure substance can then be obtained by combining derivatives of $F(\rho, T)$. The dimensionless Helmholtz free energy $\tilde{a} = F/(R_A T)$ is commonly split into a part $\tilde{a}^o(\rho, T)$, which represents the properties of the ideal gas at given temperature and density, and a residual part $\tilde{a}^r(\rho, T)$, which takes into account the dense fluid behavior. While statistical thermodynamics can predict the behavior of fluids in the ideal gas state with high accuracy, no physically founded equation is known that accurately describes the actual thermodynamic behavior in the whole fluid region. Thus, an equation for the residual fluid behavior, in this case for the residual part of Helmholtz free energy $\tilde{a}^r$, has to be determined in an empirical way. However, as Helmholtz free energy is not accessible to direct measurements, a suitable mathematical structure and some fitted coefficients have to be determined from properties for which experimental data are available. Hence, all the physical properties are contained in the mathematical form given to Helmholtz free energy.

In the wide-range equation of state for argon developed by Tegeler et al. [4], the residual part of Helmholtz free energy $\tilde{a}^r(\rho, T)$ contains polynomial terms, Gaussian terms, and exponential terms, resulting in a total of 41 coefficients (named $n_i$ in [4]), which represent the number of mathematically distinct entities (with each mathematical entity containing several adjustable parameters). This equation of state is valid for the fluid region delimited by

$$83.8058 \text{ K} < T < 700 \text{ K},$$

and

$$0 \text{ MPa} < P < 1000 \text{ MPa}.$$

The large number ($\sim$120) of adjustable parameters of the equation of state of Tegeler et al. (see Table 30 in [4]) are determined by a sophisticated fitting technique that is a powerful mathematical tool and a practical way for representing data sets (by assigning weights to each of them subjectively). This technique provides an easily practical overall numerical representation of the data, but it also allows for the completion of the representation of measurable quantities in areas where no measurements have been made. However, passing in a set of data points does not mean that the obtained variations have a physical meaning or that the physical ideas underlying mathematical representation are unique. For example, the following drawbacks of the equation of state of Tegeler et al. [4] can be noticed:

1. Extrapolation of the equation for the isochoric heat capacity in regions of high or low density and high temperature is non-physical.
2. The extrapolation of polynomial developments does not generally give valid results; indeed, polynomial development is very sensitive (i.e., instable) with respect to the values of its coefficients, and these coefficients cannot be truncated, even slightly. Therefore, all the coefficients $n_i$ of Tegeler et al. [4]'s model have 14 digits, and the coefficients thus have no physical sense.
3. The model applies to the pure fluid phases and cannot, in its actual form, take into account particular properties inside the liquid–vapor coexistence region. Moreover, the model gives negative values of $C_V$ on some isotherms inside the liquid–vapor coexistence region ($C_V < 0$ is never observed for classical thermodynamic systems). This implies, for example, some non-physical variations in the liquid spinodal curve.

The aim of this paper is not to increase the precision of the equation of state of Tegeler et al. [4] in its own domain of validity but instead to develop a new equation of state based on different physical ideas that can fill the drawbacks previously expressed in order to obtain a more physical description of the experimental thermodynamic data of argon in a broader temperature and pressure range. In the classical approach, the ideal part of the free energy is generally determined from the well-known properties of the ideal gas heat capacities. We propose to also extend the classical approach to the residual part; therefore, the proposed new equation of state is based on an original empirical description of the

isochoric heat capacity $C_V(\rho, T)$ containing the metastable states explicitly. Then, the thermodynamic properties (internal energy, entropy, and free energy) are obtained by combining the integration of functions involving $C_V(\rho, T)$. For instance, internal energy $U$ can be deduced from $U(\rho, T) = \int C_V(\rho, T)dT + U_0(\rho) + \text{constant}$, where $U_0(\rho)$ is an arbitrary function of density. In this way, possible data noise is smoothed. However, an integration process introduces arbitrary functions (e.g., $U_0(\rho)$). These functions can be deduced from a comparison between calculated and experimental data. The pressure equation of state $P(\rho, T)$ was chosen because it is the largest available data set. The set of experimental data taken into account by the model of Tegeler et al. [4] will be further extended with the inclusion of the L'Air Liquide database [5], thus extending the temperature validity range of this new modeling compared to that of Tegeler et al. The interest of this new approach is that it can be easily extended to all other fluids that exhibit a first-order transition with metastable states.

*Hereafter, the model of Tegeler et al. [4] will be simply named the* **TSW model**.

## 2. A New Equation of State for Isochoric Heat Capacity

As stated previously, the present approach starts with the empirical description of a chosen thermodynamic quantity. An experimentally measured quantity is chosen for description, which is not the case for Helmholtz free energy. The quantity that has the simplest mathematical and physical comprehensive variation is the isochoric heat capacity $C_V$ as a function of density $\rho$ and temperature $T$. Starting with this quantity, we therefore lose the advantage of the description provided by Helmholtz free energy, from which all other thermodynamic quantities can be obtained by derivation. However, it allows us to more easily introduce new physical bases, in particular non-extensivity, and simplicity is enhanced. Indeed, the number of coefficients $\alpha_i$ (without $\alpha_{\text{crit,b}}$) for the description of $C_V$ is 11 (see Table 1), and it will be shown in the next section that the number of coefficients $\alpha_i$ for the description of Helmholtz free energy is only **26** compared to **41** for the TSW model.

**Table 1.** Coefficients and exponents of Equations (6)–(12).

| $i$ | $\varepsilon_i$ | $\alpha_i$ |
|---|---|---|
| reg,1 | | 11.23233957 |
| reg,1a | 1.1178177 | |
| reg,1b | 0.23513928 | |
| reg,2 | | 0.53278931 |
| reg,2a | 2.9322362 | |
| reg,2b | 15.5957 | |
| m,1 | | 0.07079238 |
| m,2 | | 0.33623345 |
| m,3 | | 1.3019754 |
| m,4 | | $-0.24008716$ |
| m,5a | 14.4899 | |
| m,5b | 7.20862 | |
| nonreg,1 | | 0.089409 |
| nonreg,1a | 0.71915 | |
| nonreg,1b | 0.22569 | |
| nonreg,2 | | 0.015481 |
| nonreg,2a | 1.3401 | |
| nonreg,2b | 0.29485 | |
| div,1 | | 102.06515 |
| div,1a | 0.9218165 | |
| div,1b | 1.1328347 | |
| div,2 | | 120.40518 |
| div,2a | 0.12035802 | |
| div,2b | 4.424004 | |

**Table 1.** *Cont.*

| $i$ | $\varepsilon_i$ | $\alpha_i$ |
|---|---|---|
| crit,a | 0.80803 | 701.52 |
| crit,b | 1.134 | 4.27385 |
| crit,c | 1.436786 | |
| crit,d | 123.1335 | |
| crit,e | 2.205614 | |
| crit,f | 26.32662 | |
| crit,g | 4.437711 | |

After choosing the thermodynamic quantity to be described, one must find a mathematical structure for its representation. A virial-like development is an easy and widespread approximation. The problem with polynomial terms is that they introduce very small oscillations that are not physical. To avoid such effects, it is assumed that the description must not contain any form of polynomial expression. Thus, the description is formulated in terms of power laws and exponentials with density-dependent exponents. We shall see that with such a description, we obtain a dozen different characteristic densities among the parameters of the model instead of only $\rho_c$ (which is consistent with the fact that argon does not follow the law of corresponding states). Therefore, all the equations of states are expressed in a dimensionless form according to the variables $\rho$ and $T$, which lead to simpler expressions than if one had considered the dimensionless variables $\rho/\rho_c$ and $T/T_c$. In addition, the most suitable units for density $\rho$ and temperature $T$ have been chosen as g/cm$^3$ and Kelvin, respectively.

As for Helmholtz free energy, the isochoric heat capacity is split into a part $C_V^{\mathrm{o}}$, which represents the properties of the ideal gas and a part $C_V^{\mathrm{r}}$, which takes into account the residual fluid behavior at given $T$ and $\rho$. Note here that the ideal part of free energy is, in fact, determined by the known properties of $C_V^{\mathrm{o}}$ in the classical approach. Because argon is monoatomic, only the translational contribution to the ideal gas heat capacity $C_{P,tr}^{\mathrm{o}} = \frac{5}{2}R_{\mathrm{A}}$ has to be taken into account, so it is deduced from Mayer's law that $C_V^{\mathrm{o}} = \frac{3}{2}R_{\mathrm{A}}$. In dimensionless form, the isochoric heat capacity is written as follows:

$$\tilde{c}_V(\rho, T) = \frac{C_V(\rho, T)}{R_{\mathrm{A}}} = \tilde{c}_V^{\mathrm{o}} + \tilde{c}_V^{\mathrm{r}}(\rho, T) = \frac{3}{2}\left(1 + \frac{2}{3}\,\tilde{c}_V^{\mathrm{r}}(\rho, T)\right) \tag{1}$$

To take into account all fluid domains, including the liquid–vapor coexistence region and the region around the critical point, $\tilde{c}_V^{\mathrm{r}}(\rho, T)$ must be split up into three terms—regular, non-regular, and critical—such that

$$\tilde{c}_V^{\mathrm{r}}(\rho, T) = \tilde{c}_{V,\mathrm{reg}}^{r} + \tilde{c}_{V,\mathrm{nonreg}}^{r} + \tilde{c}_{V,\mathrm{crit}}^{r} \tag{2a}$$

with

$$\left\{\begin{array}{l} \tilde{c}_{V,\mathrm{reg}}^{\mathrm{r}} = n_{\mathrm{reg}}(\rho)\left\{1 - \exp\left(-\left(\lambda\frac{T}{T_c}\right)^{1-(m(\rho)-1)}\right)\right\}\left(\frac{T}{T_c}\right)^{m(\rho)-1} \\[2mm] \tilde{c}_{V,\mathrm{nonreg}}^{r} = n_{\mathrm{nonreg}}(\rho)\,\exp\left(-\left(\frac{T_{\mathrm{div}}(\rho)}{T}\right)^{-3/2}\right)\frac{1}{1-\frac{T_{\mathrm{div}}(\rho)}{T}} \\[2mm] \tilde{c}_{V,\mathrm{crit}}^{r} = n_{\mathrm{crit}}(\rho)\left(\frac{T_{\mathrm{div}}(\rho)}{T}\right)^{\varepsilon_{\mathrm{crit}}(\rho)} \end{array}\right\} \text{ for } T \geq T_{\mathrm{div}} \tag{2b}$$

where $\lambda = 6.8494$ and $n_{\mathrm{reg}}(\rho)$, $m(\rho)$, $n_{\mathrm{nonreg}}(\rho)$, $T_{\mathrm{div}}(\rho)$, $n_{\mathrm{crit}}(\rho)$, and $\varepsilon_{\mathrm{crit}}(\rho)$ are empirical functions determined from the best fit of NIST [6] and Ronchi [7] data, whose expressions are given further on. In other words, it is assumed here that these two data sets are a priori consistent with each other.

An important feature of the Ronchi model is to predict the appearance of a maximum on the isochoric heat capacity $C_V$ along isochors. A maximum of $C_V$ along isochors has been experimentally observed in several fluids, for example, water. Consequently, the ex-

trapolation of $C_V$ along isochors from a given model must show a maximum, as predicted by the Ronchi model. This predicted behavior of $C_V$ along isochors constitutes the main interest of the Ronchi model.

The Ronchi calculated data cover the largest available temperature range of 300 to 2300 K and the largest available pressure range of 9.9 to 47,058.9 MPa (420 data points). It is important to notice that they are consistent with many experimental data that were used by Tegeler et al. [4] but which they assigned to groups 2–3. The data of Ronchi were assigned to Group 2 by Tegeler et al. From the data of Ronchi, the highest available density is called $\rho_{\mathrm{max,Ronc}}$, and its value is given in Table 2.

**Table 2.** Characteristic values of densities of argon and the corresponding molar volumes used in Equations (6)–(12).

| $i$ | $\rho_i$ (g/cm$^3$) | $V_i$ (cm$^3$/mole) |
|---|---|---|
| t,Gas | 0.0040546 | 9852.51318 |
| t,Liq | 1.41680 | 28.1959 |
| c | 0.53559 | 74.5857 |
| crit,a | 0.51182 | 78.0502 |
| crit,b | 0.73085 | 54.6589 |
| max,Ronc | 3.35697 | 11.9 |
| reg,Ronc | 3.53159 | 11.3116 |
| m,Ronc | 3.67875 | 10.8591 |
| u,1 | 6.61153 | 6.04217 |
| u,2 | 3.99925 | 9.98884 |
| u,3 | 3.90870 | 10.22026 |
| s,1 | 1.50915 | 26.47047 |
| s,4 | 1.18697 | 33.65528 |
| sRonc,1 | 3.28898 | 12.146 |
| sRonc,2 | 4.31602 | 9.25574 |

By construction, a part of the Ronchi [7] and NIST [6] data overlaps. For their common range of density values, it is observed in Figure 1 that the deviation is always less than 2.5%, so the data of Ronchi can be considered to be consistent with the data from NIST. *Hereafter, the term "NIST" will simply be used to refer to the data in [6] and the term "Ronchi" to refer to the data in [7].*

The relations constituting Equation (2) are therefore consistent with two sets of coherent data, but at this stage of the theoretical development, it must be strongly emphasized that these relations are only valid for temperatures $T \geq T_{\mathrm{div}}(\rho)$, i.e., in particular for all states in the single-phase region. In this way, $T_{\mathrm{div}}(\rho)$ defines a divergence curve (i.e., it defines an asymptotic curve), and we will see later that it is related to the spinodal curve. We shall see in Section 3.4 how this relation is transformed for $T < T_{\mathrm{div}}(\rho)$ (i.e., for states inside the coexistence region).

It is very important to notice that with the present model, once the mathematical form of the regular term is chosen, it is not possible to envisage any mathematical form for the two other residual terms. Indeed, the two remaining terms must have a consistent mathematical form with that of the first one; otherwise, the amplitude terms $n_i$ become erratic functions of density and are no longer smooth functions. The mathematical forms are certainly not unique, but there are strong constraints on them. This is a fundamental difference from the classical fitting approach to the free energy function, where there is no mathematical constraint between the different terms that are summed.

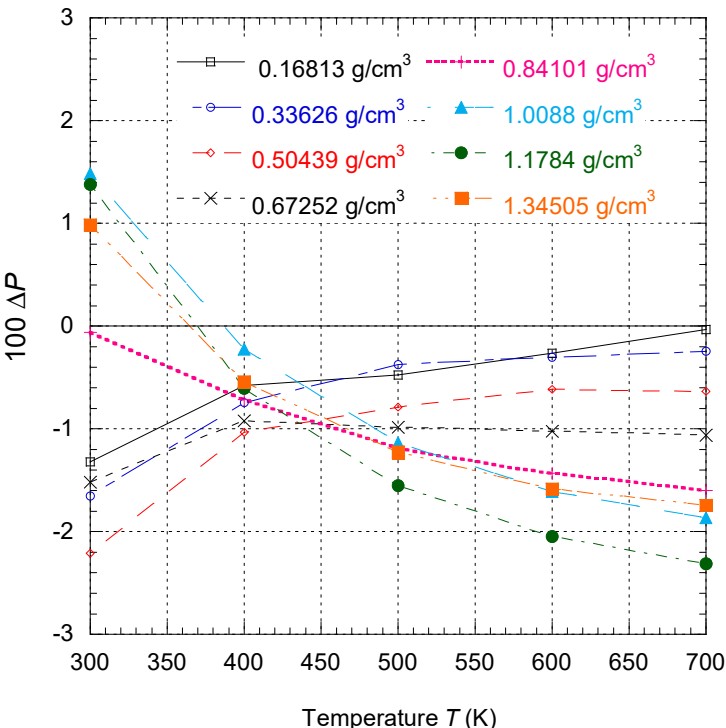

**Figure 1.** Percentage deviations of pressure $\Delta P = (P_{\text{Ronchi}} - P_{\text{NIST}})/P_{\text{Ronchi}}$ on different isochors in the common region of data from NIST [6] and Ronchi [7]. The lines are eye guides.

The three terms of the residual part of $C_V$ are now clarified.

- The first term $\widetilde{c}_{V,\text{reg}}^{\mathfrak{r}}$, which is a simple power law, is called "**regular**". It shows no singularity and can be calculated for temperatures from $T_t$ (triple-point temperature) up to infinity. When $T \to 0$ K following isochors, this term can be approximated by

$$\widetilde{c}_{V,\text{reg}}^{\mathfrak{r}}(\rho, T \to 0 \text{ K}) \cong \frac{3}{2} n_{\text{reg}}(\rho)\, \lambda \frac{T}{T_c} \tag{3}$$

which tends towards zero as a linear law.

For $T \gg T_c/\lambda$, $\widetilde{c}_{V,\text{reg}}^{\mathfrak{r}}$ reduces to

$$\widetilde{c}_{V,\text{reg}}^{\mathfrak{r}}(\rho, T \gg T_c/\lambda) \cong \frac{3}{2} n_{\text{reg}}(\rho)\left(\frac{T}{T_c}\right)^{m(\rho)-1} \tag{4}$$

The characteristic temperature $T_c/\lambda$ = 22.0 K is not the Debye temperature of argon, which is equal to 85 K. This new characteristic temperature was chosen to minimize the relative error of $C_V$ on the saturated vapor pressure curve so that *the term containing $\lambda T/T_c$ becomes important only for temperatures smaller than the triple-point temperature* (i.e., for $T \ll T_t$ with $\lambda$ = 6.8494).

- The second term $\widetilde{c}_{V,\text{nonreg}}^{\mathfrak{r}}$, called "**non regular**", presents an asymptote for $T = T_{\text{div}}(\rho)$ (i.e., $C_V$ is infinite for this value of temperature). This term is only significant near the liquid–vapor coexistence region. We can also note that the divergence is weak.
- The third term $\widetilde{c}_{V,\text{crit}}^{\mathfrak{r}}$ is important only in a small region around the critical point. This term allows us to reproduce the very sharp evolution of $C_V$ very close to the critical point. It can be understood as the macroscopic contribution of the critical fluctuations. This term plays the same mathematical role as the contribution of the four last terms in the second derivative with temperature of the residual free energy in the TSW model.

We have pointed out that the regular term $\widetilde{c}_{V,\text{reg}}^{\mathfrak{r}}$ tends to zero when $T$ tends to zero. It will be shown in Section 3.4 that for $T < T_{\text{div}}$, the non-regular and critical terms also

have a limit equal to zero when $T \to 0$ K. Hence, $\widetilde{c}_V^{\tau}(\rho, T) \to 0$ if $T \to 0$; this result is in agreement with the third law of thermodynamics (Nernst–Planck assumption). Because $\widetilde{c}_V = \widetilde{c}_V^0 + \widetilde{c}_V^{\tau}$, this law imposes $\widetilde{c}_V \to 0$ if $T \to 0$ and then $\widetilde{c}_V^{o} \to 0$ if $T \to 0$. To reach this result, $\widetilde{c}_V^{o}$ is rewritten in the following form:

$$\widetilde{c}_V^{o}(T) = \frac{3}{2}\left(1 - \exp\left(-\lambda_0 \frac{T}{T_c}\right)\right) \tag{5}$$

where $\lambda_0 = 18.2121$.

In the expression of $\widetilde{c}_V^{\tau}(\rho, T)$, all coefficients depend on density $\rho$ in the following way:

$$n_{\text{reg}}(\rho) = \alpha_{\text{reg},1}\left(\frac{\rho}{\rho + \rho_{\text{t,Liq}}}\right)^{\varepsilon_{\text{reg},1a}} \exp\left(-\left(\frac{\rho}{\rho_{\text{t,Gas}}}\right)^{\varepsilon_{\text{reg},1b}}\right) \\ + \alpha_{\text{reg},2}\left(\frac{\rho}{\rho_{\text{t,Liq}}}\right)^{\varepsilon_{\text{reg},2a}}\left\{1 - \exp\left(-\left(\frac{\rho}{\rho_{\text{reg,Ronc}}}\right)^{-\varepsilon_{\text{reg},2b}}\right)\right\} \tag{6}$$

$$m(\rho) = -\alpha_{m,1} + \alpha_{m,2}\exp\left(-\left(\frac{\rho}{\rho_{\text{t,Liq}}}\right)^{3/2}\right) + \alpha_{m,3}\left(\frac{\rho}{\rho_c}\right)^{3/2}\exp\left(-\frac{\rho}{\rho_c}\right) \\ -\alpha_{m,4}\ln\left(\frac{\rho}{\rho_c}\right) + m_{\text{Ronc}}(\rho) \tag{7}$$

$$m_{\text{Ronc}}(\rho) = \begin{cases} \left[\alpha_{m,1} + \alpha_{m,4}\ln\left(\frac{\rho}{\rho_c}\right)\right]\left(1 + \frac{\rho}{\rho_{\text{m,Ronc}}}\right)^{\varepsilon_{m,5a}}\exp\left(-\exp\left(\left(\frac{\rho_{\text{m,Ronc}}}{\rho}\right)^{\varepsilon_{m,5b}}\right)\right) & \text{for } \rho \geq \frac{M}{12.9} \text{ g/cm}^3 \\ 0 \text{ otherwise} \end{cases} \tag{8}$$

$$n_{\text{nonreg}}(\rho) = \begin{cases} \alpha_{\text{nonreg},1}\left(\frac{\rho}{\rho_{\text{t,Gas}}}\right)^{\varepsilon_{\text{nonreg},1a}}\exp\left(-\left(\frac{\rho_{\text{t,Liq}}}{\rho_{\text{t,Gas}}}\right)^{\varepsilon_{\text{nonreg},1b}}\left(\frac{\rho}{\rho_{\text{t,Liq}}-\rho}\right)^{\varepsilon_{\text{nonreg},1b}}\right) \\ +\alpha_{\text{nonreg},2}\left(\frac{\rho}{\rho_{\text{t,Gas}}}\right)^{\varepsilon_{\text{nonreg},2a}}\exp\left(-\left(\frac{\rho_{\text{t,Liq}}}{\rho_{\text{t,Gas}}}\right)^{\varepsilon_{\text{nonreg},2b}}\left(\frac{\rho}{\rho_{\text{t,Liq}}-\rho}\right)^{\varepsilon_{\text{nonreg},2b}}\right) & \text{for } \rho \leq \rho_{\text{t,Liq}} \\ 0 \text{ otherwise} \end{cases} \tag{9}$$

$$T_{\text{div}}(\rho) = \alpha_{\text{div},1}\left(\frac{\rho}{\rho_c}\right)^{\varepsilon_{\text{div},1a}}\exp\left(-\left(\frac{\rho}{\rho_c}\right)^{\varepsilon_{\text{div},1b}}\right) + \alpha_{\text{div},2}\left(\frac{\rho}{\rho_{\text{t,Liq}}}\right)^{\varepsilon_{\text{div},2a}}\exp\left(-\left(\frac{\rho}{\rho_{\text{t,Liq}}}\right)^{\varepsilon_{\text{div},2b}}\right) \tag{10}$$

$$n_{\text{crit}}(\rho) = \alpha_{\text{crit},a}\left(\frac{\rho}{\rho_c}\right)^{\varepsilon_{\text{crit},a}}\exp\left(-\left(\left(\alpha_{\text{crit},b}\frac{\rho - \rho_c}{\rho_c}\right)^2\right)^{\varepsilon_{\text{crit},b}}\right) \tag{11}$$

$$\varepsilon_{\text{crit}}(\rho) = \varepsilon_{\text{crit},c} + \varepsilon_{\text{crit},d}\exp\left(-\left(\varepsilon_{\text{crit},e}\frac{\rho - \rho_{\text{crit},a}}{\rho_{\text{crit},a}}\right)^2\right) + \varepsilon_{\text{crit},f}\exp\left(-\left(\varepsilon_{\text{crit},g}\frac{\rho - \rho_{\text{crit},b}}{\rho_{\text{crit},b}}\right)^2\right) \tag{12}$$

where $\varepsilon_i$ are exponents, and $\alpha_i$ are characteristic coefficients. Table 1 lists the values of these parameters.

Note that the function $m(\rho)$ is decomposed into two parts so that Equation (8) can represent Ronchi data at very high density (i.e., for $\rho \geq \frac{M}{12.9}$ g/cm$^3$). Indeed, the variations imposed by Ronchi data are too complex to be taken into account by a single function.

Now, some explanations will be given for the properties of these coefficients. Most of them involve the three characteristic densities of argon:

- the density $\rho_{\text{t,Liq}}$ of liquid at the triple point;
- the density $\rho_{\text{t,Gas}}$ of gas at the triple point;
- the critical density $\rho_c$.

Moreover, two other characteristic densities, $\rho_{\text{reg,Ronc}}$ and $\rho_{\text{m,Ronc}}$, have to be added in view to correctly fit the data of Ronchi at very high densities. All values of these characteristic densities are given in Table 2.

Obviously, other mathematical forms for Equations (6)–(12) could be used, but the proposed equations are the simplest ones that have been found and that lead to an accurate fitting of the whole data set. The consequences of this representation will be seen in Section 3.1.

The density dependence of the $n_i$ coefficients is shown in Figure 2. Each coefficient is equal to zero when $\rho \to 0$ and $\rho \to \infty$ and gets through a maximum in between (the maximum of $n_{\mathrm{reg}}$ really occurs but is outside the range of density shown in Figure 2).

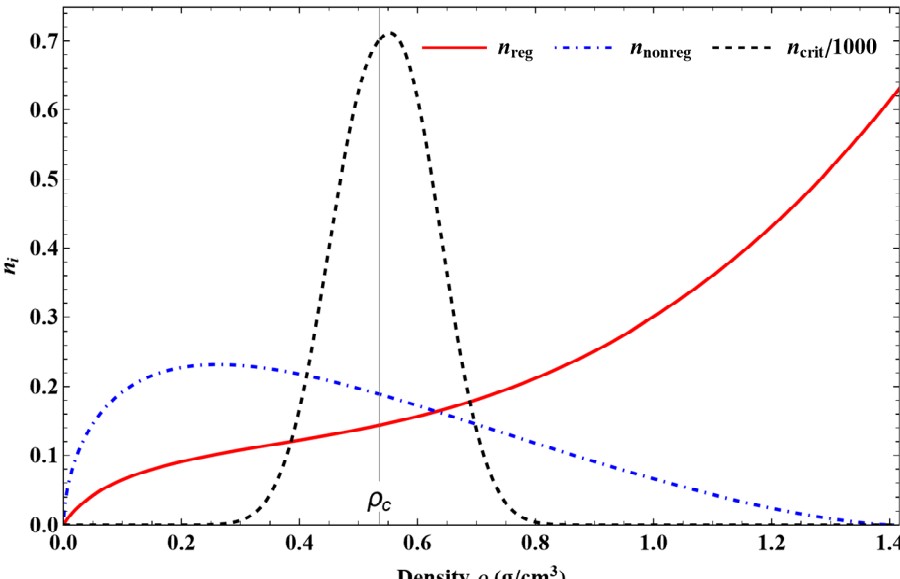

**Figure 2.** Variations with density of the functions $n_{\mathrm{reg}}(\rho)$ (red curve), $n_{\mathrm{nonreg}}(\rho)$ (blue dot-dashed curve), and $n_{\mathrm{crit}}(\rho)$ (black dashed curve) between $\rho = 0$ and $\rho = \rho_{\mathrm{t,Liq}}$.

The density dependence of exponent $m$ is shown in Figure 3. This coefficient is always strictly smaller than one, and it tends to $-\infty$ when $\rho \to 0$ and $\rho \to \infty$. Then, there are two density values for which $m = 0$. This means that, in the region where $T \gg T_c/\lambda$, $\widetilde{c}_{V,\mathrm{reg}}^{\tau}$ is always decreasing along isochors when the temperature is increasing.

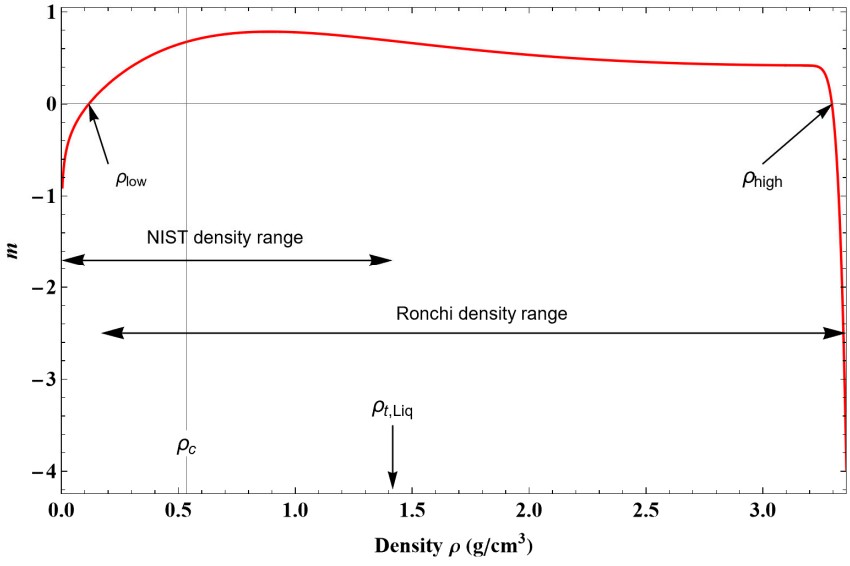

**Figure 3.** Variation with density of the function $m(\rho)$ from $\rho_{\mathrm{t,Gas}}$ to $\rho_{\mathrm{max,Ronc}}$.

The characteristic temperature $T_{\mathrm{div}}$ as a function of density defines a curve $T_{\mathrm{div}}(\rho)$ that lies entirely inside the vapor–liquid coexistence region defined by $T_{\mathrm{sat}}(\rho)$ (see Figure 4).

For a first-order phase transition, the divergence of $C_V$ must occur on the spinodal curve (i.e., loci of thermodynamic mechanical instability), corresponding to

$$\left(\frac{\partial P}{\partial V}\right)_T = 0 \tag{13}$$

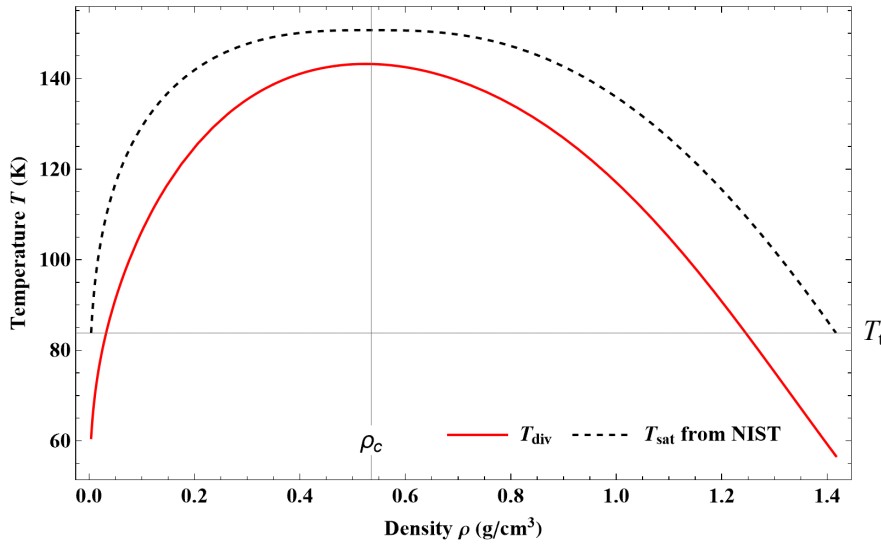

**Figure 4.** Variations with density of the functions $T_{\mathrm{div}}(\rho)$ (red curve) and $T_{\mathrm{sat}}(\rho)$ (black dashed curve) from $\rho_{\mathrm{t,Gas}}$ to $\rho_{\mathrm{t,Liq}}$. The curve $T_{\mathrm{sat}}(\rho)$ is deduced from NIST data [6].

Because no experimental data of the spinodal curve can be found in all the density ranges from $\rho_{\mathrm{t,Gas}}$ to $\rho_{\mathrm{t,Liq}}$, $T_{\mathrm{div}}$ was only determined by fitting the data of $C_V$ from NIST. If this set of data is accurate and consistent enough with the $P\rho T$ data set, one should be able to identify $T_{\mathrm{div}}(\rho)$ as the spinodal temperature curve. We will discuss the results obtained for the spinodal states in more detail in Section 4.3.3.

From Equation (2), it is also easy to see that the second thermodynamic instability (i.e., the thermal instability), defined by

$$C_V < 0 \tag{14}$$

will never occur in the present approach, contrary to the TSW model.

Consequently, with Equation (2) being valid for $T > T_{\mathrm{div}}$, this relation can be used in the vapor–liquid coexistence region by crossing $T_{\mathrm{sat}}(\rho)$ till approximately the spinodal curve. No trouble occurs as long as $T > T_{\mathrm{div}}$, though the model is based on a pure fluid description. The fact that there is no discontinuity of $C_V$ when crossing the coexistence curve (except at the critical point) is a characteristic of a first-order transition. We shall see in Section 3.4 how to treat the crossing of the divergence curve defined by $T_{\mathrm{div}}(\rho)$. Finally, it can be noticed that $T_{\mathrm{div}} = 0$ for $\rho = 0$ and $T_{\mathrm{div}} \to 0$ when $\rho \to \infty$; hence, $T_{\mathrm{div}}(\rho)$ shows the right density dependence, which allows us to investigate the fluid properties from the gas phase up to the sublimation curve.

The flexibility of the present method is now illustrated from the equation of state for the isochoric heat capacity. If one wants to represent the data from NIST instead of the data of Ronchi for densities higher than $\rho_{\mathrm{t,Liq}}$, it is only necessary to change the values of the couple $(\rho_{\mathrm{reg,Ronc}}, \varepsilon_{\mathrm{reg,2b}})$ and the mathematical form of the exponent function $m(\rho)$. In this case, Equation (7) must be replaced by the following function:

$$
m_{\mathrm{NIST}}(\rho) = \alpha_{\mathrm{m},1} - \alpha_{\mathrm{m},2} \ln\left(\frac{\rho}{\rho_{\mathrm{m},2}} + \frac{\rho_{\mathrm{m},2}}{\rho}\right) + \alpha_{\mathrm{m},3}\left(\frac{\rho}{\rho_{\mathrm{m},3}}\right)^{\varepsilon_{\mathrm{m},3a}} \exp\left(-\left(\frac{\rho}{\rho_{\mathrm{m},3}}\right)^{\varepsilon_{\mathrm{m},3b}}\right)
$$
$$
+ \alpha_{m,4} \exp\left(\left(\frac{\rho_{\mathrm{m},4b}}{\rho_{\mathrm{m},4a}}\right)^{2\varepsilon_{m,4}} \left(\frac{\rho_{\mathrm{m},4a}}{\rho-1}\right)^{2\varepsilon_{m,4}}\right) + m_{\mathrm{Extrapol}}(\rho) \tag{7bis}
$$

with

$$m_{\text{Extrapol}}(\rho) = \frac{\rho_{\text{t,Liq}}}{\rho}\left[-\frac{\alpha_{\text{m},5}}{\varepsilon_{\text{m},5}-1}\left(\frac{\rho}{\rho_{\text{m},4a}}\right)^{\varepsilon_{m,5}} + \alpha_{m,6}\left(\frac{\rho}{\rho_{\text{m},4a}}\right)^{\varepsilon_{m,6}} E_{\varepsilon_{m,6}}\left(\frac{\rho}{\rho_{\text{m},6}}\right)\right]$$

where $E_n(z) = \int_1^\infty \frac{e^{-zt}}{t^n}dt$ represents the exponential integral function. The corresponding parameters have the following values:

- Coefficients: $\alpha_{\text{m},1} = 0.48962315$, $\alpha_{\text{m},2} = 0.24014465$, $\alpha_{\text{m},3} = 1.0932969$, $\alpha_{\text{m},4} = 0.08936644$, $\alpha_{\text{m},5} = 67.4598$, and $\alpha_{\text{m},6} = 1331.29$.
- Exponents: $\varepsilon_{\text{m},3a} = 1.56671$, $\varepsilon_{\text{m},3b} = 0.930273$, $\varepsilon_{\text{m},4} = 4.785$, $\varepsilon_{\text{m},5} = 166.594$, $\varepsilon_{\text{m},6} = 5.93118$, and $\varepsilon_{\text{reg},2b} = 5.248961$.
- Characteristic densities in g/cm$^3$: $\rho_{\text{m},2} = 1.35802$, $\rho_{\text{m},3} = 0.449618$, $\rho_{\text{m},4a} = 3.30149$, $\rho_{\text{m},4b} = 4.05911$, and $\rho_{\text{m},6} = 24.5967$, and the new value for $\rho_{\text{reg,Ronc}}$ is now equal to 2.22915.

It immediately follows that this new function needs more parameters than for Equation (7), but the global shape of the function $m_{\text{NIST}}(\rho)$ is very similar to that of Equation (7), except that this new function has a strong oscillation around the density value $\rho = 1.8$ g/cm$^3$. This oscillation is needed for a good representation of the data, but it is physically difficult to understand. Equation (7bis) is therefore of no practical use compared to Equation (7) and will not be analyzed further.

## 3. Thermodynamic Properties Derived from Isochoric Heat Capacity

Because Helmholtz free energy versus density and temperature is one of the four basic forms of an equation of state, we focus here on the process for deducing its expression. For this purpose, we used the thermodynamic relation:

$$C_V = \left(\frac{\partial U}{\partial T}\right)_V = T\left(\frac{\partial S}{\partial T}\right)_V = -T\left(\frac{\partial^2 F}{\partial T^2}\right)_V \tag{15}$$

Consequently, $F$ can be deduced from (i) two successive integrations of $C_V$ or (ii) a single integration of $C_V$ to calculate $U$ and $S$ and then use the thermodynamic relation:

$$F = U - TS \tag{16}$$

with $U(\rho, T) = \int C_V(\rho, T)dT + U_0(\rho) + \text{constant}$, $S(\rho, T) = \int \frac{C_V(\rho,T)}{T}dT + S_0(\rho) + \text{constant}$ and $C_V(\rho, T) = R_A\left(\widetilde{c}_V^o(T) + \widetilde{c}_V^r(\rho, T)\right)$ (given by Equations (1), (2), and (5)).

We chose the second approach that the two integrations to find $U$ and $S$ induces the existence of two arbitrary functions, $U_0(\rho)$ and $S_0(\rho)$, respectively, which are simpler to determine than directly finding the arbitrary function for $F$. The later simply writes $F_0(\rho, T) = U_0(\rho) - TS_0(\rho)$. It will be seen in Section 3.1 how the two arbitrary functions $U_0(\rho)$ and $S_0(\rho)$ can be determined.

There is no difficulty in finding a primitive of $\widetilde{c}_V^o(T)$ for $U$ or $S$. For the residual part of $C_V$ (see Equation (2)), there is also no difficulty in finding a primitive of $\widetilde{c}_{V,\text{reg}}^r$ and $\widetilde{c}_{V,\text{crit}}^r$. However, for $\widetilde{c}_{V,\text{reg}}^r$, when $T \gg T_c/\lambda$, two expressions can be obtained for the primitive of $U$ depending on whether the value of $m(\rho)$ is zero or not, that is to say, a power law if $m \neq 0$ and a logarithmic law if $m = 0$. It can be seen in Figure 3 that there are two values of $\rho$ for which $m = 0$, namely, $\rho_{\text{low}} = 0.11726382$ g/cm$^3$ and $\rho_{\text{high}} = 3.29510771$ g/cm$^3$. To obtain a single expression that is uniformly valid, the primitive is written as follows:

$$\int \widetilde{c}_{V,\text{reg}}^r(T \gg T_c/\lambda)dT = \frac{3}{2}n_{\text{reg}}(\rho)T_c\frac{(T/T_c)^{m(\rho)}-1}{m(\rho)} \tag{17}$$

Using the Hospital's rule, it can be easily verified that $\lim\limits_{m\to 0}\frac{(T/T_c)^{m(\rho)}-1}{m(\rho)} = \ln\left(\frac{T}{T_c}\right)$, which corresponds to the right expression for the primitive when $m = 0$.

The same problem occurs for the primitive of $S$, but this time, the expression depends on whether $m = 1$ or not. For argon, the value $m = 1$ is never reached, but to maintain a general expression, we proceed in the same manner to determine the expression for the primitive of $S$.

For the integration of $C_V$, the only term for which it may be difficult to find a primitive is $\tilde{c}^{\mathrm{r}}_{V,\mathrm{nonreg}}$ (see Equation (2)). It could be integrated numerically, but a reference state must be chosen; this will be shown in Section 3.4. To find a primitive, it is also possible to perform a series expansion of the term $(1 - x)^{-1}$ with $x = T_{\mathrm{div}}/T$. Hence, $\tilde{c}^{\mathrm{r}}_{V,\mathrm{nonreg}}$ can be written in the following form:

$$\tilde{c}^{\mathrm{r}}_{V,\mathrm{nonreg}}(\rho, x) = \frac{3}{2} n_{\mathrm{nonreg}}(\rho) \sum_{k=0}^{\infty} \exp\left(-x^{-3/2}\right) x^k, \; k \in \mathbb{N} \tag{18}$$

A primitive for each term of the series can be obtained. For a practical calculation, the series expansion must be truncated. The convergence is slower as $x$ approaches the unit value, and as a result, the number of terms that must be considered increases. An empirical formula for calculating the number of terms required is given below so that the residual error due to truncation is less than 0.1% (except for $x > 0.99$ because the function weakly diverges as $T \to T_{\mathrm{div}}$):

$$k_{\max}(x) = 1 + \left\lfloor 400 \frac{\exp[\exp(-8.52\,|1 - x|) - 1]}{1 + 16.6\,|1 - x|} \right\rfloor \tag{19}$$

where $\lfloor \bullet \rfloor$ represents the integer part function.

Finally, the equations for $U$ and $S$ can be written in standard dimensionless form (i.e., an ideal gas part and a residual one) as

$$\tilde{u}(\rho, T) = \frac{U(\rho, T)}{R_{\mathrm{A}} T} = \frac{3}{2}\left[1 + \frac{T_c}{\lambda_0 T} \exp\left(-\frac{\lambda_0 T}{T_c}\right)\right] + \tilde{u}^*(\rho, T) + \tilde{u}_0(\rho) \tag{20}$$

with

$$\begin{aligned}
\tilde{u}^*(\rho, T) &= \frac{3}{2} n_{\mathrm{reg}}(\rho) \frac{T_c}{T} \left\{ \frac{\left(\frac{T}{T_c}\right)^{m(\rho)} - 1}{m(\rho)} + \frac{\lambda^{-m(\rho)}}{2 - m(\rho)} \Gamma\left(\frac{m(\rho)}{2 - m(\rho)}, \left(\frac{\lambda T}{T_c}\right)^{2 - m(\rho)}\right) \right\} \\
&\quad - n_{\mathrm{nonreg}}(\rho) \frac{T_{\mathrm{div}}(\rho)}{T} \sum_{k=0}^{\infty} \Gamma\left(-\frac{2}{3}(k-1), \left(\frac{T_{\mathrm{div}}(\rho)}{T}\right)^{-3/2}\right) + \frac{3}{2} n_{\mathrm{crit}}(\rho) \frac{(T_{\mathrm{div}}(\rho)/T)^{\varepsilon_{\mathrm{crit}}(\rho)}}{1 - \varepsilon_{\mathrm{crit}}(\rho)}
\end{aligned} \tag{21}$$

and

$$\tilde{s}(\rho, T) = \frac{S(\rho, T)}{R_{\mathrm{A}}} = \frac{3}{2}\left[\ln(T) - \mathrm{Ei}\left(-\frac{\lambda_0 T}{T_c}\right)\right] + \tilde{s}^*(\rho, T) + \tilde{s}_0(\rho) \tag{22}$$

with

$$\begin{aligned}
\tilde{s}^*(\rho, T) &= \frac{3}{2} n_{\mathrm{reg}}(\rho) \left\{ \frac{\left(\frac{T}{T_c}\right)^{m(\rho)-1} - 1}{m(\rho) - 1} + \frac{\lambda^{1-m(\rho)}}{2 - m(\rho)} \Gamma\left(\frac{m(\rho)-1}{2 - m(\rho)}, \left(\frac{\lambda T}{T_c}\right)^{2 - m(\rho)}\right) \right\} \\
&\quad - n_{\mathrm{nonreg}}(\rho) \sum_{k=0}^{\infty} \Gamma\left(-\frac{2}{3}k, \left(\frac{T_{\mathrm{div}}(\rho)}{T}\right)^{-3/2}\right) - \frac{3}{2} n_{\mathrm{crit}}(\rho)\, \varepsilon_{\mathrm{crit}}(\rho)^{-1} \left(\frac{T_{\mathrm{div}}(\rho)}{T}\right)^{\varepsilon_{\mathrm{crit}}(\rho)}
\end{aligned} \tag{23}$$

where $\Gamma(a, z) = \int_z^{\infty} t^{a-1} \exp(-t)\,dt$ represents the incomplete gamma function, and $\mathrm{Ei}(z) = \int_{-z}^{\infty} \frac{\exp(-t)}{t}\,dt$ represents the exponential integral function.

In the ideal gas limit, the relations for internal energy and entropy must be written as

$$U^{\mathrm{o}}(T) = \frac{3}{2} R_{\mathrm{A}} T + U_0^{\mathrm{o}} \tag{24}$$

$$S^{\mathrm{o}}(\rho, T) = \frac{3}{2} R_{\mathrm{A}} \ln(T) - R_{\mathrm{A}} \ln(\rho) + S_0^{\mathrm{o}} \tag{25}$$

where $U_0^o$ and $S_0^o$ are arbitrary constants. These formulas can be rewritten as follows (using Equation (5) for $\widetilde{c}_V^o$):

$$U^o(T) = \frac{3}{2} R_A T \left[ 1 + \frac{T_c}{\lambda_0 T} \exp\left( -\frac{\lambda_0 T}{T_c} \right) \right] + U_0^o \tag{26}$$

$$S^o(\rho, T) = \frac{3}{2} R_A \left[ \ln(T) - \mathrm{Ei}\left( -\frac{\lambda_0 T}{T_c} \right) \right] - R_A \ln(\rho) + S_0^o \tag{27}$$

Equations (26) and (27) are used to express the Helmholtz free energy and its various derivatives as shown in Table 3.

**Table 3.** The ideal gas part $\widetilde{a}^\circ$ of the dimensionless Helmholtz free energy function and its derivatives.

$$\widetilde{a}^\circ = \frac{3}{2}\left[ 1 - \ln(T) + \frac{T_c}{\lambda_0 T}\exp\left(-\frac{\lambda_0 T}{T_c}\right) + \mathrm{Ei}\left(-\frac{\lambda_0 T}{T_c}\right) \right] + \ln(\rho) + \varsigma_0 \frac{T_c}{T} - \omega_0$$

$$T_c\left(\frac{\partial \widetilde{a}^\circ}{\partial T}\right)_\rho = -\frac{T_c}{T}\left[\frac{3}{2}\left(1 + \frac{T_c}{\lambda_0 T}\exp\left(-\frac{\lambda_0 T}{T_c}\right)\right) + \varsigma_0 \frac{T_c}{T}\right]$$

$$T_c^2\left(\frac{\partial^2 \widetilde{a}^\circ}{\partial T^2}\right)_\rho = \frac{3}{2}\left(\frac{T_c}{T}\right)^2\left(1 + \frac{4}{3}\frac{T_c}{T}\varsigma_0\right) + \frac{3}{2}\left(\frac{T_c}{T}\right)^2\left(1 + 2\frac{T_c}{\lambda_0 T}\right)\exp\left(-\frac{\lambda_0 T}{T_c}\right)$$

$$\rho_c\left(\frac{\partial \widetilde{a}^\circ}{\partial \rho}\right)_T = \frac{\rho_c}{\rho}$$

$$\rho_c^2\left(\frac{\partial^2 \widetilde{a}^\circ}{\partial \rho^2}\right)_T = -\left(\frac{\rho_c}{\rho}\right)^2$$

$$\left(\frac{\partial^2 \widetilde{a}^\circ}{\partial \rho \partial T}\right) = 0$$

The above expressions can now be used to rearrange Equations (20) and (22) in order to extract the residual part for the internal energy (i.e., $\widetilde{u}(\rho, T)$ minus the ideal gas part) and for the entropy:

$$\widetilde{u}^{\mathrm{r}}(\rho, T) = \widetilde{u}^*(\rho, T) + \widetilde{u}_0(\rho) - \varsigma_0 \frac{T_c}{T} \quad \text{with} \quad \varsigma_0 = \frac{U_0^o}{R_A T_c} \tag{28}$$

$$\widetilde{s}^{\mathrm{r}}(\rho, T) = \widetilde{s}^*(\rho, T) + \widetilde{s}_0(\rho) + \ln(\rho) - \omega_0 \quad \text{with} \quad \omega_0 = \frac{S_0^o}{R_A} \tag{29}$$

where $\varsigma_0$ and $\omega_0$ are two arbitrary constants. In view of fitting NIST data, the constant values must be such that $\varsigma_0 = -0.00070133$ and $\omega_0 = 2.71428$.

### 3.1. Determination of the Arbitrary Functions for Internal Energy and Entropy

The two arbitrary functions $U_0(\rho)$ and $S_0(\rho)$ can be determined in two different ways. One way is to find the difference between previously published data of $U$ (or $S$) and the $U$ (or $S$) values calculated by the present modeling and then find a function ($U_0(\rho)$ or $S_0(\rho)$) that best fits this difference. However, this method could be problematic as $U$ and $S$ are not measured quantities and depend on a chosen reference state. Another way is to use a new experimentally measured quantity, namely, pressure $P$, and by using the following relationship:

$$P = -\left(\frac{\partial F}{\partial V}\right)_T = \rho^2 \left(\frac{\partial F}{\partial \rho}\right)_T \tag{30}$$

Along isochors, from Equations (15) and (16), it is deduced that $F = \int C_V dT + U_0 - T\left(\int \frac{C_V}{T} dT + S_0\right)$, and its derivative versus $V$ (or $\rho$) gives $P - P_{C_V} = U_0' - T S_0'$, with $P_{C_V} = \frac{\partial}{\partial V}\left(-\int C_V dT + T\int \frac{C_V}{T} dT\right)$, $U_0' = \frac{\partial}{\partial V}(U_0)$, and $S_0' = \frac{\partial}{\partial V}(S_0)$. Here, $P_{C_V}$ is calculated from the $C_V$ values given by Equation (1). For a given isochor of density $\rho$, the difference $P - P_{C_V}$ must be a straight line (of slope $S_0'$ and ordinate at origin $U_0'$) if the $C_V$ values are well predicted by Equation (1). This is effectively observed in Figure 5, which displays $P - P_{C_V}$ versus $T$ on different isochors (i.e., the quasi-infinite curvature is a consequence of the extremely good representation of $C_V$ along isochors). The best and simplest functions that represent $U_0'(\rho)$ and $S_0'(\rho)$ are

$$\hat{u}'_0(\rho) = \frac{U'_0(\rho)}{\frac{3}{2}R_A T_c} = \frac{\alpha_{u,0}}{\rho_{t,Liq}} - \frac{1}{\rho}\left(\frac{\rho_{t,Liq}}{\rho}\right)\left[\alpha_{u,1}\exp\left(-\frac{\rho_{u,1}}{\rho}\right) - \alpha_{u,2}\left(\frac{\rho_{u,2}}{\rho_{t,Liq}}\right)^{\varepsilon_{u,2}}\left(\frac{\rho}{\rho_{u,2}-\rho}\right)^{\varepsilon_{u,2}}\right.$$
$$\left.+\alpha_{u,3}\left(\frac{\rho_{u,3}}{\rho_{t,Liq}}\right)^{\varepsilon_{u,3}}\left(\frac{\rho}{\rho_{u,3}-\rho}\right)^{\varepsilon_{u,3}} + \alpha_{u,4}\left(\frac{\rho}{\rho_{t,Liq}}\right)^{\varepsilon_{u,4}} - \alpha_{u,5}\left(\frac{\rho}{\rho_{t,Liq}}\right)^3\exp\left(-\left(\frac{\rho_{u,1}}{\rho}\right)^2\right) + \alpha_{u,6}\left(\frac{\rho}{\rho+\rho_c}\right)^{\varepsilon_{u,6}}\right] \tag{31}$$

and

$$\widetilde{s}'_0(\rho) = \frac{S'_0(\rho)}{R_A} = -\frac{1}{\rho}\left[1 + \alpha_{s,1}\left(\frac{\rho}{\rho_{t,Liq}}\right)^{\varepsilon_{s,1}-1}\left(\frac{\rho_{s,1}}{\rho+\rho_{s,1}}\right)^{\varepsilon_{s,1}} - \alpha_{s,2}\left(\frac{\rho}{\rho_{t,Liq}}\right)^{\varepsilon_{s,2}-1}\ln\left(\frac{\rho}{\rho_{t,Liq}}\right) + \alpha_{s,3}\left(\frac{\rho}{\rho_{t,Liq}}\right)^{\varepsilon_{s,3}-1}\ln\left(\frac{\rho}{\rho_{t,Liq}}\right)\right.$$
$$\left.+\alpha_{s,4}\left(\frac{\rho}{\rho_{t,Liq}}\right)^{\varepsilon_{s,4}-1}\exp\left(-\frac{\rho}{\rho_{s,4}}\right) + \alpha_{s,5}\left(\frac{\rho}{\rho_{t,Liq}}\right)\exp\left(-\left(\frac{\rho}{\rho_{s,1}}\right)^{\varepsilon_{s,5}}\right) + \alpha_{s,6}\left(\frac{\rho}{\rho_{t,Liq}}\right)\exp\left(-\left(\frac{\rho-\rho_{s,4}}{\rho_c}\right)^2\right)\right]$$
$$+\widetilde{s}'_{0,Ronc}(\rho) \tag{32}$$

with

$$\widetilde{s}'_{0,Ronc}(\rho) = \frac{1}{\rho_{sRonc,1}}\left[\alpha_{sRonc,1}\left(\frac{\rho}{\rho_{sRonc,1}}\right)^{\varepsilon_{sRonc,1}} - \alpha_{sRonc,2}\left(\frac{\rho}{\rho_{sRonc,1}}\right)\exp\left(-\frac{\rho_{sRonc,2}}{\rho}\right)\right]\left[1 - \exp\left(-\left(\frac{\rho}{\rho_c}\right)^2\right)\right] \tag{33}$$

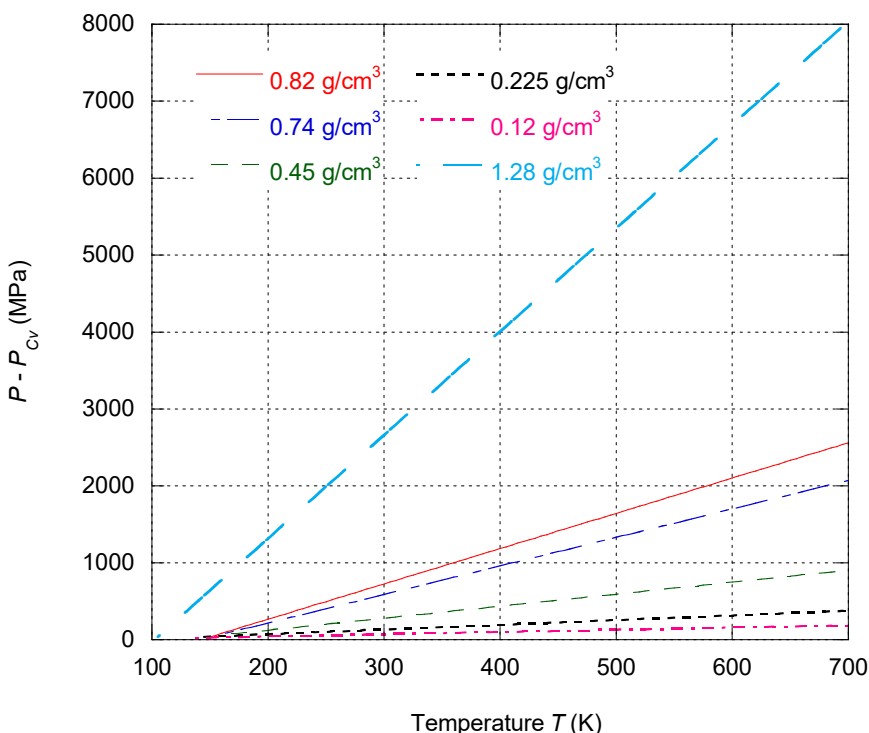

**Figure 5.** Variation of $P - P_{C_V}$ as a function of temperature for different isochors such that $\rho < \rho_{t,Liq}$.

Before continuing, it is worth noting that $\lim_{\rho\to 0}\frac{\rho U'_0(\rho)}{\frac{3}{2}R_A T_c} = 0$ and $\lim_{\rho\to 0}\frac{\rho S'_0(\rho)}{R_A} = -1$.

A primitive of expressions (31) and (32) leads to the functions $U_0(\rho)$ and $S_0(\rho)$ such that

$$\hat{u}_0(\rho) = \frac{U_0(\rho)}{\frac{3}{2}R_A T_c} = \alpha_{u,0}\frac{\rho}{\rho_{t,Liq}} - \frac{\alpha_{u,1}}{\varepsilon_{u,1}-1}\left(\frac{\rho_{t,Liq}}{\rho_{u,1}}\right)\exp\left(-\frac{\rho_{u,1}}{\rho}\right) + \frac{\alpha_{u,2}}{\varepsilon_{u,2}-1}\left(\frac{\rho_{u,2}}{\rho_{t,Liq}}\right)^{\varepsilon_{u,2}-1}\left(\frac{\rho}{\rho_{u,2}-\rho}\right)^{\varepsilon_{u,2}-1}$$
$$-\frac{\alpha_{u,3}}{\varepsilon_{u,3}-1}\left(\frac{\rho_{u,3}}{\rho_{t,Liq}}\right)^{\varepsilon_{u,3}-1}\left(\frac{\rho}{\rho_{u,3}-\rho}\right)^{\varepsilon_{u,3}-1} - \frac{\alpha_{u,4}}{\varepsilon_{u,4}-1}\left(\frac{\rho}{\rho_{t,Liq}}\right)^{\varepsilon_{u,4}-1}$$
$$+\frac{1}{2}\alpha_{u,5}\left(\frac{\rho}{\rho_{t,Liq}}\right)^2\left[\exp\left(-\left(\frac{\rho_{u,1}}{\rho}\right)^2\right) - \left(\frac{\rho_{u,1}}{\rho}\right)^2\Gamma\left(0,\left(\frac{\rho_{u,1}}{\rho}\right)^2\right)\right]$$
$$-\frac{\alpha_{u,6}}{\varepsilon_{u,6}-1}\left(\frac{\rho_{t,Liq}}{\rho_c}\right)\left(\frac{\rho}{\rho+\rho_c}\right)^{\varepsilon_{u,6}-1} \tag{34}$$

and

$$\widetilde{s}_0(\rho) = \frac{S_0(\rho)}{R_A} = -\ln\left(\frac{\rho}{\rho_{t,Liq}}\right) - \frac{\alpha_{s,1}}{\varepsilon_{s,1}-1}\left(\frac{\rho}{\rho_{t,Liq}}\right)^{\varepsilon_{s,1}-1}\left(\frac{\rho_{s,1}}{\rho+\rho_{s,1}}\right)^{\varepsilon_{s,1}-1}$$
$$+ \frac{\alpha_{s,2}}{(\varepsilon_{s,2}-1)^2}\left(\frac{\rho}{\rho_{t,Liq}}\right)^{\varepsilon_{s,2}-1}\left[-1+(\varepsilon_{s,2}-1)\ln\left(\frac{\rho}{\rho_{t,Liq}}\right)\right]$$
$$- \frac{\alpha_{s,3}}{(\varepsilon_{s,3}-1)^2}\left(\frac{\rho}{\rho_{t,Liq}}\right)^{\varepsilon_{s,3}-1}\left[-1+(\varepsilon_{s,3}-1)\ln\left(\frac{\rho}{\rho_{t,Liq}}\right)\right] - \alpha_{s,4}\left(\frac{\rho}{\rho_{t,Liq}}\right)^{\varepsilon_{s,4}-1}E_{2-\varepsilon_{s,4}}\left(\frac{\rho}{\rho_{s,4}}\right)$$
$$+ \frac{\alpha_{s,5}}{\varepsilon_{s,5}}\left(\frac{\rho}{\rho_{t,Liq}}\right)E_{\frac{\varepsilon_{s,4}-1}{\varepsilon_{s,4}}}\left(\left(\frac{\rho}{\rho_{s,1}}\right)^{\varepsilon_{s,5}}\right) - \frac{\sqrt{\pi}}{2}\alpha_{s,6}\left(\frac{\rho_c}{\rho_{t,Liq}}\right)\mathrm{erf}\left(\frac{\rho-\rho_{s,4}}{\rho_c}\right) + \widetilde{s}_{0,Ronc}(\rho) \tag{35}$$

with

$$\widetilde{s}_{0,Ronc}(\rho) = \frac{\alpha_{sRonc,1}}{2(\varepsilon_{sRonc,1}+1)}\left(\frac{\rho}{\rho_{sRonc,1}}\right)^{1+\varepsilon_{sRonc,1}}\left[2+(1+\varepsilon_{sRonc,1})E_{\frac{1-\varepsilon_{sRonc,1}}{2}}\left(\left(\frac{\rho}{\rho_c}\right)^2\right)\right]$$
$$- \frac{\alpha_{sRonc,2}}{\rho_{sRonc,1}}\int_0^\rho\left(\frac{t}{\rho_{sRonc,1}}\right)\exp\left(-\frac{\rho_{sRonc,2}}{t}\right)\left(1-\exp\left(-\left(\frac{t}{\rho_c}\right)^2\right)\right)dt \tag{36}$$

where $E_n(z) = \int_1^\infty \frac{e^{-zt}}{t^n}dt$ represents the exponential integral function, and $\mathrm{erf}(x)$ represents the error function.

The coefficient and exponent values appearing in these equations are given in Table 4. The density dependence of the terms $\widetilde{u}_0$ and $\widetilde{s}_0$ are shown in Figure 6.

**Table 4.** Coefficients and exponents for $\hat{u}_0$ and $\widetilde{s}_0$.

| $i$ | $\varepsilon_i$ | $\alpha_i$ |
|---|---|---|
| u,0 | | 16.86969325 |
| u,1 | | 71.08169282 |
| u,2 | 2.57795090 | 12.16671437 |
| u,3 | 2.01916041 | 22.41395798 |
| u,4 | 12.94678106 | 0.13352634 |
| u,5 | | 16612.44198645 |
| u,6 | 1.78738624 | 0.04855950 |
| s,1 | 2.23951150 | 11.75732913 |
| s,2 | 3.18259094 | 9.91697667 |
| s,3 | 2.71140252 | 12.27973100 |
| s,4 | 1.55994791 | 0.04075918 |
| s,5 | 21.47158258 | 0.31499626 |
| s,6 | | 0.46391511 |
| sRonc,1 | 62.32164244 | 57.01690712 |
| sRonc,2 | | 187.65045674 |

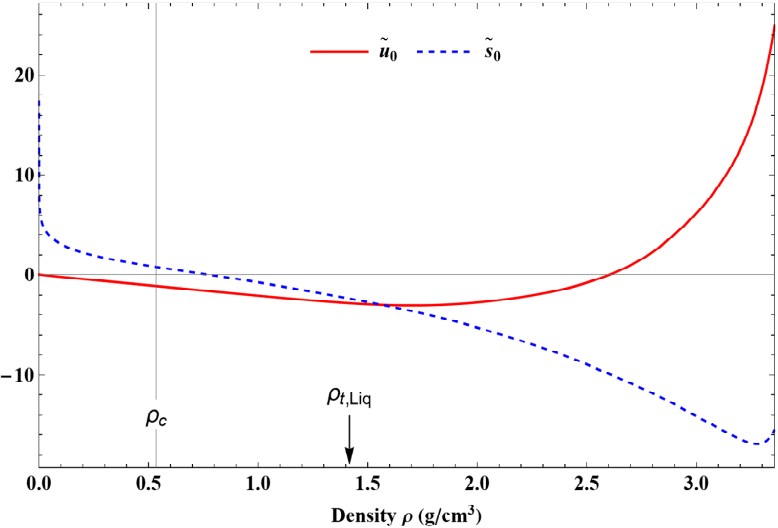

**Figure 6.** Variations with density of the functions $\widetilde{u}_0(\rho)$ (red curve) and $\widetilde{s}_0(\rho)$ (blue dashed curve) between $\rho = 0$ and $\rho = \rho_{max,Ronc}$.

It is important to remember that the two primitives above depend on an arbitrary constant $\alpha_{u0}$ and $\alpha_{s0}$, respectively. Moreover, it must be ensured that the dimensionless form for internal energy is such that

$$\widetilde{u}_0(\rho) = \frac{T_c}{T} \left[ \frac{3}{2} \hat{u}_0(\rho) + \alpha_{u0} \right] \tag{37}$$

and for the entropy,

$$\widetilde{s}_0^*(\rho) = \widetilde{s}_0(\rho) + \ln(\rho) \tag{38}$$

From Equations (20), (22), and (34)–(38), the expression for Helmholtz free energy can be easily deduced. In dimensionless form, this one writes as follows:

$$\widetilde{a}(\rho, T) = \frac{F}{R_A T} = \frac{U}{R_A T} - \frac{S}{R_A} = \widetilde{u}(\rho, T) - \widetilde{s}(\rho, T) = \widetilde{a}^o(\rho, T) + \widetilde{a}^r(\rho, T) \tag{39}$$

where $\widetilde{a}^o(\rho, T)$ is given in Table 3 and

$$\widetilde{a}^r(\rho, T) = \underbrace{\widetilde{a}_{\text{reg}}^r(\rho, T) + \widetilde{a}_{\text{nonreg}}^r(\rho, T) + \widetilde{a}_{\text{crit}}^r(\rho, T)}_{\widetilde{u}^* - \widetilde{s}^*} + \widetilde{a}_0^r(\rho, T) \tag{40}$$

with

$$\widetilde{a}_0^r(\rho, T) = \widetilde{u}_0(\rho, T) - [\widetilde{s}_0^*(\rho) - \omega_0] - \varsigma_0 \frac{T_c}{T} = \frac{T_c}{T} \left[ \frac{3}{2} \hat{u}_0(\rho) + \alpha_{u0} - \varsigma_0 \right] - [\widetilde{s}_0^*(\rho) + \alpha_{s0} - \omega_0] \tag{41}$$

For the sake of simplicity, the two constants $\alpha_{u0}$ and $\alpha_{s0}$ can be chosen such that $\alpha_{u0} = \varsigma_0$ and $\alpha_{s0} = \omega_0$. It follows that

$$\widetilde{a}_0^r(\rho, T) = \frac{3}{2} \frac{T_c}{T} \hat{u}_0(\rho) - \widetilde{s}_0^*(\rho) \tag{42}$$

and

$$\widetilde{u}^r(\rho, T) = \widetilde{u}^*(\rho, T) + \frac{3}{2} \frac{T_c}{T} \hat{u}_0(\rho) \tag{43}$$

$$\widetilde{s}^r(\rho, T) = \widetilde{s}^*(\rho, T) + \widetilde{s}_0^*(\rho) \tag{44}$$

It appears that if one wants to represent the data from NIST instead of the data of Ronchi for densities higher than $\rho_{t,\text{Liq}}$, it is only necessary to change the mathematical form of Equations (31)–(33). For example, the new function for $\hat{u}_0'(\rho)$ can be written with the same mathematical terms as in Equation (31) but without the two last terms and with different values of the parameters. Therefore, it can be understood that the global shape of the new functions $\hat{u}_0'(\rho)$ and $\hat{s}_0'(\rho)$ will have very similar variations. This remark also shows that the two data sets discussed above for high densities can be represented by only small variations in the shape of the two derivative functions $\hat{u}_0'(\rho)$ and $\hat{s}_0'(\rho)$. Once these two functions are determined, all the thermodynamic equations of state are known.

Table 5 summarizes the various terms making up the residual part of the Helmholtz free energy.

**Table 5.** Mathematical expressions of the dimensionless terms in the residual part $\widetilde{a}^r$ of Helmholtz free energy for $T \geq T_{\text{div}}$.

$$\widetilde{a}_{\text{reg}}^r = \frac{3}{2} n_{\text{reg}} \frac{T_c}{T} \left\{ \frac{\left(\frac{T}{T_c}\right)^m - 1}{m} + \frac{\lambda^{-m}}{2-m} \Gamma\left(\frac{m}{2-m}, \left(\frac{\lambda T}{T_c}\right)^{2-m}\right) \right\}$$

$$\quad - \frac{3}{2} n_{\text{reg}} \left\{ \frac{\left(\frac{T}{T_c}\right)^{m-1} - 1}{m-1} + \frac{\lambda^{1-m}}{2-m} \Gamma\left(\frac{m-1}{2-m}, \left(\frac{\lambda T}{T_c}\right)^{2-m}\right) \right\}$$

$$\widetilde{a}_{\text{nonreg}}^r = n_{\text{nonreg}} \sum_{k=0}^{\infty} \left\{ \Gamma\left(-\frac{2}{3}k, \left(\frac{T_{div}}{T}\right)^{-3/2}\right) - \frac{T_{div}}{T} \Gamma\left(-\frac{2}{3}(k-1), \left(\frac{T_{div}}{T}\right)^{-3/2}\right) \right\}$$

$$\widetilde{a}_{\text{crit}}^r = \frac{3}{2} n_{\text{crit}} \frac{(T_{\text{div}}/T)^{\varepsilon_{\text{crit}}}}{(1-\varepsilon_{crit})\, \varepsilon_{crit}}$$

$$\widetilde{a}_0^r = \frac{3}{2} \frac{T_c}{T} \hat{u}_0(\rho) - \{\widetilde{s}_0(\rho) + \ln(\rho)\}$$

### 3.2. Analytic Expression of the Thermal Equation of State for $T \geq T_{div}$

The thermal state equation $P = P(\rho, T)$, which is a fundamental equation to calculate the basic thermal properties of argon, can easily be established using Equations (30) and (39) for free energy. Free energy is made up of four terms coming from the residual part of Helmholtz free energy and a term that represents the behavior of the ideal gas:

$$P(\rho, T) = P_{\text{reg}}(\rho, T) + P_{\text{nonreg}}(\rho, T) + P_{\text{crit}}(\rho, T) + P_0(\rho, T) + \rho R_A T \quad (45)$$

or in a dimensionless form

$$Z = \frac{P}{\rho R_A T} = \underbrace{\frac{P_{\text{reg}}(\rho, T)}{\rho R_A T}}_{Z_{\text{reg}}} + \underbrace{\frac{P_{\text{nonreg}}(\rho, T)}{\rho R_A T}}_{Z_{\text{nonreg}}} + \underbrace{\frac{P_{\text{crit}}(\rho, T)}{\rho R_A T}}_{Z_{\text{crit}}} + \underbrace{\frac{P_0(\rho, T)}{\rho R_A T}}_{Z_0} + 1 \quad (46)$$

$$Z_{\text{reg}} = \rho \left( \frac{\partial \widetilde{a}_{\text{reg}}^{\text{r}}}{\partial \rho} \right)_T$$

with

$$Z_{\text{nonreg}} = \rho \left( \frac{\partial \widetilde{a}_{\text{nonreg}}^{\text{r}}}{\partial \rho} \right)_T = -\rho \, n'_{\text{nonreg}}(\rho) \sum_{k=0}^{\infty} \left[ x \, \Gamma\left( -\frac{2}{3}(k-1), x^{-3/2} \right) - \Gamma\left( -\frac{2}{3}k, x^{-3/2} \right) \right]$$
$$- n_{\text{nonreg}}(\rho) \frac{\rho \, T'_{\text{div}}(\rho)}{T} \sum_{k=0}^{\infty} \Gamma\left( -\frac{2}{3}(k-1), x^{-3/2} \right) \quad (47)$$

$$Z_0 = \rho \left( \frac{\partial \widetilde{a}_0^{\text{r}}}{\partial \rho} \right)_T + 1 = \rho \left[ \frac{3}{2} \frac{T_c}{T} \hat{u}'_0(\rho) - \hat{s}'_0(\rho) \right] \quad (48)$$

$$Z_{\text{crit}} = \rho \left( \frac{\partial \widetilde{a}_{\text{crit}}^{\text{r}}}{\partial \rho} \right)_T = \frac{3}{2}\rho \, n'_{\text{crit}}(\rho) \frac{(T_{\text{div}}(\rho)/T)^{\varepsilon_{\text{crit}}(\rho)}}{\varepsilon_{\text{crit}}(\rho) \, (1-\varepsilon_{\text{crit}}(\rho))}$$
$$+ \frac{3}{2}\rho \, n_{\text{crit}}(\rho) \left\{ \frac{2\varepsilon_{\text{crit}}(\rho)-1}{\varepsilon_{\text{crit}}(\rho)^2 (\varepsilon_{\text{crit}}(\rho)-1)^2} \varepsilon'_{\text{crit}}(\rho) \left( \frac{T_{\text{div}}(\rho)}{T} \right)^{\varepsilon_{\text{crit}}(\rho)} \right.$$
$$\left. - \frac{(T_{\text{div}}(\rho)/T)^{\varepsilon_{\text{crit}}(\rho)}}{\varepsilon_{\text{crit}}(\rho) \, (\varepsilon_{\text{crit}}(\rho)-1)} \left[ \varepsilon'_{\text{crit}}(\rho) \ln\left( \frac{T_{\text{div}}(\rho)}{T} \right) + \varepsilon_{\text{crit}}(\rho) \frac{T'_{\text{div}}(\rho)}{T_{\text{div}}(\rho)} \right] \right\} \quad (49)$$

It is recalled that $x = T_{\text{div}}/T$ in the expression of $Z_{\text{nonreg}}$. $Z_{\text{reg}}(\rho, T)$ displays too many terms, and its expression is given in Appendix A. The expressions of the first derivatives of Equations (6)–(12) are listed in Appendix B. From the expression of these factors, it is easy to see that $Z_{\text{reg}} = Z_{\text{nonreg}} = Z_{\text{crit}} = 0$ and $Z_0 = 1$ for $\rho \to 0$; therefore, $Z \to 1$ for any temperature when density tends to zero. In a certain range of temperatures, isotherms intersect the line $Z = 1$ for $\rho$ values that are not identically zero. As thermodynamic quantities corresponding to $Z = 1$ are physically important and not easy to find in the literature, they are listed in Table 6.

**Table 6.** Thermodynamic properties corresponding to $Z = 1$ (i.e., ideal curve) deduced from Equation (46).

| $T/T_c$ | $P/P_c$ | $\rho/\rho_c$ | $C_V/R_A$ | $C_P/R_A$ | $c / \sqrt{\frac{R_A T}{M}}$ |
|---|---|---|---|---|---|
| 0.56408 | 5.3565 | 2.7491 | 2.7545 | 5.1425 | 7.1030 |
| 0.62500 | 5.7641 | 2.6699 | 2.6239 | 5.0445 | 6.5008 |
| 0.69444 | 6.1935 | 2.5819 | 2.4987 | 4.9633 | 5.9013 |
| 0.76389 | 6.5850 | 2.4956 | 2.3942 | 4.9091 | 5.3765 |
| 0.83333 | 6.9372 | 2.4100 | 2.3054 | 4.8869 | 4.9093 |
| 0.90278 | 7.2480 | 2.3242 | 2.2290 | 4.8871 | 4.4926 |
| 0.97222 | 7.5154 | 2.2379 | 2.1626 | 4.8993 | 4.1212 |
| 1.0417 | 7.7380 | 2.1505 | 2.1044 | 4.9136 | 3.7909 |
| 1.1111 | 7.9149 | 2.0622 | 2.0532 | 4.9213 | 3.4978 |
| 1.1806 | 8.0462 | 1.9731 | 2.0078 | 4.9165 | 3.2384 |

**Table 6.** *Cont.*

| $T/T_c$ | $P/P_c$ | $\rho/\rho_c$ | $C_V/R_A$ | $C_P/R_A$ | $c\left/\sqrt{\frac{R_A T}{M}}\right.$ |
|---|---|---|---|---|---|
| 1.2500 | 8.1324 | 1.8835 | 1.9673 | 4.8967 | 3.0090 |
| 1.3194 | 8.1744 | 1.7935 | 1.9309 | 4.8621 | 2.8062 |
| 1.3889 | 8.1729 | 1.7036 | 1.8979 | 4.8148 | 2.6267 |
| 1.4583 | 8.1284 | 1.6136 | 1.8676 | 4.7565 | 2.4674 |
| 1.5278 | 8.0406 | 1.5236 | 1.8395 | 4.6881 | 2.3258 |
| 1.5972 | 7.9091 | 1.4335 | 1.8133 | 4.6093 | 2.1994 |
| 1.6667 | 7.7330 | 1.3432 | 1.7887 | 4.5188 | 2.0865 |
| 1.7361 | 7.5118 | 1.2526 | 1.7653 | 4.4151 | 1.9854 |
| 1.8056 | 7.2453 | 1.1617 | 1.7429 | 4.2976 | 1.8947 |
| 1.8750 | 6.9343 | 1.0707 | 1.7214 | 4.1672 | 1.8133 |
| 1.9444 | 6.5801 | 0.97968 | 1.7005 | 4.0261 | 1.7402 |
| 2.0139 | 6.1845 | 0.88903 | 1.6803 | 3.8775 | 1.6743 |
| 2.0833 | 5.7492 | 0.79890 | 1.6605 | 3.7247 | 1.6150 |
| 2.1528 | 5.2754 | 0.70942 | 1.6412 | 3.5712 | 1.5617 |
| 2.2222 | 4.7635 | 0.62056 | 1.6223 | 3.4196 | 1.5137 |
| 2.2917 | 4.2127 | 0.53218 | 1.6038 | 3.2719 | 1.4707 |
| 2.3611 | 3.6210 | 0.44397 | 1.5857 | 3.1294 | 1.4322 |
| 2.4306 | 2.9847 | 0.35551 | 1.5682 | 2.9927 | 1.3977 |
| 2.5000 | 2.2996 | 0.26630 | 1.5511 | 2.8616 | 1.3667 |
| 2.5500 | 1.7737 | 0.20137 | 1.5389 | 2.7700 | 1.3462 |
| 2.6000 | 1.2211 | 0.13596 | 1.5266 | 2.6803 | 1.3270 |
| 2.6500 | 0.66245 | 0.072369 | 1.5144 | 2.5944 | 1.3095 |
| 2.6900 | 0.29167 | 0.031390 | 1.5061 | 2.5396 | 1.2987 |
| 2.7100 | 0.14787 | 0.015796 | 1.5029 | 2.5193 | 1.2947 |
| 2.7125 | 0.11575 | 0.012354 | 1.5022 | 2.5149 | 1.2939 |

*3.3. The Liquid–Vapor Coexistence Curve*

At a given temperature $T$, vapor pressure and densities of the coexisting phases can be determined by the simultaneous resolution of the equations:

$$\frac{P_{\text{sat}}}{\rho_{\sigma l} R_A T} = 1 + \rho_{\sigma l}\left(\frac{\partial \widetilde{a}_{\sigma l}^{\text{r}}}{\partial \rho_{\sigma l}}\right)_T = Z(\rho_{\sigma l}, T) \tag{50}$$

$$\frac{P_{\text{sat}}}{\rho_{\sigma v} R_A T} = 1 + \rho_{\sigma v}\left(\frac{\partial \widetilde{a}_{\sigma v}^{\text{r}}}{\partial \rho_{\sigma v}}\right)_T = Z(\rho_{\sigma v}, T) \tag{51}$$

$$\frac{P_{\text{sat}}}{R_A T}\left(\frac{1}{\rho_{\sigma v}} - \frac{1}{\rho_{\sigma l}}\right) - \ln\left(\frac{\rho_{\sigma l}}{\rho_{\sigma v}}\right) = \widetilde{a}^{\text{r}}(\rho_{\sigma l}, T) - \widetilde{a}^{\text{r}}(\rho_{\sigma v}, T) \tag{52}$$

where the indices $\sigma l$ and $\sigma v$ represent the liquid and the vapor coexistence states, respectively.

These equations represent the phase equilibrium conditions, i.e., the equality of pressure, temperature, and specific Gibbs energy (Maxwell criterion) in the coexisting phases. The calculated values on the liquid–vapor coexistence curve (vapor pressure, saturated liquid density, saturated vapor density, etc.) are given in Table 12. Approximate formulas for representing the pressure and densities of liquid and vapor as a function of temperature along the coexistence curve are given in Appendix C.

*3.4. Thermodynamic State inside the Liquid–Vapor Coexistence Curve for $T < T_{div}(\rho)$*

The thermodynamic properties of argon are calculated from the isochoric heat capacity equation $C_V(\rho, T)$. However, with the equation only being valid for $T \geq T_{\text{div}}$, a new equation, valid for $T < T_{\text{div}}$ (i.e., inside the liquid–vapor coexistence region), has to be established. This requires solving three mathematical problems.

- First, an expression of $C_V(\rho, T)$ for $T < T_{\text{div}}(\rho)$ has to be found.

- Secondly, to integrate $C_V$, the artificial divergence introduced with the term $\widetilde{c}^{\,\mathrm{r}}_{V,\mathrm{nonreg}}$ must be removed in order to have a finite value of $C_V$ for $T = T_{\mathrm{div}}(\rho)$.
- Finally, for the integration of $C_V$, a reference state must also be chosen.

The procedure used to develop the modified equation is now presented. It will be shown that this new formulation leads to a better description of the two-phase thermodynamic properties than the polynomial approach.

### 3.4.1. Expression of $C_V$

The two terms in $C_V(\rho, T)$ creating difficulties are $\widetilde{c}^{\,\mathrm{r}}_{V,\mathrm{nonreg}}$ and $\widetilde{c}^{\,\mathrm{r}}_{V,\mathrm{crit}}$. For $T < T_{\mathrm{div}}$, $\widetilde{c}^{\,\mathrm{r}}_{V,\mathrm{nonreg}}$ becomes negative, which has no physical meaning; indeed, the thermodynamic thermal stability has always to be satisfied. And the term $\widetilde{c}^{\,\mathrm{r}}_{V,\mathrm{crit}}$, for $T < T_{\mathrm{div}}$, diverges when $T \to 0$, which also has no physical meaning. The easiest way to solve these problems is to take a symmetric function by changing the variable $T_{\mathrm{div}}/T$ into $T/T_{\mathrm{div}}$; hence, the following is obtained:

$$
\underset{\substack{V,\mathrm{nonreg}\\ \mathrm{inside}}}{\widetilde{c}^{\,\mathrm{r}}}(\rho, T < T_{\mathrm{div}}) = \frac{3}{2} n_{\mathrm{nonreg}}(\rho) \exp\left[-\left(\frac{T}{T_{\mathrm{div}}(\rho)}\right)^{-3/2}\right] \frac{1}{1 - \frac{T}{T_{\mathrm{div}}(\rho)}} \tag{53}
$$

$$
\underset{\substack{V,\mathrm{crit}\\ \mathrm{inside}}}{\widetilde{c}^{\,\mathrm{r}}}(\rho, T < T_{\mathrm{div}}) = \frac{3}{2} n_{\mathrm{crit}}(\rho) \left(\frac{T}{T_{\mathrm{div}}(\rho)}\right)^{\varepsilon_{\mathrm{crit}}(\rho)} \tag{54}
$$

However, a problem remains as the two equations ($\widetilde{c}^{\,\mathrm{r}}_{V,\mathrm{nonreg}}$ and $\underset{\substack{V,\mathrm{nonreg}\\ \mathrm{inside}}}{\widetilde{c}^{\,\mathrm{r}}}$) become infinite for $T = T_{\mathrm{div}}$. In fact, this is the consequence of the extensive nature of $C_V$. Therefore, this divergence can be removed by explicitly introducing a finite number $N_V$ of particles into the equations for $C_V$. $N_V$ has to be the largest possible without being infinite (which is the condition for an extensive property). Then, as these equations must converge for $T = T_{\mathrm{div}}$, the terms $\frac{1}{1-\frac{T}{T_{\mathrm{div}}}}$ and $\frac{1}{1-\frac{T_{\mathrm{div}}}{T}}$ have to be corrected so that the two equations must tend to the same finite value for $T = T_{\mathrm{div}}$. The following functions have the required properties:

- $\frac{1}{1-\frac{T}{T_{\mathrm{div}}}}$ is replaced by $\frac{1-N_V^{-(1-\frac{T}{T_{\mathrm{div}}})}}{1-\frac{T}{T_{\mathrm{div}}}}$, so $\widetilde{c}^{\,\mathrm{r}}_{V,\mathrm{nonreg}}$ now becomes

$$
\underset{\substack{V,\mathrm{nonreg}\\ \mathrm{outside}}}{\widetilde{c}^{\,\mathrm{r}}} = \frac{3}{2} n_{\mathrm{nonreg}}(\rho) \exp\left[-\left(\frac{T_{\mathrm{div}}(\rho)}{T}\right)^{-3/2}\right] \frac{1-N_V^{-(1-\frac{T}{T_{\mathrm{div}}})}}{1-\frac{T}{T_{\mathrm{div}}}} \tag{55}
$$

- and $\frac{1}{1-\frac{T_{\mathrm{div}}}{T}}$ is replaced by $\frac{1-N_V^{-(1-\frac{T_{\mathrm{div}}}{T})}}{1-\frac{T_{\mathrm{div}}}{T}}$, so $\underset{\substack{V,\mathrm{nonreg}\\ \mathrm{inside}}}{\widetilde{c}^{\,\mathrm{r}}}$ now becomes

$$
\underset{\substack{V,\mathrm{nonreg}\\ \mathrm{inside}}}{\widetilde{c}^{\,\mathrm{r}}} = \frac{3}{2} n_{\mathrm{nonreg}}(\rho) \exp\left[-\left(\frac{T}{T_{\mathrm{div}}(\rho)}\right)^{-3/2}\right] \frac{1-N_V^{-(1-\frac{T_{\mathrm{div}}}{T})}}{1-\frac{T_{\mathrm{div}}}{T}} \tag{56}
$$

The two corrections tend to $\ln(N_V)$ when $T \to T_{\mathrm{div}}$. $N_V$ may be thought of as a quantity representing the number of particles in the volume $V$ for a given experiment, so it can be written as

$$
N_V = fmol\, \mathfrak{N}_a \frac{\rho}{\rho_c} \tag{57}
$$

where $fmol = 10^{20}$ is an arbitrary constant required to remove the divergence. This means that, near the transition, $C_V$ and its related quantities are no longer extensive quantities. This is not surprising because sample size effects are known to exist around the phase transition. Thus, the divergence occurs only for an infinite number of particles. This non-

extensive contribution introduced in Equations (55) and (56) has also been used to revisit liquid physics in order to explain rheological behavior under a wide variety of thermodynamic and mechanical conditions [8–11].

Outside the liquid–vapor coexistence region, the percentage deviation between $\tilde{c}^{\,\mathrm{r}}_{V,\mathrm{nonreg}}$ calculated by Equation (18), and $\tilde{c}^{\,\mathrm{r}}_{V,\mathrm{nonreg}}$, calculated by Equation (55), is shown in Figure 7. It is observed that the difference is only significant in the close vicinity of the critical point.

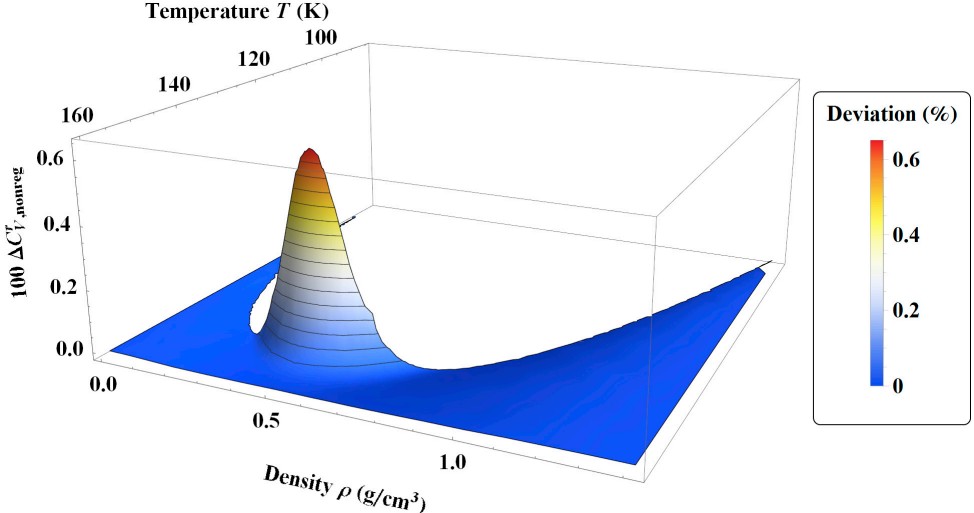

**Figure 7.** Percentage deviations of the non-regular term of the isochoric heat capacity $\Delta C^r_{V,\mathrm{nonreg}} =$

$$\left( \underset{\substack{V,\mathrm{nonreg} \\ \text{Eq. (18)}}}{C^r} - \underset{\substack{V,\mathrm{nonreg} \\ \text{Eq. (55)}}}{C^r} \right) \Bigg/ \underset{\substack{V,\mathrm{nonreg} \\ \text{Eq. (18)}}}{C^r} \quad \text{for the single-phase region in the temperature range of } T_t$$

to 160 K and density range of $\rho_{t,\mathrm{Gas}}$ to $\rho_{t,\mathrm{Liq}}$.

Figure 8 shows the behavior of $C_V$ on two isotherms that are crossing the coexistence phase. One can observe that, on both isotherms, the new model always gives positive values of $C_V$ with maximum values, as has been experimentally observed. It can be noticed that the TSW model leads to erroneous $C_V$ variations in this coexistence region.

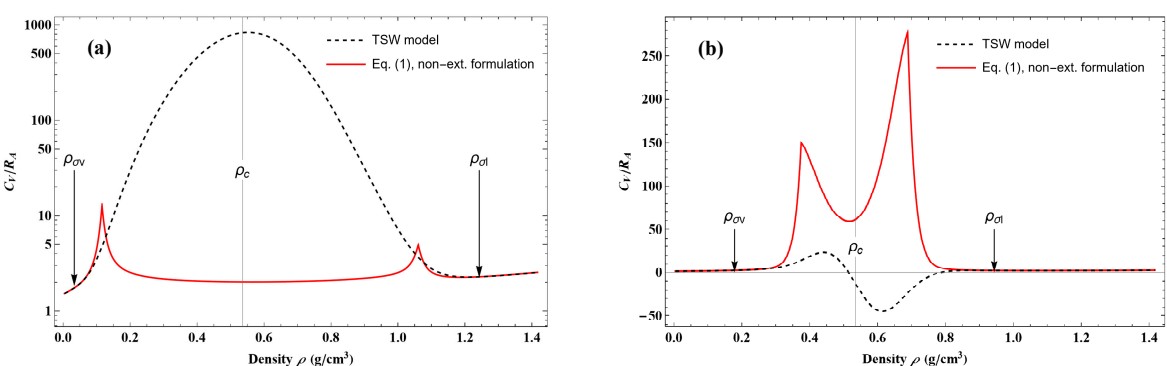

**Figure 8.** Variations with density of the dimensionless isochoric heat capacity $C_V/R_A$ from $\rho_{t,\mathrm{Gas}}$ to $\rho_{t,\mathrm{Liq}}$ along two isotherms: (**a**) 110 and (**b**) 140 K. The plotted curves correspond to values calculated from Equation (1) with Equations (53)–(57) (red curves) and from the TSW model (black dashed curves).

### 3.4.2. Choice of a Reference State

The functions $U$, $S$, and $F$ are obtained by successive integrations of $C_V$ along isochors; this means a reference temperature is necessary. From Equations (55) and (56), it is obvious that the only state that is identical for all isochors is $T$ infinite.

Due to the fact that, firstly, the development of an equation such as Equation (18) for $\widetilde{c}^{\,\mathrm{r}}_{V,\mathrm{nonreg}}$ becomes more complex with a much slower convergence of the series and, secondly, there are two expressions for this term inside the liquid–vapor coexistence region, it is preferable and easier to numerically integrate these terms. Thus, the respective expressions for $\widetilde{u}^{\,\mathrm{r}}_{\mathrm{nonreg}}$ and $\widetilde{s}^{\,\mathrm{r}}_{\mathrm{nonreg}}$ are now

$$\widetilde{u}^{\,\mathrm{r}}_{\mathrm{nonreg}}(\rho,T) = T^{-1}\int_{\infty}^{T} \widetilde{c}^{\,\mathrm{r}}_{V,\mathrm{nonreg}}(\rho,t)\,dt \tag{58}$$

$$\widetilde{s}^{\,\mathrm{r}}_{\mathrm{nonreg}}(\rho,T) = \int_{\infty}^{T} \widetilde{c}^{\,\mathrm{r}}_{V,\mathrm{nonreg}}(\rho,t)\,t^{-1}\,dt \tag{59}$$

with $\widetilde{c}^{\,\mathrm{r}}_{V,\mathrm{nonreg}}(\rho,\mathrm{T}) = \begin{cases} \widetilde{c}^{\,\mathrm{r}}_{\substack{V,\mathrm{nonreg}\\ \mathrm{outside}}}(\rho,\mathrm{T}) \text{ if } T \geq T_{\mathrm{div}} \\[2ex] \widetilde{c}^{\,\mathrm{r}}_{\substack{V,\mathrm{nonreg}\\ \mathrm{inside}}}(\rho,\mathrm{T}) \text{ otherwise} \end{cases}$ .

To complete the description, the expression for the non-regular compressibility factor is also given, i.e.,

$$Z_{\mathrm{nonreg}} = \rho\left(\frac{\partial \widetilde{a}^{\,\mathrm{r}}_{\mathrm{nonreg}}}{\partial \rho}\right)_{T} = \frac{\rho}{T}\int_{\infty}^{T}\left(1 - \frac{T}{t}\right)\frac{\partial}{\partial \rho}\widetilde{c}^{\,\mathrm{r}}_{V,\mathrm{nonreg}}(\rho,t)\,dt \tag{60}$$

To calculate the partial derivative of $\widetilde{c}^{\,\mathrm{r}}_{V,\mathrm{nonreg}}$, $N_V$ must be considered constant because the experiments imagined here to measure the pressure are performed on a closed system. Thus, the expressions of the partial derivative of $\widetilde{c}^{\,\mathrm{r}}_{V,\mathrm{nonreg}}$ are

$$\frac{\partial}{\partial \rho}\widetilde{c}^{\,\mathrm{r}}_{\substack{V,\mathrm{nonreg}\\ \mathrm{outside}}}(\rho,T) = \frac{3}{2}\frac{\exp\left(-\left(\frac{T_{\mathrm{div}}(\rho)}{T}\right)^{-3/2}\right)}{1-\frac{T_{\mathrm{div}}(\rho)}{T}}\left\{n'_{\mathrm{nonreg}}(\rho)\left(1 - N_V^{-1+\frac{T_{\mathrm{div}}(\rho)}{T}}\right)\right.$$
$$\left. + n_{\mathrm{nonreg}}(\rho)\frac{T_{\mathrm{div}}(\rho)}{T}\left[\frac{1-N_V^{-1+\frac{T_{\mathrm{div}}(\rho)}{T}}}{1-\frac{T_{\mathrm{div}}(\rho)}{T}} - N_V^{-1+\frac{T_{\mathrm{div}}(\rho)}{T}}\ln(N_V) + \frac{3}{2}\left(\frac{T_{\mathrm{div}}(\rho)}{T}\right)^{-5/2}\left(1-N_V^{-1+\frac{T_{\mathrm{div}}(\rho)}{T}}\right)\right]\right\} \tag{61}$$

$$\frac{\partial}{\partial \rho}\widetilde{c}^{\,\mathrm{r}}_{\substack{V,\mathrm{nonreg}\\ \mathrm{inside}}}(\rho,T) = \frac{3}{2}\frac{\exp\left(-\left(\frac{T}{T_{\mathrm{div}}(\rho)}\right)^{-3/2}\right)}{1-\frac{T}{T_{\mathrm{div}}(\rho)}}\left\{n'_{\mathrm{nonreg}}(\rho)\left(1 - N_V^{-1+\frac{T}{T_{\mathrm{div}}(\rho)}}\right)\right.$$
$$\left. - n_{\mathrm{nonreg}}(\rho)\left(\frac{T}{T_{\mathrm{div}}(\rho)}\right)^2\frac{T'_{div}(\rho)}{T}\left[\frac{1-N_V^{-1+\frac{T}{T_{\mathrm{div}}(\rho)}}}{1-\frac{T}{T_{\mathrm{div}}(\rho)}} - N_V^{-1+\frac{T}{T_{\mathrm{div}}(\rho)}}\ln(N_V) + \frac{3}{2}\left(\frac{T}{T_{\mathrm{div}}(\rho)}\right)^{-5/2}\left(1-N_V^{-1+\frac{T}{T_{\mathrm{div}}(\rho)}}\right)\right]\right\} \tag{62}$$

In the same manner as previously, one can deduce primitives for $U$ and $S$ corresponding to the term $\widetilde{c}^{\,\mathrm{r}}_{\substack{\mathrm{crit}\\\mathrm{inside}}}$ as follows:

$$\widetilde{u}^{\,\mathrm{r}}_{\substack{\mathrm{crit}\\\mathrm{inside}}} = \frac{3}{2}n_{\mathrm{crit}}(\rho)\left[\frac{(T/T_{\mathrm{div}}(\rho))^{\varepsilon_{\mathrm{crit}}(\rho)}}{1+\varepsilon_{\mathrm{crit}}(\rho)} + \frac{2\varepsilon_{\mathrm{crit}}(\rho)}{1-\varepsilon_{\mathrm{crit}}(\rho)^2}\right] \tag{63}$$

$$\widetilde{s}^{\,\mathrm{r}}_{\substack{\mathrm{crit}\\\mathrm{inside}}} = \frac{3}{2}n_{\mathrm{crit}}(\rho)\,\varepsilon_{\mathrm{crit}}(\rho)^{-1}\left[\left(\frac{T}{T_{\mathrm{div}}(\rho)}\right)^{\varepsilon_{\mathrm{crit}}(\rho)} - 2\right] \tag{64}$$

Then it is deduced that

$$\widetilde{a}^{\text{r}}_{\substack{\text{crit}\\\text{inside}}}(\rho, T) = \frac{3}{2}n_{\text{crit}}(\rho)\left[\frac{(T/T_{\text{div}}(\rho))^{\varepsilon_{\text{crit}}(\rho)}}{\varepsilon_{\text{crit}}(\rho)\,(1+\varepsilon_{\text{crit}}(\rho))} + \frac{2}{1-\varepsilon_{\text{crit}}(\rho)^2}\right] \tag{65}$$

and

$$\frac{P_{\substack{\text{crit}\\\text{inside}}}(\rho,T)}{\rho R_A T} = \frac{3}{2}\rho\,n'_{\text{crit}}(\rho)\left[\frac{(T/T_{\text{div}}(\rho))^{\varepsilon_{\text{crit}}(\rho)}}{\varepsilon_{\text{crit}}(\rho)\,(1+\varepsilon_{\text{crit}}(\rho))} + \frac{2}{1-\varepsilon_{\text{crit}}(\rho)^2}\right]$$
$$+ \frac{3}{2}\rho\,n_{\text{crit}}(\rho)\left\{-\frac{1+2\varepsilon_{\text{crit}}(\rho)}{\varepsilon_{\text{crit}}(\rho)^2(1+\varepsilon_{\text{crit}}(\rho))^2}\varepsilon'_{\text{crit}}(\rho)\left(\frac{T}{T_{\text{div}}(\rho)}\right)^{\varepsilon_{\text{crit}}(\rho)} + \frac{4\varepsilon_{\text{crit}}(\rho)\,\varepsilon'_{\text{crit}}(\rho)}{\left(1-\varepsilon_{\text{crit}}(\rho)^2\right)^2}\right.$$
$$\left.+\frac{(T/T_{\text{div}}(\rho))^{\varepsilon_{\text{crit}}(\rho)}}{\varepsilon_{\text{crit}}(\rho)\,(1+\varepsilon_{\text{crit}}(\rho))}\left[\varepsilon'_{\text{crit}}(\rho)\ln\left(\frac{T}{T_{\text{div}}(\rho)}\right) - \varepsilon_{\text{crit}}(\rho)\frac{T'_{\text{div}}(\rho)}{T_{\text{div}}(\rho)}\right]\right\} \tag{66}$$

We emphasize here that the choice of these expressions to describe the coexistence region has no effect on the properties of the pure fluid up to the saturation curve.

The present modeling with these new expressions of regular and non-critical terms is referred to as the "***non-extensive formulation***". The non-extensive residual part of Helmholtz energy $\widetilde{a}^{\text{r}}(\rho, T)$ and its partial derivatives with temperature are given in Table 7. The first partial derivative with density can be easily deduced from the expression of the compressibility factor Z. Then, the second partial derivatives $\frac{\partial^2 \widetilde{a}^{\text{r}}}{\partial\rho\partial T}$ and $\left.\frac{\partial^2 \widetilde{a}^{\text{r}}}{\partial\rho^2}\right|_T$ can be obtained from the first derivatives of the compressibility factor, which are given in Tables 8 and 9.

**Table 7.** Mathematical expressions of the dimensionless terms for the non-extensive residual part of the Helmholtz function and its partial derivatives with temperature.

$\widetilde{a}^{\text{r}}_{\text{reg}} = \frac{3}{2}n_{\text{reg}}\frac{T_c}{T}\left\{\frac{1}{m}\left[\left(\frac{T}{T_c}\right)^m - 1\right] + \frac{\lambda^{-m}}{2-m}\Gamma\left(\frac{m}{2-m},\left(\frac{\lambda T}{T_c}\right)^{2-m}\right)\right\}$

$\qquad -\frac{3}{2}n_{\text{reg}}\left\{\frac{1}{m-1}\left[\left(\frac{T}{T_c}\right)^{m-1} - 1\right] + \frac{\lambda^{1-m}}{2-m}\Gamma\left(\frac{m-1}{2-m},\left(\frac{\lambda T}{T_c}\right)^{2-m}\right)\right\}$

$T_c\left(\frac{\partial\widetilde{a}^{\text{r}}_{\text{reg}}}{\partial T}\right)_\rho = \frac{3}{2}n_{\text{reg}}\left(\frac{T_c}{T}\right)^2 m^{-1}\left\{1 - \left(\frac{T}{T_c}\right)^m + \frac{\lambda^{-m}m}{m-2}\Gamma\left(\frac{m}{2-m},\left(\frac{\lambda T}{T_c}\right)^{2-m}\right)\right\}$

$T_c^2\left(\frac{\partial^2\widetilde{a}^{\text{r}}_{\text{reg}}}{\partial T^2}\right)_\rho = \frac{3}{2}n_{\text{reg}}\left(\frac{T_c}{T}\right)^3\left\{\left(\frac{T}{T_c}\right)^m\left(\frac{2-m}{m} + \exp\left(-\left(\frac{\lambda T}{T_c}\right)^{2-m}\right)\right) - \frac{2}{m} + \frac{2\lambda^{-m}}{m-2}\Gamma\left(\frac{m}{2-m},\left(\frac{\lambda T}{T_c}\right)^{2-m}\right)\right\}$

$\widetilde{a}^{\text{r}}_{\text{nonreg}} = \int_\infty^T \widetilde{c}^{\text{r}}_{V,\text{nonreg}}\left(1 - \frac{T}{t}\right)\frac{dt}{T}$

$T_c\left(\frac{\partial\widetilde{a}^{\text{r}}_{\text{nonreg}}}{\partial T}\right)_\rho = -\frac{T_c}{T^2}\int_\infty^T \widetilde{c}^{\text{r}}_{V,\text{nonreg}}\,dt,\quad T_c^2\left(\frac{\partial^2\widetilde{a}^{\text{r}}_{\text{nonreg}}}{\partial T^2}\right)_\rho = -\left(\frac{T_c}{T}\right)^2\widetilde{c}^{\text{r}}_{V,\text{nonreg}} + 2\frac{T_c^2}{T^3}\int_\infty^T \widetilde{c}^{\text{r}}_{V,\text{nonreg}}\,dt$

$\widetilde{a}^{\text{r}}_{\text{crit}} = \frac{3}{2}n_{\text{crit}}\begin{cases}\frac{(T_{\text{div}}/T)^{\varepsilon_{\text{crit}}}}{(1-\varepsilon_{\text{crit}})\,\varepsilon_{\text{crit}}} & \text{if } T \geq T_{\text{div}} \\ \frac{(T/T_{\text{div}})^{\varepsilon_{\text{crit}}}}{(1+\varepsilon_{\text{crit}})\,\varepsilon_{\text{crit}}} + \frac{2}{1-\varepsilon_{\text{crit}}^2} & \text{otherwise}\end{cases}$

$T_c\left(\frac{\partial\widetilde{a}^{\text{r}}_{\text{crit}}}{\partial T}\right)_\rho = \frac{3}{2}n_{\text{crit}}\frac{T_c}{T}\begin{cases}\frac{1}{\varepsilon_{\text{crit}}-1}\left(\frac{T_{\text{div}}}{T}\right)^{\varepsilon_{\text{crit}}} & \text{if } T \geq T_{\text{div}} \\ \frac{1}{1+\varepsilon_{\text{crit}}}\left(\frac{T}{T_{\text{div}}}\right)^{\varepsilon_{\text{crit}}} & \text{otherwise}\end{cases}$

$T_c^2\left(\frac{\partial^2\widetilde{a}^{\text{r}}_{\text{crit}}}{\partial T^2}\right)_\rho = \frac{3}{2}n_{\text{crit}}\left(\frac{T_c}{T}\right)^2\begin{cases}\frac{1+\varepsilon_{\text{crit}}}{\varepsilon_{\text{crit}}-1}\left(\frac{T_{\text{div}}}{T}\right)^{\varepsilon_{\text{crit}}} & \text{if } T \geq T_{\text{div}} \\ \frac{\varepsilon_{\text{crit}}-1}{1+\varepsilon_{\text{crit}}}\left(\frac{T}{T_{\text{div}}}\right)^{\varepsilon_{\text{crit}}} & \text{otherwise}\end{cases}$

$\widetilde{a}^{\text{r}}_0 = \frac{3}{2}\frac{T_c}{T}\hat{u}_0(\rho) - \{\widetilde{s}_0(\rho) + \ln(\rho)\}$

$T_c\left(\frac{\partial\widetilde{a}^{\text{r}}_0}{\partial T}\right)_\rho = -\frac{3}{2}\left(\frac{T_c}{T}\right)^2\hat{u}_0(\rho)$

$T_c^2\left(\frac{\partial^2\widetilde{a}^{\text{r}}_0}{\partial T^2}\right)_\rho = 3\left(\frac{T_c}{T}\right)^3\hat{u}_0(\rho)$

**Table 8.** Mathematical expressions of the first partial derivatives with temperature of the compressibility factor Z. All these derivatives are in $K^{-1}$.

$\left(\frac{\partial Z_{\text{reg}}}{\partial T}\right)_\rho$ is given in Appendix A

$\left(\frac{\partial Z_{\text{nonreg}}}{\partial T}\right)_\rho = -\frac{\rho}{T^2}\int_\infty^T \left.\frac{\partial \tilde{c}_{V,\text{nonreg}}^\tau}{\partial\rho}\right|_T dt$

$\left(\frac{\partial Z_{\text{crit}}}{\partial T}\right)_\rho = \frac{3}{2}\frac{\rho}{T}n'_{\text{crit}}\begin{cases}\frac{1}{\varepsilon_{\text{crit}}-1}\left(\frac{T_{\text{div}}}{T}\right)^{\varepsilon_{\text{crit}}} & \text{if } T \geq T_{\text{div}}\\ \frac{1}{1+\varepsilon_{\text{crit}}}\left(\frac{T}{T_{\text{div}}}\right)^{\varepsilon_{\text{crit}}} & \text{otherwise}\end{cases}$

$+\frac{3}{2}\rho\, n_{\text{crit}}\begin{cases}\frac{(T_{\text{div}}/T)^{\varepsilon_{\text{crit}}-1}}{T^2(\varepsilon_{\text{crit}}-1)^2}\left\{\varepsilon_{\text{crit}}(\varepsilon_{\text{crit}}-1)T'_{\text{div}}+T_{\text{div}}\varepsilon'_{\text{crit}}\left((\varepsilon_{\text{crit}}-1)\ln\left(\frac{T_{\text{div}}}{T}\right)\right)-1\right\} & \text{if } T \geq T_{\text{div}}\\ \frac{(T/T_{\text{div}})^{\varepsilon_{\text{crit}}-1}}{T_{\text{div}}^2(1+\varepsilon_{\text{crit}})^2}\left\{\varepsilon_{\text{crit}}(1+\varepsilon_{\text{crit}})T'_{\text{div}}-T_{\text{div}}\varepsilon'_{\text{crit}}\left((1+\varepsilon_{\text{crit}})\ln\left(\frac{T}{T_{\text{div}}}\right)\right)-1\right\} & \text{otherwise}\end{cases}$

$\left(\frac{\partial Z_0}{\partial T}\right)_\rho = -\frac{3}{2}\frac{\rho}{T}\frac{T_c}{T}\hat{u}'_0(\rho)$

**Table 9.** Mathematical expressions of the first partial derivatives with density of the compressibility factor Z for $T \geq T_{\text{div}}$. All these derivatives are in $cm^3/g$.

$\left(\frac{\partial Z_{\text{reg}}}{\partial\rho}\right)_T$ is given in Appendix A

$\left(\frac{\partial Z_{\text{nonreg}}}{\partial\rho}\right)_T = \frac{1}{T}\int_\infty^T\left(1-\frac{T}{t}\right)\left.\frac{\partial\tilde{c}_{V,\text{nonreg}}}{\partial\rho}\right|_T dt + \frac{\rho}{T}\int_\infty^T\left(1-\frac{T}{t}\right)\left.\frac{\partial^2\tilde{c}_{V,\text{nonreg}}}{\partial\rho^2}\right|_T dt$

with

$\left.\frac{\partial^2\tilde{c}_{V,\text{nonreg}}^\tau}{\partial\rho^2}\right|_T = -\frac{3}{2}\frac{T'_{\text{div}}}{(T-T_{\text{div}})}\left[n'_{\text{nonreg}}+\frac{T'_{\text{div}}}{(T-T_{\text{div}})}n_{\text{nonreg}}+\frac{1}{2}\frac{T''_{\text{div}}}{T'_{\text{div}}}n_{\text{nonreg}}\right]\exp\left(-\left(\frac{T}{T_{\text{div}}}\right)^{3/2}\right)$

$\times\left\{3\left(\frac{T}{T_{\text{div}}}\right)^{5/2}\left(-1+N_V^{-1+\frac{T_{\text{div}}}{T}}\right)+2N_V^{-1+\frac{T_{\text{div}}}{T}}\ln(N_V)\right\}$

$-\frac{3}{8}\frac{n_{\text{nonreg}}\,T'^2_{\text{div}}}{T_{\text{div}}(T-T_{\text{div}})}\exp\left(-\left(\frac{T}{T_{\text{div}}}\right)^{3/2}\right)\left(\frac{T}{T_{\text{div}}}\right)^{5/2}\left\{3\left(3\left(\frac{T}{T_{\text{div}}}\right)^{3/2}-5\right)\left(-1+N_V^{-1+\frac{T_{\text{div}}}{T}}\right)\right.$

$\left.+4N_V^{-1+\frac{T_{\text{div}}}{T}}\ln(N_V)\frac{T_{\text{div}}}{T}\left(3+\ln(N_V)\left(\frac{T_{\text{div}}}{T}\right)^{5/2}\right)\right\}$

$-\frac{3}{2}\frac{T}{(T-T_{\text{div}})^3}\exp\left(-\left(\frac{T}{T_{\text{div}}}\right)^{3/2}\right)\left(-1+N_V^{-1+\frac{T_{\text{div}}}{T}}\right)\left\{2(T-T_{\text{div}})\,T'_{\text{div}}\,n'_{\text{nonreg}}+(T-T_{\text{div}})^2 n''_{\text{nonreg}}\right.$

$\left.+\left(2T'^2_{div}+(T-T_{\text{div}})\,T''_{\text{div}}\right)n_{\text{nonreg}}\right\}$

$\left(\frac{\partial Z_{\text{crit}}}{\partial\rho}\right)_T = \frac{3}{2}\left(n'_{\text{crit}}+\rho\,n''_{\text{crit}}\right)\frac{(T_{\text{div}}/T)^{\varepsilon_{\text{crit}}}}{(1-\varepsilon_{\text{crit}})\varepsilon_{\text{crit}}}+\frac{3}{2}\left(n_{\text{crit}}+2\rho\,n'_{\text{crit}}\right)\frac{(T_{\text{div}}/T)^{\varepsilon_{\text{crit}}}}{(\varepsilon_{\text{crit}}-1)}\left\{\frac{2\varepsilon_{\text{crit}}-1}{(\varepsilon_{\text{crit}}-1)\varepsilon_{\text{crit}}}\frac{\varepsilon'_{\text{crit}}}{\varepsilon_{\text{crit}}}-\frac{T'_{\text{div}}}{T_{\text{div}}}-\ln\left(\frac{T_{\text{div}}}{T}\right)\frac{\varepsilon'_{\text{crit}}}{\varepsilon_{\text{crit}}}\right\}$

$-\frac{3}{2}\rho\,n_{\text{crit}}\frac{(T_{\text{div}}/T)^{\varepsilon_{\text{crit}}}}{(\varepsilon_{\text{crit}}-1)^2}\left\{(\varepsilon_{\text{crit}}-1)^2\left(\frac{T'_{\text{div}}}{T_{\text{div}}}\right)^2+\frac{2\varepsilon'^2_{\text{crit}}}{(\varepsilon_{\text{crit}}-1)\varepsilon^3_{\text{crit}}}+\frac{T'_{\text{div}}}{T_{\text{div}}}\left[(\varepsilon_{\text{crit}}-1)\frac{T''_{\text{div}}}{T'_{\text{div}}}+2\varepsilon'_{\text{crit}}\left(-1+\ln\left(\left(\frac{T_{\text{div}}}{T}\right)^{\varepsilon_{\text{crit}}-1}\right)\right)\right]\right.$

$\left.+\frac{1}{\varepsilon^2_{\text{crit}}}\left[\varepsilon''_{\text{crit}}\left(1-\varepsilon_{\text{crit}}\left(2+\ln\left(\frac{T_{\text{div}}}{T}\right)\right)+\varepsilon^2_{\text{crit}}\ln\left(\frac{T_{\text{div}}}{T}\right)\right)+6\,\varepsilon'^2_{\text{crit}}+\varepsilon'^2_{\text{crit}}\ln\left(\frac{T_{\text{div}}}{T}\right)\left(2-\varepsilon_{\text{crit}}\left(4+\ln\left(\frac{T_{\text{div}}}{T}\right)\right)+\varepsilon^2_{\text{crit}}\ln\left(\frac{T_{\text{div}}}{T}\right)\right)\right]\right\}$

$\left(\frac{\partial Z_0}{\partial\rho}\right)_T = \frac{3}{2}\frac{T_c}{T}\left[\hat{u}'_0(\rho)+\rho\,\hat{u}''_0(\rho)\right]-\left[\tilde{s}'_0(\rho)+\rho\,\tilde{s}''_0(\rho)\right]$

Table 26 of [4] summarizes how to calculate the thermodynamic properties from the empirical description of Helmholtz free energy and its derivatives. In the present approach, the same thermodynamic properties are deduced from the isochoric heat capacity equation and the thermal equation of state, which are now two experimentally measured quantities. *A Mathematica application with the new equations of state corresponding to the non-extensive formulation can be found in the Supplementary Materials.*

## 4. Comparison of the New Equation of State with Experimental Data and the TSW Model

In this section, the quality of the new equation of state in its **non-extensive formulation** (see Tables 3 and 7) is analyzed in comparison with selected experimental and theoretical data. Most figures also show a comparison with the values calculated using the so-called reference equation of state established by Tegeler et al. [4], which has been called the TSW model here.

### 4.1. Melting-Phase Transition

In the TSW model, their Equation (2.7) gives the melting pressure variation (see [4]). However, they arbitrarily discard some data sets, for example, the data of Zha et al. [12]. It is clear that these data are scattered, but as they are obtained at high temperatures and pressures, it should be interesting to use them. New and more accurate data from Datchi et al. [13] are almost in the same temperature range as those by Zha et al. [12] and are consistent with these data. It is possible to have a complete view of the melting line for the range of temperature corresponding to Ronchi's data set by adding the data of Jephcoat et al. [14].

Thus, using a two-parameter Simon–Glatzel-type function, it is possible to represent in a coherent manner and with continuity the data of Hardy et al. [15], Zha et al. [12], Datchi et al. [13] and Jephcoat et al. [14]:

$$P_m - P_t = a_1 \left[ \left( \frac{T}{T_t} \right)^{a_2} - 1 \right] \tag{67}$$

with $a_1$ = 225.2858 MPa and $a_2$ = 1.5284. This equation is used in the following to represent the melting line.

It must be noticed that for determining the parameters in Equation (67), the data from Bridgman [16], Lahr et al. [17], Crawford et al. [18] and L'Air Liquide [5] have also been used.

Figure 9 compares some different data sets with values calculated from Equation (67) (solid line), Equation (2) written by Datchi et al. [13] (dashed curve), and Equation (1) written by Abramson [19] (dot dashed curve). Equation (67) is very close to the function written by Datchi et al. [13], and both equations are consistent with the data of Jephcoat et al. [14]. The main difference between Equation (67) and Equation (2) from Datchi et al. [13] is at low temperature, where this last one is very inaccurate and cannot be used when approaching the triple point. Equation (1) from Abramson is determined for the representation of its own data, and it can be seen that the extrapolation of this function is not consistent with the data of Jephcoat et al. [14]. Also, Equation (1) of Abramson [19] is not very accurate at low temperatures.

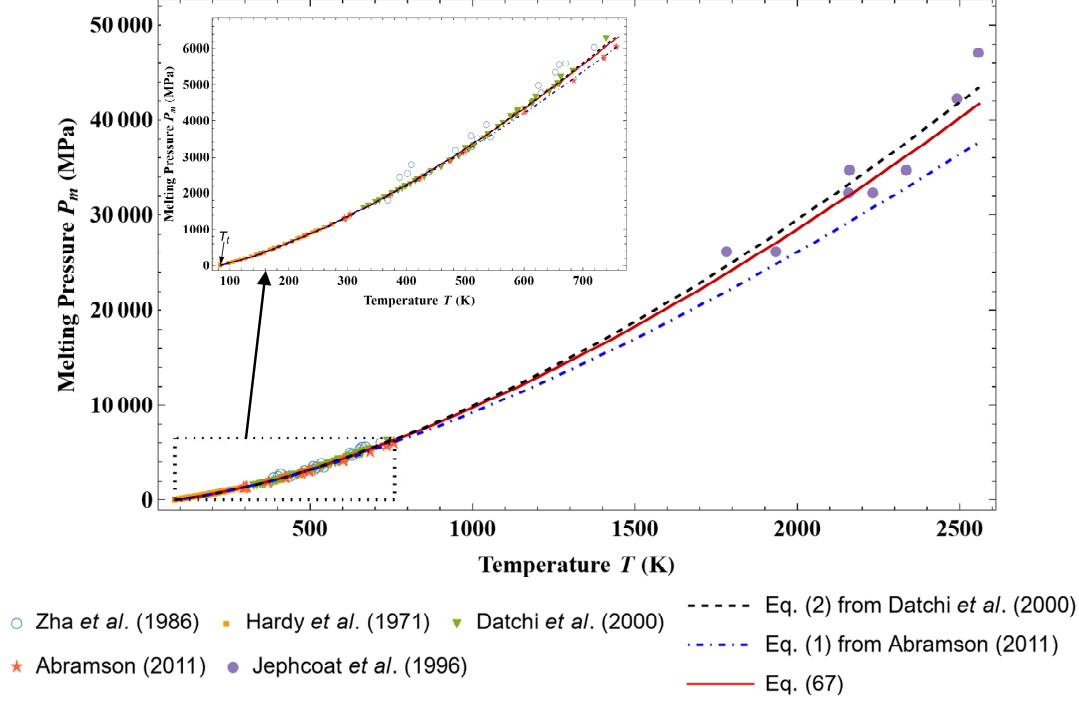

**Figure 9.** Plot of the melting pressure data determined by Zha et al. [12], Hardy et al. [15], Datchi et al. [13], Abramson [19], and Jephcoat et al. [14]. The plotted curves correspond to values calculated from Equation (67) (solid line), Equation (2) given by Datchi et al. (dashed curve), and Equation (1) given by Abramson (dot-dashed curve).

Some of the previous authors have also measured the liquid density on the melting line. But, as can be seen in Figure 10, all the data sets have a large dispersion, which makes their representation difficult. In particular, the data of Lahr et al. [17] at low temperatures seem incompatible with the other data sets, and at high temperatures, these data are incompatible with the data of Crawford et al. [18]. For these reasons, two equations are proposed here that give greater importance at high temperatures: the data from Lahr et al. [17] and the data from Crawford et al. [18]:

$$T_{\mathrm{m,Low}}(\rho) = T_\mathrm{t} + 1015.4 \times \left(\frac{\rho}{\rho_{\mathrm{t,Liq}}} - 1\right)^{1.843} + 250.46 \times \left(\frac{\rho}{\rho_{\mathrm{t,Liq}}} - 1\right) \qquad \text{(67Low)}$$

$$T_{\mathrm{m,High}}(\rho) = T_\mathrm{t} + 677.25 \times \left(\frac{\rho}{\rho_{\mathrm{t,Liq}}} - 1\right)^{1.236} + 94.955 \times \left[1 - \exp\left(-\left(\frac{\rho - \rho_{\mathrm{t,Liq}}}{0.25}\right)^{10.442}\right)\right] \qquad \text{(67High)}$$

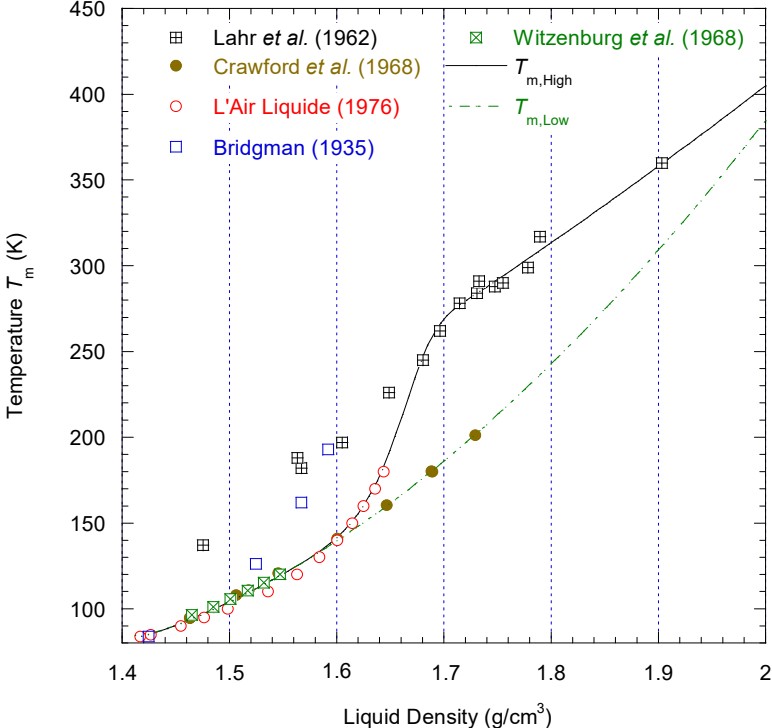

**Figure 10.** Plot of the liquid density data on the melting line determined by Bridgman [16], Lahr et al. [17], Crawford et al. [18], Witzenburg et al. [20], and L'Air Liquide [5]. The plotted curves correspond to values calculated from the functions $T_{\mathrm{m,Low}}(\rho)$ (i.e., Equation (67Low), dashed line) and $T_{\mathrm{m,High}}(\rho)$ (i.e., Equation (67High), solid line).

These two empirical functions are represented in Figure 10. As can be seen, for a liquid density smaller than 1.6 g/cm$^3$ (i.e., $\rho < \rho_{\mathrm{t,Sol}}$), the two functions are almost identical.

### 4.2. Single-Phase Region

#### 4.2.1. Isochoric Heat Capacities

The present modeling is mainly based on $C_V$ data provided by NIST, but because the data from NIST are identical, with some exceptions, to the numerical values deduced from Equation (4.1) of the TSW model [4], a comparison between the results obtained with the two models is necessary. In the pressure–temperature region covered by NIST, the relative differences in $\Delta C_V$ observed between the TSW model and the present one are less than the uncertainties given in Figure 44 of [4]. The most important relative difference is obtained in the vicinity of the critical point, as shown in Figure 11. It can also be noticed in Figure 11 that

outside the critical region, the error oscillates almost regularly with density (for all temperatures). These oscillations come from the different mathematical forms used in the two models. For the present modeling, a perfectly smooth monotonic function has been used for $C_V$, while the TSW model uses a polynomial equation. This polynomial equation induced small oscillations on $C_V$ and these oscillations can be seen on $\Delta C_V$ (see Figure 11). Such oscillations, more or less amplified, should also appear in other relative differences between thermodynamic quantities calculated from the two models.

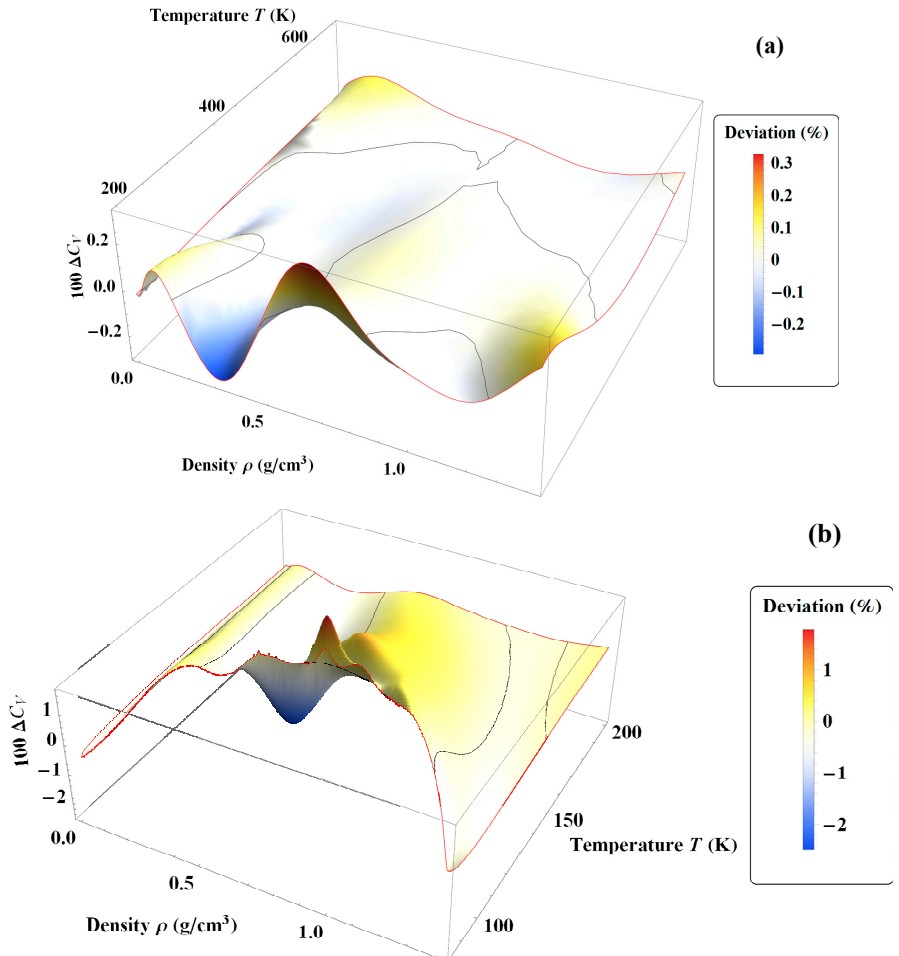

**Figure 11.** Percentage deviations of isochoric heat capacity $\Delta C_V = (C_{V\,\text{TSW model}} - C_{V\,\text{Eq. (1)}})/C_{V\,\text{TSW model}}$ between the TSW model and Equation (1) in the density range between $\rho_{t,\text{Gas}}$ and $\rho_{t,\text{Liq}}$: (**a**) temperature range between 200 and 700 K; (**b**) temperature range between $T_t$ and 200 K. The black lines correspond to values of $\Delta C_V$ equal to zero.

In the paper of Ronchi, there are no $C_V$ data, so no direct comparison is possible with the present modeling. However, in the region covered by Ronchi, Vrabec et al. [21] have calculated $C_V$ data using a molecular dynamics calculation based on a (12, 6) Lennard-Jones potential. Figure 12 shows plots of the isochoric heat capacity on three high density isochors. The isochors with $\rho$ = 1.196 g/cm$^3$ and $\rho$ = 1.393 g/cm$^3$ are smaller than the density $\rho_{t,\text{Liq}}$ of the liquid at the triple point and hence are limited by the saturated liquid line at low temperatures. The isochor $\rho$ = 1.6 g/cm$^3$ is limited by the solidification line. As can be seen, in the pressure–temperature region covered by NIST data, the difference between the present approach and the TSW model is insignificant. The difference only becomes significant for temperatures larger than 1000 K and for densities higher than $\rho_{t,\text{Liq}}$. For these conditions, the data of Vrabec et al. [21] are better fitted with the present modeling than with the TSW model.

This result was expected because the present model has been built to reproduce the data of Ronchi, which are also based on a statistical model using a potential of type (12, 7).

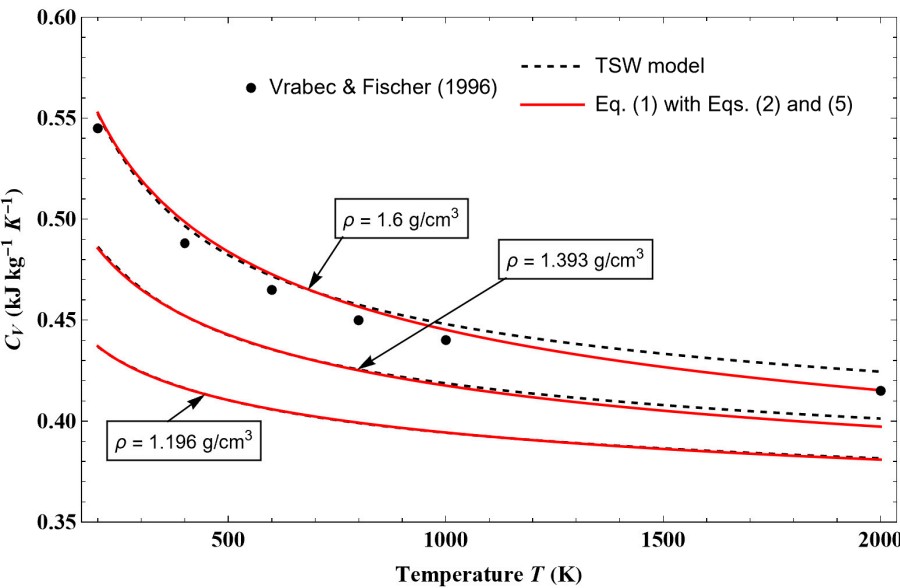

**Figure 12.** Variations with temperature of the isochoric heat capacity along three isochors for high liquid densities (i.e., $\rho \sim \rho_{t,Liq}$). The black points correspond to the data of Vrabec et al. [21]. The plotted curves correspond to the values calculated from Equation (1) (red curves) and the TSW model (black dashed curves).

In the region covered by the data of Ronchi and not covered by data from NIST, there exists the data of L'Air Liquide (685 data points [5]), which were not considered in the TSW model. Figure 13 shows plots of L'Air Liquide data [5] on their highest isotherm at 1100 K and the corresponding calculated curves from the present work and the TSW model. The maximum relative error is around 3%, and once again, these data are slightly better fitted with the present model than with the TSW model.

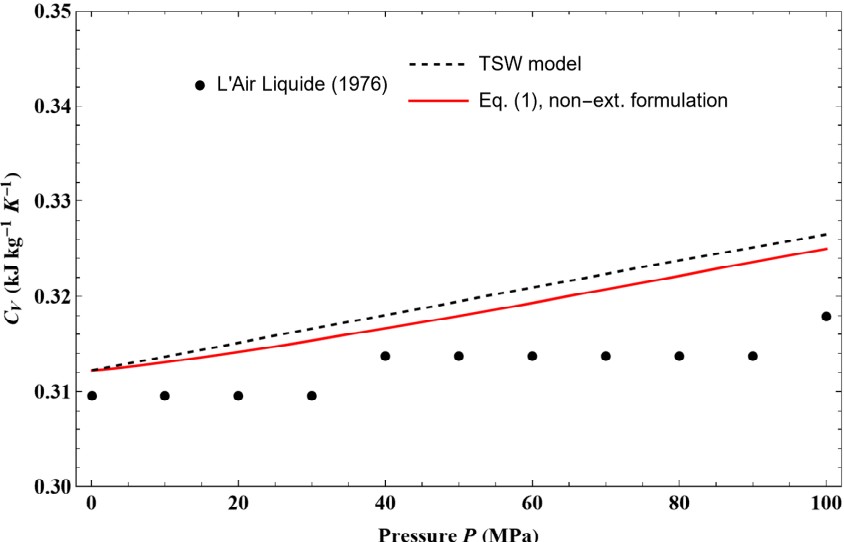

**Figure 13.** Variation with pressure of the isochoric heat capacity along the isotherm at 1100 K. The data points are from L'Air Liquide [5], and the plotted curves correspond to the values calculated from Equation (1) (red curve) and the TSW model (black dashed curve).

Even if the calculated values of Ronchi are not accurate enough, the isochors of $C_V$ show the right variation with a maximum when the temperature tends to zero, as expected for all liquids (Figure 14). These maxima of $C_V$ are observed in water, and they should also exist in argon. The maximum is also well understood as an extension in the single phase of the same very sharp maximum, which is observed in the vapor–liquid coexistence region. On the contrary, with the TSW model, $C_V$ tends to infinity when the temperature tends to zero (see Figure 14), which is an improper variation.

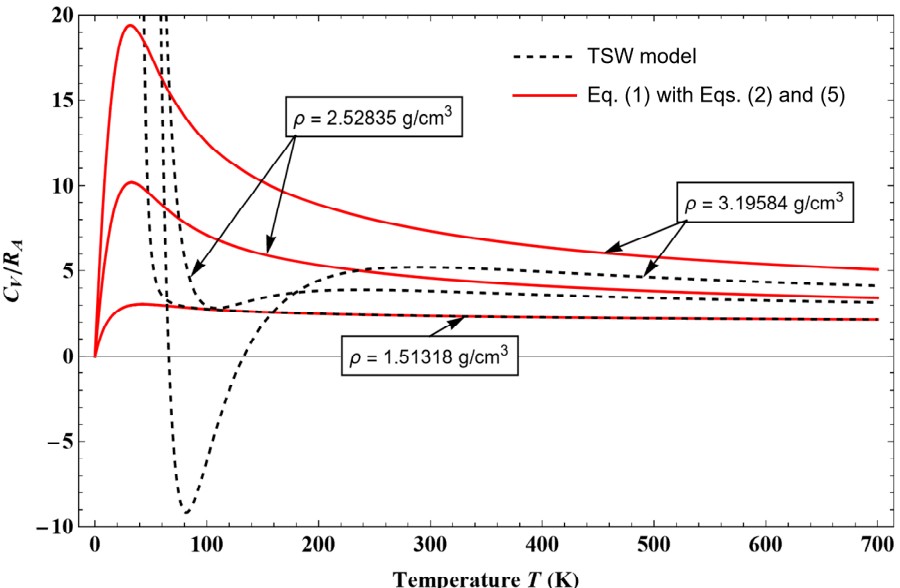

**Figure 14.** Variations with temperature of the dimensionless isochoric heat capacity along three isochors for very high liquid densities (i.e., $\rho > \rho_{t,Liq}$). The plotted curves correspond to the values calculated from Equation (1) (red curves) and the TSW model (black dashed curves).

### 4.2.2. Thermal Properties

As explained in Section 3.1, the $P\rho T$ data from NIST and Ronchi (420 data points, [7]) were used to determine the arbitrary functions $U_0(\rho)$ and $S_0(\rho)$ of the internal energy and entropy, respectively. Therefore, the equation of state $P(\rho, T)$ with the present approach depends on the accuracy obtained from the modeling of $U_0(\rho)$ and $S_0(\rho)$. Although the slopes of the straight lines $P - P_{C_V}$ are more accurately determined than the ordinates at the origin of these curves, Figures 15 and 16 show that the average absolute errors obtained for $U(\rho, T)$ and $S(\rho, T)$, respectively, on all the isochors located in the region of pressure and temperature covered by NIST are very small and of the same order of magnitude for a given temperature. Figures 15 and 16 show that, in both cases, the oscillation of the average error value is nearly centered on zero. Error bars represent the standard deviation of the absolute error, and given the value of these standard deviations from the average, it can be understood that the deviation on each isochor is nearly the same for all values of temperature. This indicates that the shape of the isochors as a function of temperature is very well reproduced, and the errors are due to small oscillations in the data arising from the mathematical form used in the TSW model. However, from the mathematical expressions we used, it is not possible to compensate for such oscillations.

Thus, for $T > T_c$ and for all densities in the range of $\rho_{t,Gas}$ to $\rho_{t,Liq}$, it is found that the relative error on pressure between the NIST data (or TSW model) and the data calculated by the present model shows a "beautiful" oscillation in density (i.e., along isotherms) between $-0.2\%$ and $+0.4\%$. In the gas phase, the relative error remains well below 0.2%, which is only reached on the coexistence curve and in the vicinity of $\rho = 0.3$ g/cm$^3$. In the liquid phase, the relative error remains well below 0.5%, except close to the coexistence line. These largest errors are due to the fact that, in dense phases, small variations in density can lead to large variations

in pressure. We will come back to this question in Section 4.3.2, but it can be noticed that, in order to clearly analyze this problem, it is necessary to look at the inverse equation $\rho(P, T)$ obtained by inversion of Equation (45).

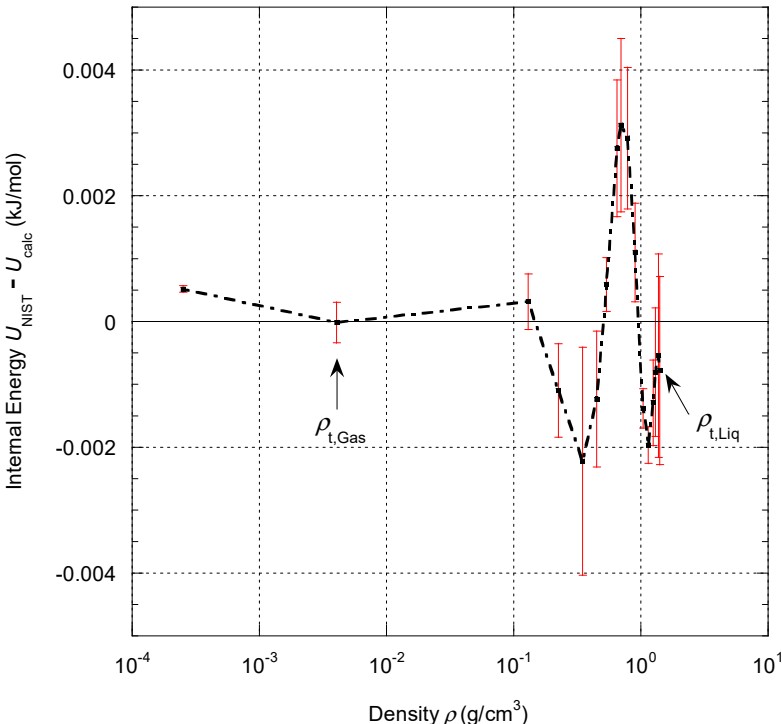

**Figure 15.** Deviations of NIST internal energy data [6] from Equation (39) along isochors up to 700 K. The error bars correspond to standard deviations. The dashed lines are eye guides.

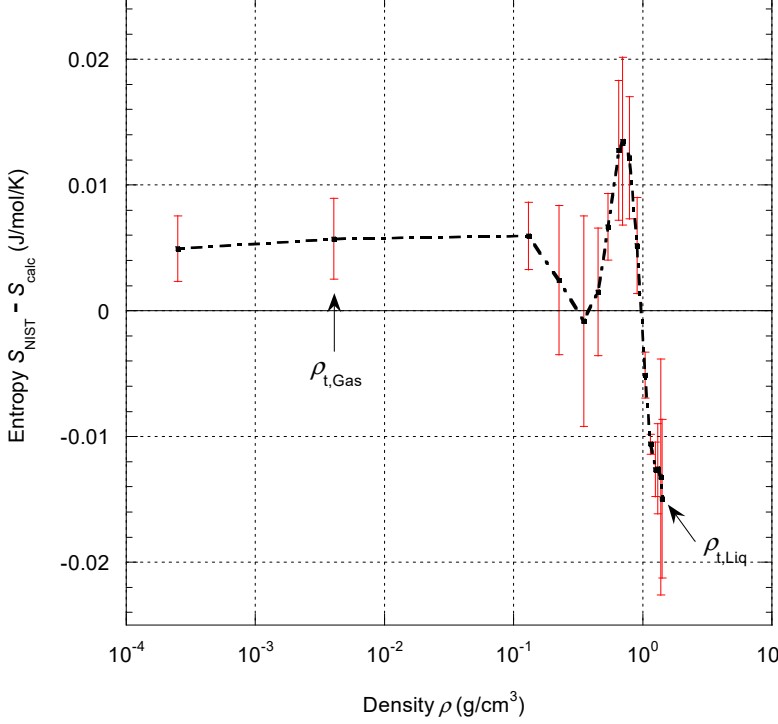

**Figure 16.** Deviations of NIST entropy data [6] from Equation (39) along isochors up to 700 K. The error bars correspond to standard deviations. The dashed lines are eye guides.

Due to the strong nonlinearity of the equation $P(\rho, T)$, irrespective of the model (TSW, Ronchi, or the present one), it is not possible to obtain an analytical form of the inverse equation $\rho(P, T)$, so a numerical method needs to be used. The calculated data $\rho(P, T)$ from the different models were compared. Figure 17 shows the relative error in density between the data calculated by the present model and the NIST data. The same tolerance range (from $\pm 0.03\%$ to $\pm 0.5\%$ in density) as proposed in Figure 42 of [4] was used. It is evident that Figure 17 and Figure 42 of [4] are comparable, though the distribution of tolerance regions is different.

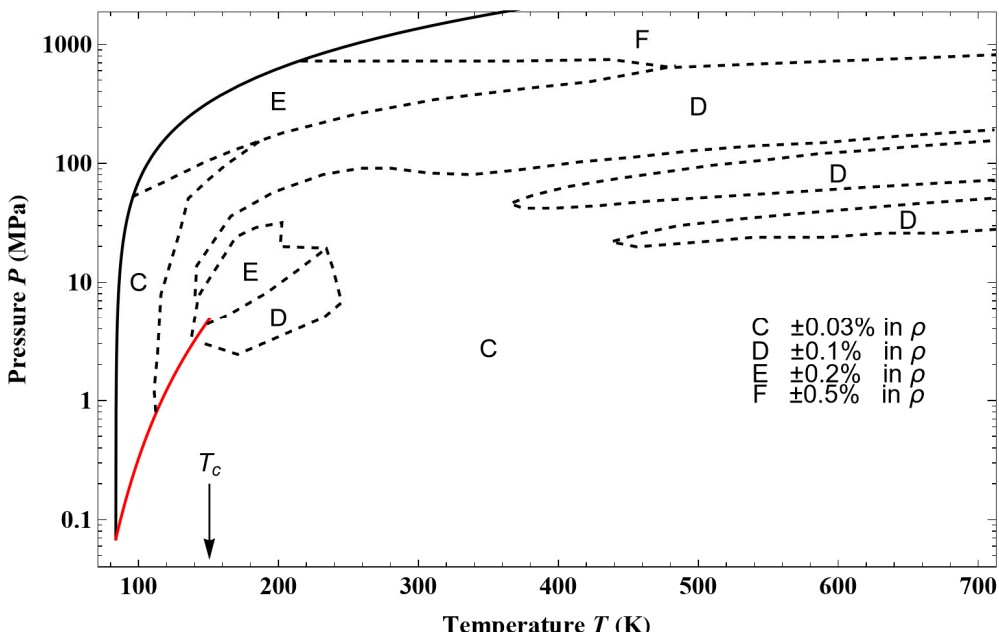

**Figure 17.** Tolerance diagram for densities calculated from inversion of Equation (45). The red curve corresponds to the saturated vapor pressure curve and the black one to the melting line.

For $P > P_c$ and for all temperatures corresponding to NIST data, it can be seen that the relative error in density and their oscillations (Figure 17) of the present modeling are lower or close to the error obtained from the TSW model (see Figure 42 of [4]), except in the vicinity of the critical point. It can, however, be noticed that in this region, the TSW model shows an uncertainty on pressure; such uncertainty is obviously smaller than the uncertainty in density.

For $P < P_c$ and for all the gaseous phase, the relative error in density (in the range $\pm 0.03$ to $\pm 0.1\%$) is close to the one given by the TSW model; globally, the error of the present model is in the range of $\pm 0.03\%$, i.e., category C in [4].

Before discussing these different tolerance diagrams, we will first look at the comparison of the calculated density data (from the TSW model and the present one) with the data from L'Air Liquide (729 data points, [5]). The accuracy claimed by L'Air Liquide on density measurements ranges between $\pm 0.1\%$ and $\pm 1.5\%$, depending on the experimental method used.

Figures 18 and 19 display the relative error in density as a function of temperature between the calculated data and L'Air Liquide data [5] along two isobars. The data along the isobar at 0.1 MPa are all in the gaseous phase, while the data along the isobar at 100 MPa range from the liquid phase to the supercritical one. Both relative errors show comparable variations with temperature. Figure 18 shows that using the TSW model, the relative errors are in agreement with the uncertainty obtained with the present model. The relative errors of the present modeling are slightly larger at low temperatures, but the error variation in all the temperature ranges is better centered on zero. This means that the shape of the isobars is better reproduced by the present modeling. In Figure 19, it can be noticed that the relative errors using the TSW model agree again with the uncertainty obtained with the present model.

The relative errors from the present model are slightly larger almost everywhere but remain in the tolerance range given by Tegeler et al.

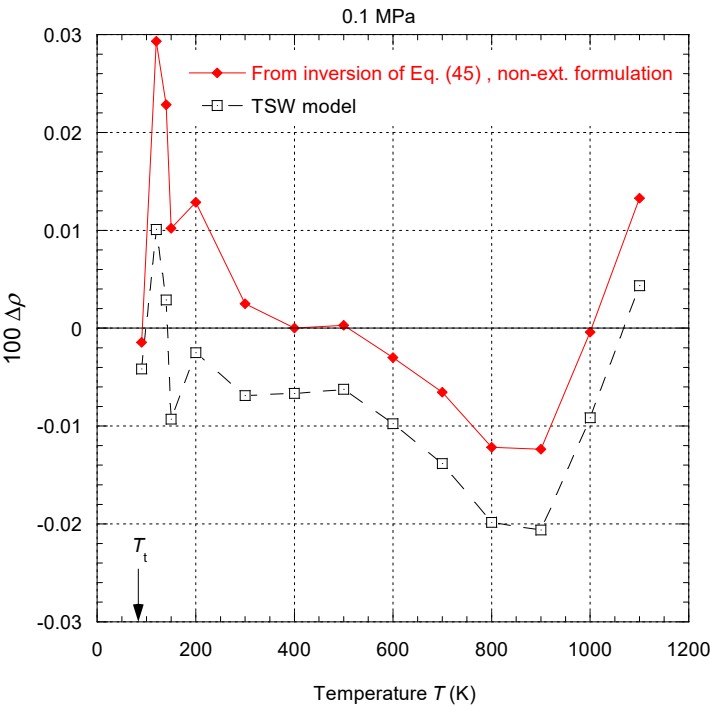

**Figure 18.** Percentage deviations of density $\Delta\rho = \left(\rho_{\text{LAir Liquide}} - \rho_{\text{calc}}\right)/\rho_{\text{LAir Liquide}}$ on the isobar at 0.1 MPa between the data of L'Air Liquide [5] and the inversion of Equation (45) (red diamonds) or the TSW model (black open squares). The lines are eye guides.

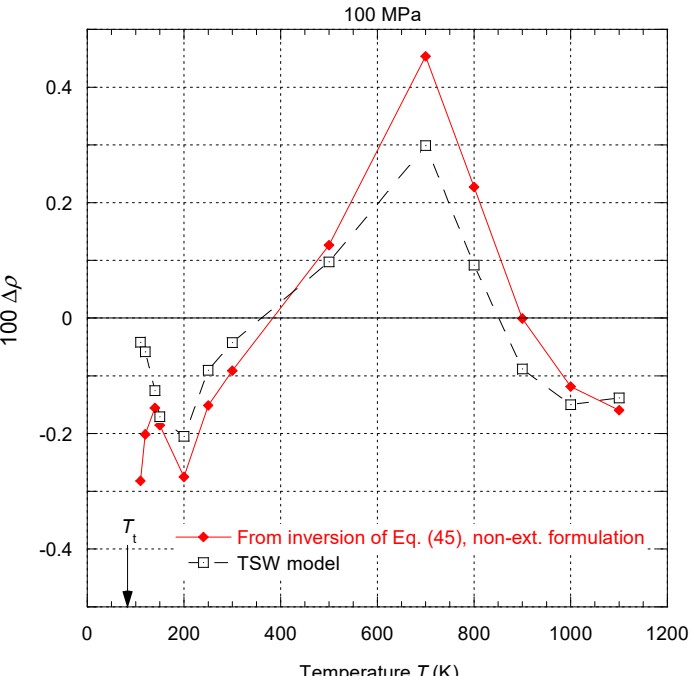

**Figure 19.** Percentage deviations of density $\Delta\rho = \left(\rho_{\text{LAir Liquide}} - \rho_{\text{calc}}\right)/\rho_{\text{LAir Liquide}}$ on the isobar at 100 MPa between the data of L'Air Liquide [5] and the inversion of Equation (45) (red diamonds) or the TSW model (black open squares). The lines are eye guides.

Some data from L'Air Liquide [5] are outside the range of NIST data but are connected to data calculated from the model of Ronchi. Such data from L'Air Liquide can be compared to the calculated data from the two models (TSW and the present one). Figure 20 shows plots of the relative errors in density between the calculated and L'Air Liquide data as a function of pressure on the highest isotherm at 1100 K. The maximum relative error is around 0.3%, and the two models lead to a similar variation with temperature. The error variation is slightly better centered on zero using the present model than the TSW one.

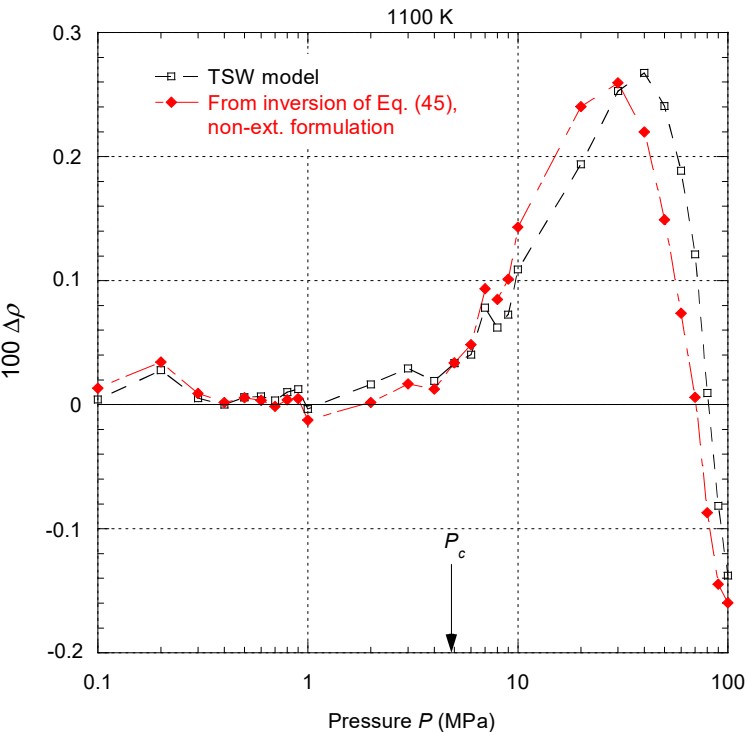

**Figure 20.** Percentage deviations of density $\Delta\rho = \left(\rho_{\text{LAir Liquide}} - \rho_{\text{calc}}\right)/\rho_{\text{LAir Liquide}}$ on the isotherm at 1100 K between the data of L'Air Liquide [5] and the inversion of Equation (45) (red diamonds) or the TSW model (black open squares). The lines are eye guides.

It finally appears that the present model can ultimately better reproduce the thermal properties in the gas phase than in the liquid phase. This is consistent with the fact that the state equations for *U* and *S* are better reproduced at low densities than at high densities. Therefore, if one wants to better reproduce the data in the liquid phase, it is necessary to increase the accuracy of these two functions going towards high densities. It can, however, be noticed that the relative errors on pressure and density as defined by NIST remain very comparable for the two models, with a few exceptions.

In the region covered by the calculated data of Ronchi, it is only possible to use the thermal equations of state $P(\rho, T)$ to compare data. Figure 21 shows the relative error on pressure versus temperature for different isochors. In the region of density covered by NIST data, the relative error from the present model below 700 K is similar to the relative error deduced from the TSW model (see Figure 1). Above 700 K, the maximum error is $-2.5\%$ on the isochor $\rho =$ 1.1784 g/cm³, and above 1000 K, the relative error on all the isochors decreases towards zero. Therefore, up to 2300 K, the overall error using the present model does not exceed the error obtained in the region covered by NIST data. Outside the region of density covered by NIST, Figure 22 shows that the relative error corresponding to the present model is in the range $\pm5\%$, except at low temperature on the two isochors $\rho =$ 1.84944 g/cm³ and $\rho =$ 2.01758 g/cm³. For these isochors, the relative error can be reduced to zero by decreasing the density value corresponding to these isochors by about 0.6%. The uncertainty of $\pm5\%$ corresponds to the uncertainty claimed by Ronchi between his model and the many experimental data he used.

If Ronchi's data are compared with the extrapolation of the TSW model, then Figure 23 shows that the relative error increases with increasing isochor density and reaches a value of 60% on the highest-density isochor. This result was already mentioned in [4].

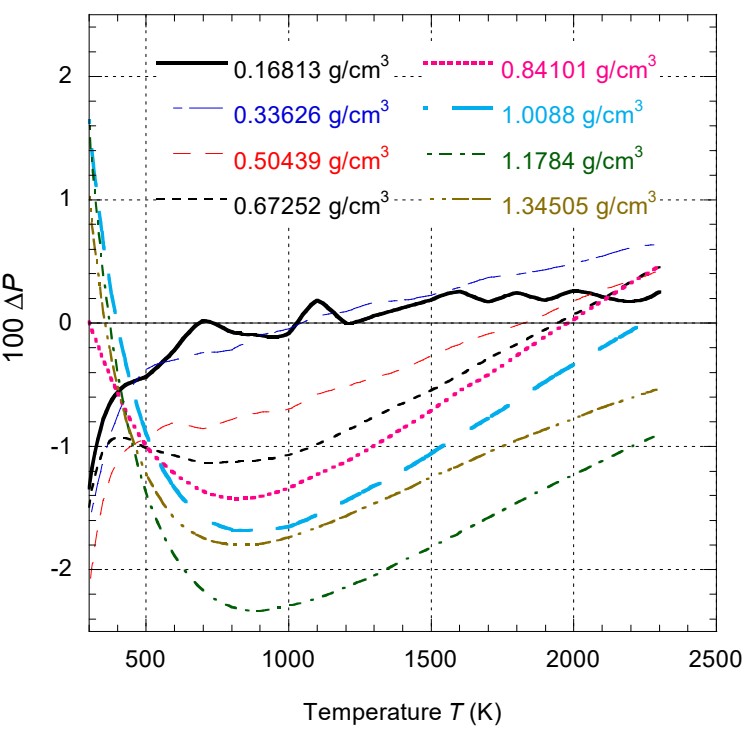

**Figure 21.** Percentage deviations of pressure $\Delta P = (P_{\text{Ronchi}} - P_{\text{calc}})/P_{\text{Ronchi}}$ between the data of Ronchi [7] and Equation (45) along isochors for densities less than $\rho_{\text{t,Liq}}$.

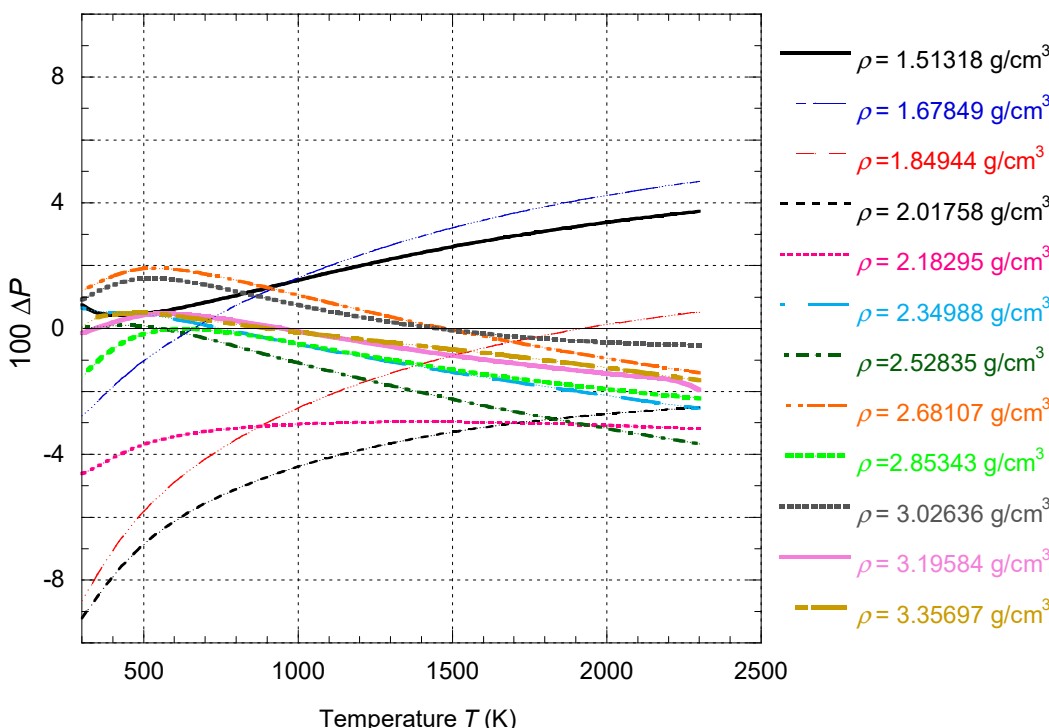

**Figure 22.** Percentage deviations of pressure $\Delta P = (P_{\text{Ronchi}} - P_{\text{calc}})/P_{\text{Ronchi}}$ between the data of Ronchi [7] and Equation (45) along isochors for densities greater than $\rho_{\text{t,Liq}}$.

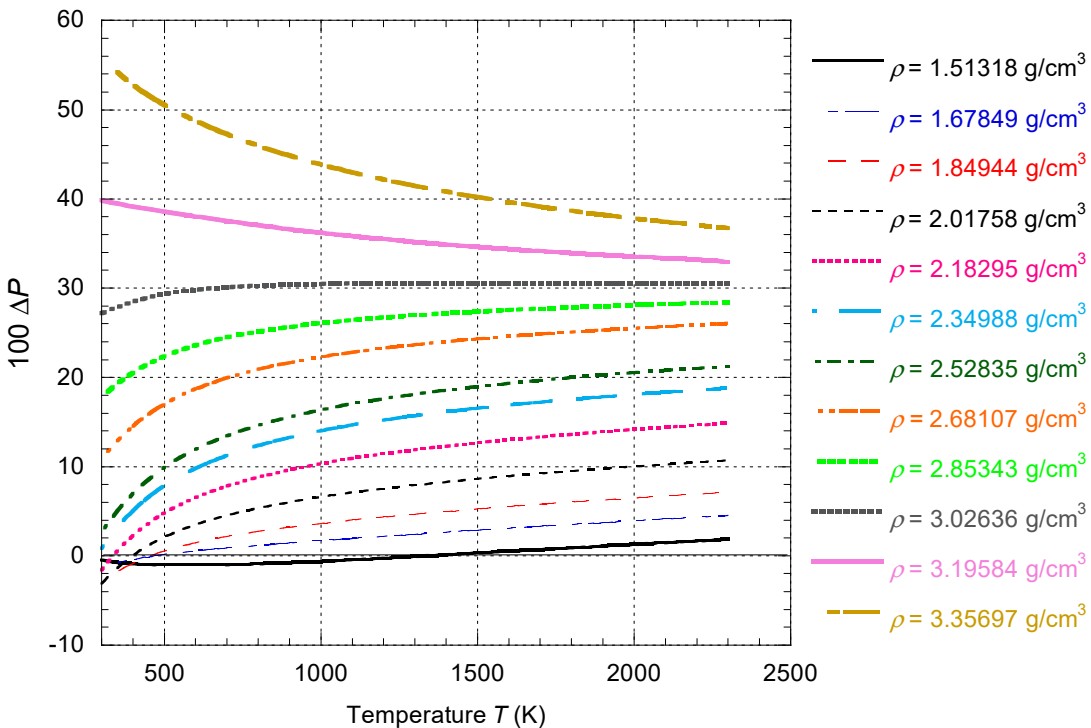

**Figure 23.** Percentage deviations of pressure $\Delta P = (P_{\text{Ronchi}} - P_{\text{TSW model}})/P_{\text{Ronchi}}$ between the data of Ronchi [7] and the TSW model along isochors for densities greater than $\rho_{\text{t,Liq}}$.

New experimental $P\rho T$ data in the supercritical phase at 300 K have been determined by Hanna et al. [22]. The results have been compared with the TSW model and are consistent with it. But due to the large error bars, the present model is also consistent with these new data.

Precise experimental $P\rho T$ data in the gaseous phase in the temperature range of 234 to 505 K have been produced by McLinden [23]. These results have been compared with the values calculated from the TSW model and are consistent with them. In the pressure and temperature ranges covered by these data, the present model has the same precision as that of the TSW model; hence, these data are also consistent with the present model.

4.2.3. Isobaric Heat Capacities, Sound Velocities, and Isothermal Throttling Coefficient

As can be observed from Table 26 of [4], the isobaric heat capacity $C_P(\rho, T)$, the speed of sound $c(\rho, T)$, and the isothermal throttling coefficient $\delta_T(\rho, T) = (\partial H/\partial P)_T$ are functions expressed with first and second derivatives of Helmholtz free energy; therefore, these quantities are more complex with respect to the quantities shown in the previous sections. Given that the present model is not built on free energy but instead on the equation of state of $C_V(\rho, T)$ and thermal state equation $P(\rho, T)$, it is preferable to express the three above quantities as

$$C_P = C_V + TVK_T \left( \frac{\partial P}{\partial T} \right)_V^2 = C_V + TV \frac{\beta^2}{K_T} \tag{68}$$

$$c^2 = \frac{V}{K_T} \frac{C_P}{C_V} \tag{69}$$

$$\delta_T = V(1 - T\beta) \tag{70}$$

where $K_T = -\frac{1}{V} \left( \frac{\partial V}{\partial P} \right)_T$ represents the isothermal compressibility coefficient, and $\beta = \frac{1}{V} \left( \frac{\partial V}{\partial T} \right)_P$ represents the isobaric coefficient of thermal expansion. These quantities include the derivatives of pressure along the two directions $\rho$ and $T$. The two quantities $c$ and $\delta_T$ are functions of $K_T$ and of the ratio $C_P/C_V$. The errors on these two last quantities will reflect in a different way the errors on the state equations for pressure and for the isochoric heat capacity.

Because $C_P$ diverges at the critical point, it is only possible to compare the two models (TSW and the present one) outside the region of coexistence. Figure 24 shows the relative error of $C_P$ between the TSW model and the present one. The relative error is everywhere inside the uncertainty given in Figure 44 of [4]. In particular, Figure 24 shows that, for most of the states, the relative error of the present model oscillates globally, without going into the details, between ±0.5%, except for high-density states and states in the vicinity of the critical point.

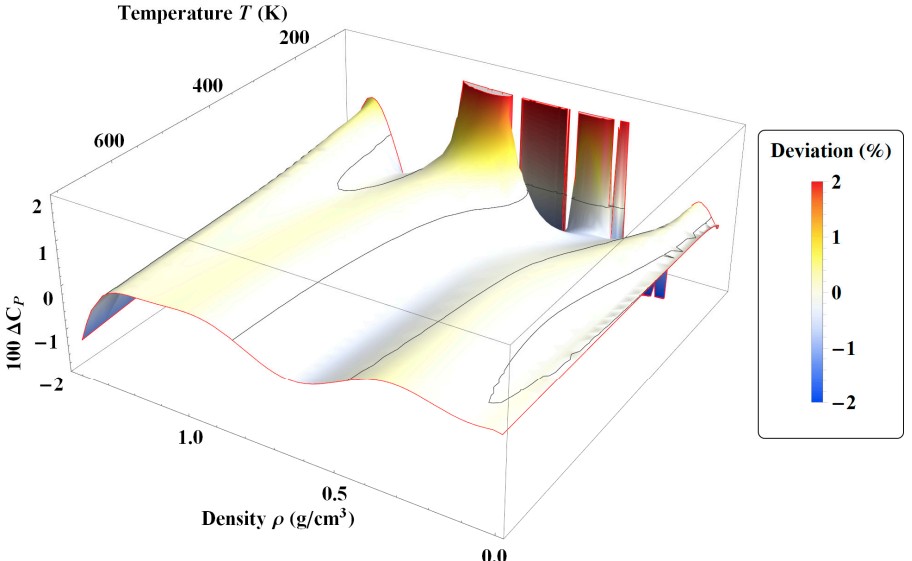

**Figure 24.** Percentage deviations of isobaric heat capacity $\Delta C_P = (C_{P\text{, TSW model}} - C_{P\text{, Eq. (68)}})/C_{P\text{, TSW model}}$ between the TSW model and Equation (68) in the density range of $\rho_{\text{t,Gas}}$ to $\rho_{\text{t,Liq}}$. The black lines correspond to values of $\Delta C_P$ equal to zero.

It is again interesting to compare the results of the two models with the data from L'Air Liquide (729 data points, [5]). Figure 25 displays the data values of L'Air Liquide [5] on two isotherms at 700 and 1100 K (i.e., the highest isotherm) and the corresponding calculated curves from the present approach and the TSW model. As for the $C_V$ data, the present model shows a closer fitting of data from L'Air Liquide [5] than the TSW model. The highest relative error (about 1%) is obtained for the isotherm at 1100 K using the TSW model.

The sound velocity $c$ does not diverge at the critical point but exhibits a very pronounced minimum. However, in the present model, $c$ is expressed on the basis of $C_P$, which diverges itself (Equation (69)). For numerical reasons, we will compare the data calculated from the two models, with the exception of the data on the respective curves of coexistence. The relative error on $c$ (see Figure 26) between the TSW model and the present one shows a very similar variation with $\rho$ and $T$ to the one displayed by $C_P$ in Figure 24. In the largest part of the ($\rho$, $T$) diagram, the relative error oscillates globally at ±0.5%, except for high-density states and near the critical point, where the error reaches 2%. In Figure 43 of [4], the tolerance diagram for $c$ shows similar uncertainties that were obtained in Figure 26; however, some regions of their diagram present lower uncertainties (±0.02% and ±0.1%).

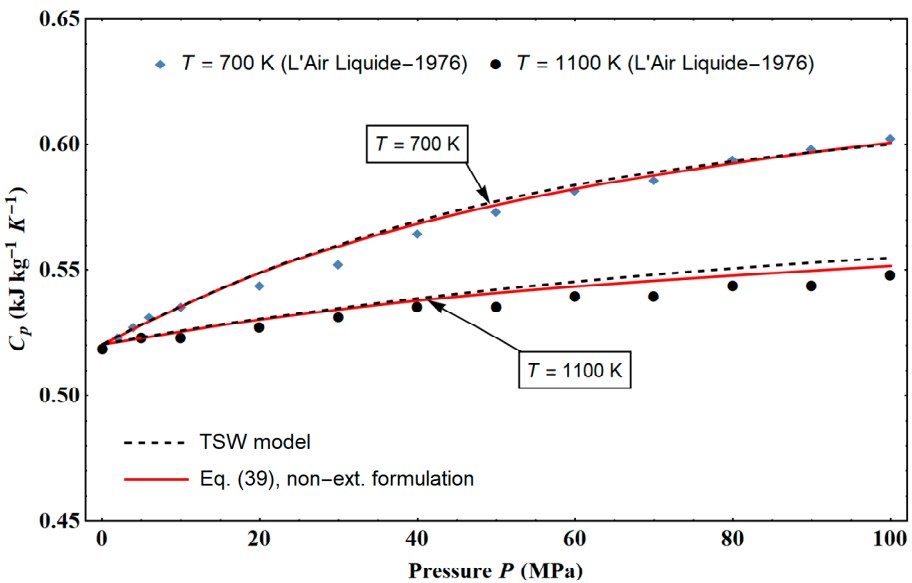

**Figure 25.** Variations with pressure of the isobaric heat capacity $C_P$ along the two isotherms at 700 and 1100 K. The blue diamonds and black points correspond to data from L'Air Liquide [5], and the plotted curves correspond to the values calculated from Equation (39) (red curves) and the TSW model (black dashed curves).

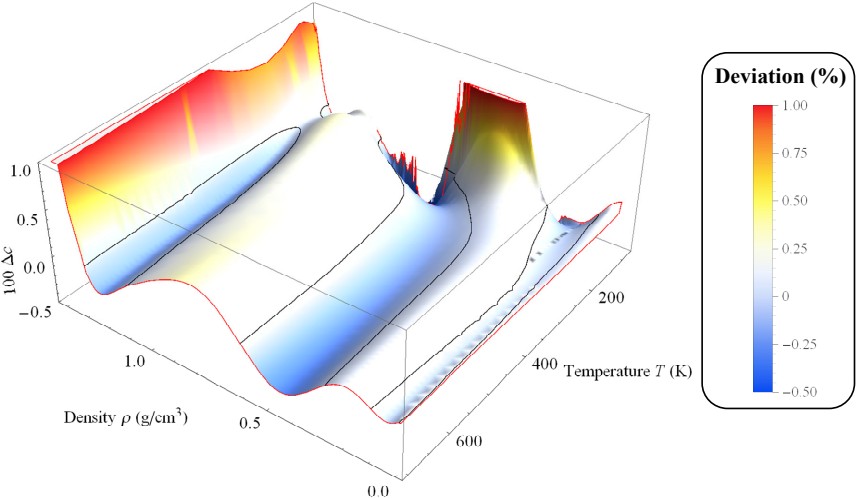

**Figure 26.** Percentage deviations of sound speed $\Delta c = \left( c_{\text{TSW model}} - c_{\text{Eq. (69)}} \right) / c_{\text{TSW model}}$ between the TSW model and Equation (69) in the density range of $\rho_{t,\text{Gas}}$ to $\rho_{t,\text{Liq}}$. The black lines correspond to values of $\Delta c$ equal to zero.

If one compares the calculated data using the two models with the data of L'Air Liquide (296 data points, [5]), it can be observed in Figures 27 and 28 that, although the corresponding errors in the present model are sometimes higher than those of the TSW model, they are generally better centered on zero. This means that the isobars and isotherm variations are better predicted using the present model. Unfortunately, there is no data on sound speed from L'Air Liquide in the range of 700 to 1100 K.

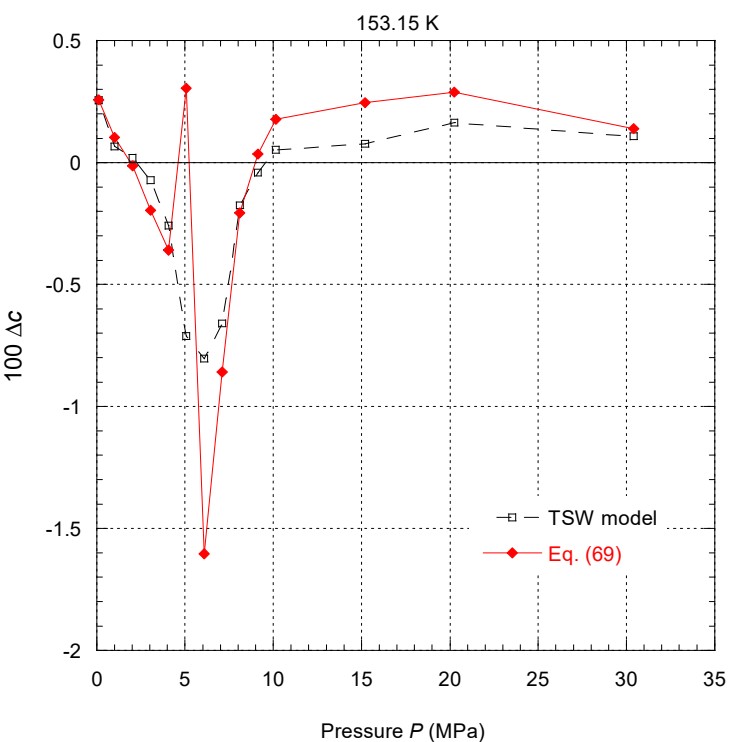

**Figure 27.** Percentage deviations of sound speed $\Delta c = \left( c_{\text{LAir Liquide}} - c_{\text{calc}} \right) / c_{\text{LAir Liquide}}$ along the isotherm at 153.15 K between the data of L'Air Liquide [5] and Equation (69) (red diamonds) or the TSW model (black open squares). The lines are eye guides.

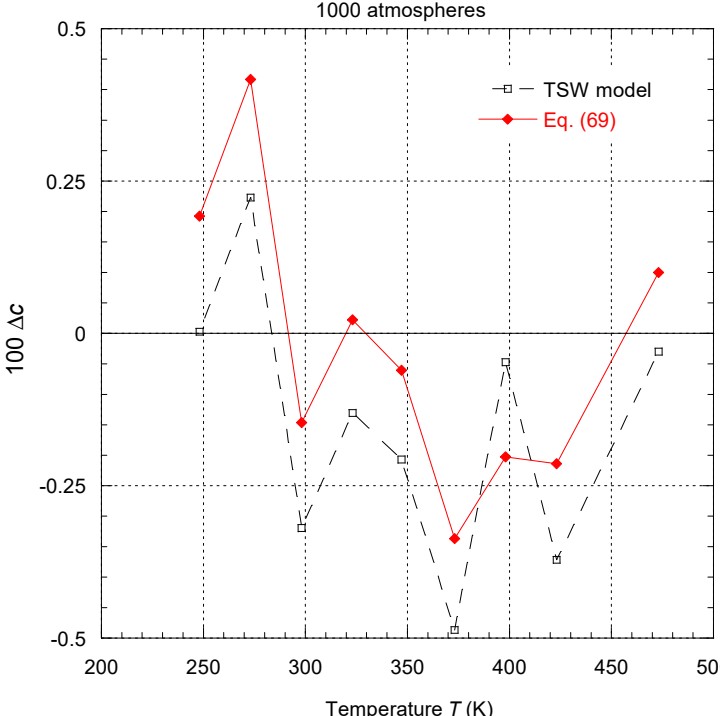

**Figure 28.** Percentage deviations of sound speed $\Delta c = \left( c_{\text{LAir Liquide}} - c_{\text{calc}} \right) / c_{\text{LAir Liquide}}$ along the isobar at 1000 atmospheres between the data of L'Air Liquide [5] and Equation (69) (red diamonds) or the TSW model (black open squares). The lines are eye guides.

Equation (70) can be easily derived from the Gibbs–Helmholtz relation, and its non-dimensional formulation $V^{-1}\delta_T$ almost reflects the behavior of the thermal expansion

coefficient. When this quantity is equal to zero, the fluid behaves as an ideal gas. Due to the fact that zero is a possible value for this function, it is not possible to conduct a relative error analysis. Figure 29 shows the absolute $\rho\delta_T$ vs. $P$ diagram for the same isotherms plotted in Figure 33 of [4]. It can be observed that the difference between the two models is very small and only more pronounced in the vicinity of the minimum of $\rho\delta_T$. On the isotherm at 162 K, the shape for the present modeling has a deeper well, which seems slightly better in light of the data from Kim [24].

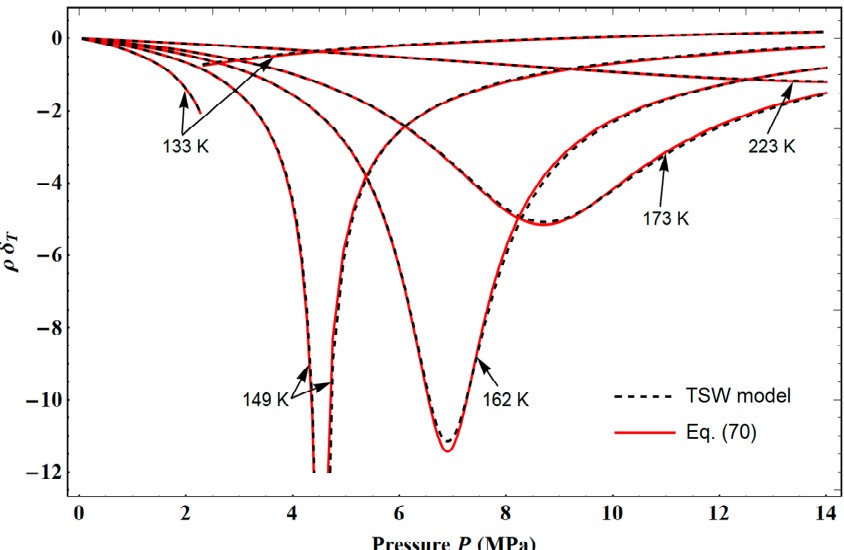

**Figure 29.** Variations with pressure of the dimensionless isothermal throttling coefficient along 5 distinct isotherms. The plotted curves correspond to values calculated from Equation (70) (red curves) and the TSW model (black dashed curves).

Finally, it is important to note that, because the present model gives overall numerical results very close to those of the TSW model, both models have the same weakness for the representation of most of the experimental data very close to the critical point.

### 4.2.4. The "Ideal Curves"

Ideal curves are curves along which one property of a real fluid is equal to the corresponding property of the hypothetical ideal gas in the same state. The most important ideal curves can be obtained from the compressibility factor $Z$ and its first derivatives, i.e., the classical ideal curve ($Z = 1$), the Boyle curve $[(\partial Z/\partial P)_T = 0$ or $(\partial Z/\partial V)_T = 0]$, the Joule–Thomson inversion curve (or Charles curve) $[(\partial Z/\partial T)_P = 0$ or $(\partial Z/\partial V)_P = 0]$, and the Amagat curve (or Joule curve) $[(\partial Z/\partial P)_\rho = 0$ or $(\partial Z/\partial T)_\rho = 0]$. For argon, all ideal curves lie within the range covered by data from NIST and Ronchi, with the exception of the high-temperature part of the Amagat curve.

Figure 30 shows the plot of the ideal curves calculated from Equation (46) and its derivatives and from the TSW model. Inside the single-phase domain, where reliable data exists, both equations show the expected variations in ideal curves. The visible differences occur for the part of each curve corresponding to very low densities. This can be explained by the fact that, in the present model, the various thermodynamic functions are designed to converge to a physically admissible value when density tends towards zero. Another difference can be seen on the high-temperature Amagat curve, which is explained by a better representation of Ronchi's data than the TSW model (as shown in Figures 22 and 23).

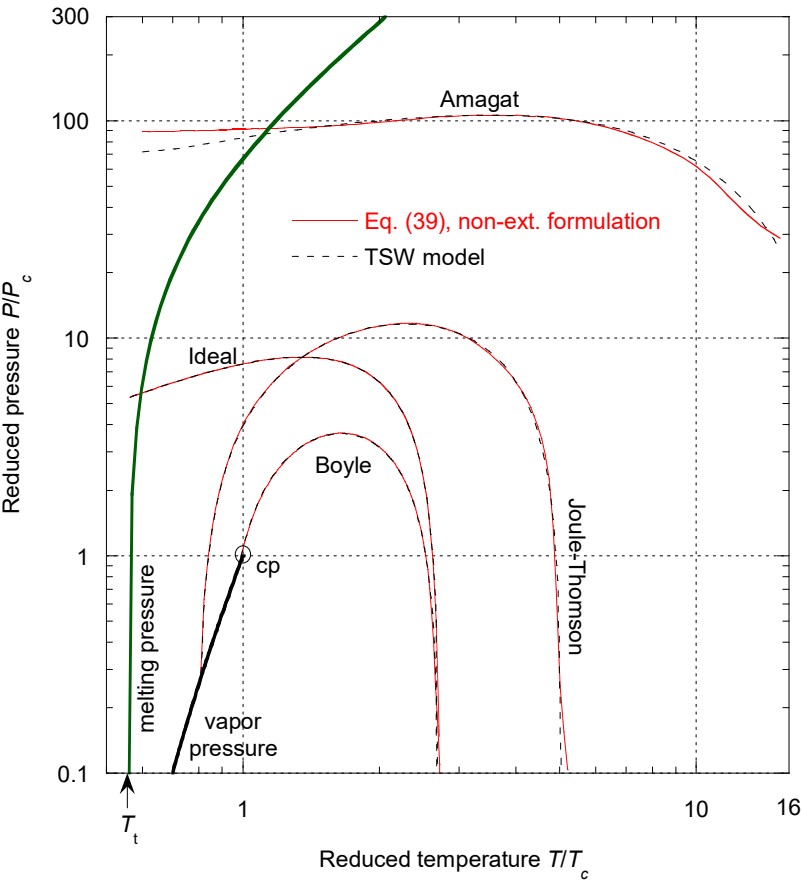

**Figure 30.** Logarithmic plot of the so-called "ideal curves" calculated from Equation (39) (red curves) and the TSW model (black dashed curves) in the temperature range covered by the data from NIST [6] and Ronchi [7]. The black curve represents the saturated vapor pressure curve, ending at the critical point cp. The dark green curve represents the melting curve calculated from Equation (67).

It thus appears that, in the pressure and temperature range covered by the NIST and Ronchi data, the compressibility factor and its first derivatives are well represented by the present model.

4.2.5. Extrapolation to High Temperatures

Tegeler et al. [4] compared their model to data resulting from the shock wave experiments of van Thiel et al. [25], Nellis et al. [26], and Grigor'ev et al. [27]. The pressure and density are calculated from the Hugoniot relations using experimental velocity measurements. All of these data are in the pressure range and density range of Ronchi's data but not in its temperature range. For example, all the data of Grigor'ev et al. [27] correspond to a temperature range of 3700 to 17,000 K. In addition, for most of these experiments, argon is ionized, and the corresponding physics is clearly not included in the present approach, but nor is it explicitly included in the TSW model.

For all data that are in Ronchi's domain, the Hugoniot curve determined with the present model is consistent with the data, as is the Hugoniot curve calculated by the TSW model. This can be easily understood because the Hugoniot states depend mainly on the behavior of the Poisson adiabatic curves, which have close variations until the melting line, as can be seen in Figure 31 for the two different initial states, which correspond to those of van Thiel et al. [25].

From these results, we suggest not extrapolating the present model outside the highest limit of Ronchi's domain.

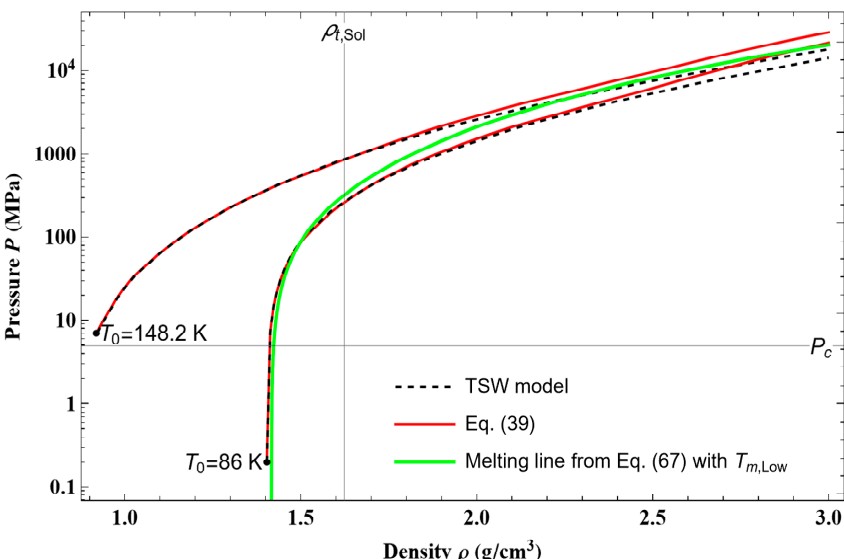

**Figure 31.** Variations with density of the Poisson adiabatic curves calculated from Equation (39) (red curves) and the TSW model (black dashed curves) for the two different initial states of van Thiel et al. [25]. The density $\rho_{t,Sol}$ represents the triple-point solid density.

### 4.3. Liquid–Vapor Phase Boundary

#### 4.3.1. Isochoric Heat Capacities

Along the coexistence curve, the relative errors between the data from NIST and the present modeling oscillate about ±0.45% (see Figure 11). This result can be compared to the uncertainties given in Figure 44 of [4]. The relative error of the present model on the saturated liquid side is smaller than the one given in Figure 44 of [4] (±0.45% instead of ±2%) and slightly larger on the saturated vapor side (±0.45% instead of ±0.3%).

Along this coexistence line, no NIST data are available in the density range of 0.5–0.6 g/cm³. However, this region, which extends on both sides of the critical point, is covered along some isochors that crossed the coexistence curve by Voronel et al. [28,29]. Figure 32 shows that the two models (TSW and the present one) lead to similar variations with $T$, and the discrepancies with the data of Voronel et al. [28,29] increase more and more as one approaches the coexistence curve. Inside the coexistence phase, the data of Voronel et al. [28,29] show a peak in $C_V$ that is not symmetrical. Such $C_V$ variation is, in any case, impossible to reproduce using the TSW model (see Figure 8). On the other hand, the present model could be modified to correctly describe such $C_V$ variations. Indeed, the parameter $T_{div}$ was inserted into the model (i.e., Equation (10)) to qualitatively describe the $C_V$ divergence inside the coexistence phase (see Figure 8). It is, however, evident from the data of Voronel et al. [28,29] that the values defined by $T_{div}$ are not quantitatively suitable. The peak position of $C_V$ along the entire coexistence curve could be used to establish a new equation for $T_{div}$, leading to a reliable fitting of $C_V$ divergence inside the coexistence phase. The other side of $T_{div}$ (i.e., for $T < T_{div}$) could also be easily modeled without changing any properties for the single-phase region. Unfortunately, the data of Voronel et al. [28,29], which are limited to a very small density range, are insufficient to be taken into account in view to improve the present model into the coexistence phase. It can also be noted that the data of Voronel et al. [28,29] have been correctly modeled by Rizi et al. [30] using the crossover model. However, this model, which contains coefficients among which a number unknown, cannot be put into practice.

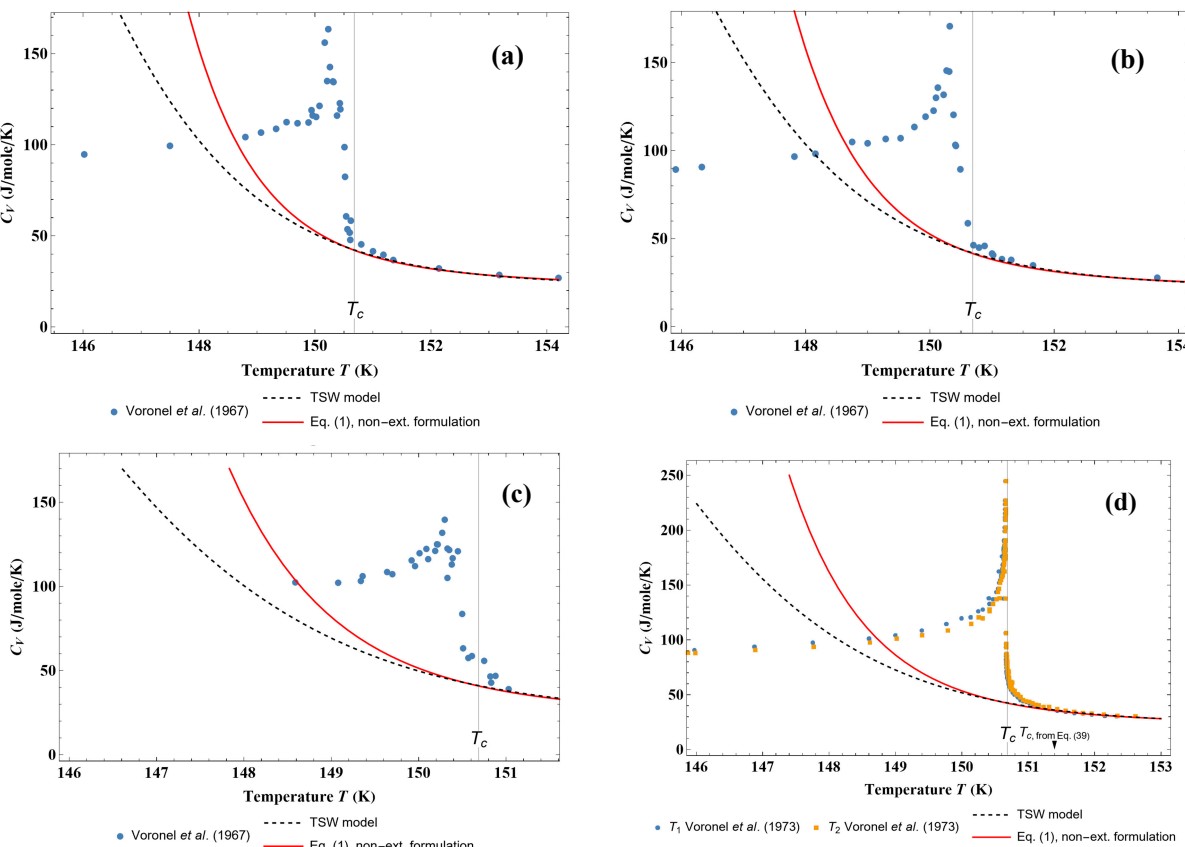

**Figure 32.** Variations with temperature of the isochoric heat capacity $C_V$ in the vicinity of the critical point along four isochors: (**a**) 0.504 g/cm$^3$, (**b**) 0.549 g/cm$^3$, (**c**) 0.560 g/cm$^3$, and (**d**) 0.531 g/cm$^3$. The data points are from Voronel et al. [28,29], and the plotted curves correspond to values calculated from Equation (1) (red curves) and the TSW model (black dashed curves).

### 4.3.2. Thermal Properties

The coexistence phase is characterized by three specific points, which are the saturated liquid triple point, the saturated vapor triple point, and the critical point. These points correspond to different well-known thermodynamic states.

From the thermal equation of state $P(\rho, T)$ of the TSW model and the present one, the specific points were calculated using the data from NIST for $\rho$ and $T$. Table 10 shows that the calculated values of the three characteristic states using the present model are globally more accurate than the ones calculated using the TSW model, particularly for the triple point on the saturated liquid curve. As shown in Table 10, near the liquid saturated curve, even a tiny variation in density produces a large variation in the calculated pressure, i.e., the density values for these states must be extremely accurate. Thus, the TSW model gives an error of 0.003% on $\rho_{t,Liq}(P_t, T_t)$, and this leads to an error of 20% on $P_t(\rho_{t,Liq}, T_t)$.

**Table 10.** Characteristic values of the coexistence line calculated from the thermal equation of state and using the NIST values.

| | Unit | NIST | TSW Model | Non-Extensive Formulation, Equation (45) |
|---|---|---|---|---|
| $P_c(\rho_c, T_c)$ | MPa | 4.863 | 4.86299 | 4.86298 |
| $\rho_c(P_c, T_c)$ | g/cm$^3$ | 0.535599 | 0.549928 | 0.535526 |
| $P_t(\rho_{t,Liq}, T_t)$ | MPa | 0.068891 | 0.082671 | 0.0688907 |
| $\rho_{t,Liq}(P_t, T_t)$ | g/cm$^3$ | 1.4168 | 1.41676 | 1.41680 |
| $P_t(\rho_{tGas}, T_t)$ | MPa | 0.068891 | 0.0688913 | 0.068891 |
| $\rho_{t,Gas}(P_t, T_t)$ | g/cm$^3$ | 0.0040546 | 0.00405458 | 0.00405460 |

At a given temperature, the vapor pressure and the densities of the coexisting phases ($\rho_{\sigma l}$ and $\rho_{\sigma v}$) can also be calculated from the Maxwell criterion of phase equilibrium conditions and, therefore, the characteristic values of the coexistence line (triple and critical points) as well. Table 11 shows these characteristic values calculated from the TSW model and the present model (i.e., Equations (50)–(52)). The data from NIST and those calculated with the TSW model are in good agreement, but this is not the case for the present model, particularly for the critical point. However, it can be noticed that $T_c$ and $T_t$ are imposed values for the TSW model, whereas only $T_t$ is fixed in the present one. Then, the critical values ($P_c$, $T_c$, and $\rho_c$) are calculated in the present approach. Equation (45) leads to better results for characteristic values than those calculated with the TSW model, while using the Maxwell equations, it is the opposite.

**Table 11.** Characteristic values of the coexistence line calculated from the Maxwell relations, i.e., Equations (50)–(52).

|  | NIST | TSW Model | Non-Extensive Formulation |
|---|---|---|---|
| $T_c$ (K) | 150.687 | 150.687 | 151.396 |
| $P_c$ (MPa) | 4.863 | 4.86295 | 4.99684 |
| $\rho_c$ (g/cm$^3$) | 0.535599 | 0.533136 | 0.543786 |
| $T_t$ (K) | 83.8058 | 83.8058 | 83.8058 |
| $P_t$ (MPa) | 0.068891 | 0.0688908 | 0.0689657 |
| $\rho_{t,\text{Liq}}$ (g/cm$^3$) | 1.4168 | 1.41676 | 1.416802 |
| $\rho_{t,\text{Gas}}$ (g/cm$^3$) | 0.0040546 | 0.00405457 | 0.00405912 |

How do we explain this result? The Maxwell equations represent the equality of pressure, temperature, and specific Gibbs energy in the coexisting phases. The present approach built all the required thermodynamic quantities ($U$, $S$, $F$, etc.) from the empirical description of the experimental data of $C_V(\rho, T)$ and $P(\rho, T)$. The accuracy of these empirical equations to describe the experimental data has been shown to be of very high quality. Therefore, the discrepancy between the critical values ($P_c$, $T_c$, and $\rho_c$) calculated with the present model and those from NIST can be attributed to the inconsistency of the data in this critical state. The agreement between the calculated liquid triple point and the experimental one is better using the present model; this means that, on the liquid side, the present isochors network is slightly twisted compared to the TSW model network. This slight distortion of the isochor network on the liquid side then has a strong impact on the construction of the coexistence curve, as already shown in Table 11. Indeed, considering that there is good agreement with the TSW model on the gas side but not as good on the liquid side, it is clear that the equilibrium conditions deduced from the Maxwell relations must be different. Table 12 gives a numerical summary of all the thermodynamic quantities on the saturation curve using the present non-extensive formulation.

**Table 12.** Thermodynamic parameters of saturated argon. For each temperature, the first line corresponds to the liquid and the second line to the gas.

| Temperature (K) | Pressure (MPa) | Density (g cm$^{-3}$) | Enthalpy (kJ kg$^{-1}$) | Entropy (kJ kg$^{-1}$ K$^{-1}$) | $C_V$ (kJ kg$^{-1}$ K$^{-1}$) | $C_P$ (kJ kg$^{-1}$ K$^{-1}$) | $c$ (m s$^{-1}$) |
|---|---|---|---|---|---|---|---|
| 83.8058 [a] | 0.068891 | 1.41680 | −121.39 | 1.3297 | 0.54864 | 1.11895 | 856.17 |
|  |  | 0.0040546 | 42.23 | 3.2824 | 0.32640 | 0.55734 | 168.02 |
| 84 | 0.070522 | 1.41561 | −121.17 | 1.3323 | 0.54795 | 1.11968 | 854.73 |
|  |  | 0.0041430 | 42.31 | 3.2786 | 0.32660 | 0.55794 | 168.17 |
| 86 | 0.088193 | 1.40321 | −118.92 | 1.3586 | 0.54097 | 1.12304 | 840.88 |
|  |  | 0.0050845 | 43.08 | 3.2425 | 0.32855 | 0.56400 | 169.74 |
| 88 | 0.109096 | 1.39072 | −116.67 | 1.3844 | 0.53426 | 1.12305 | 828.22 |
|  |  | 0.0061790 | 43.82 | 3.2063 | 0.33063 | 0.57071 | 171.24 |
| 90 | 0.133597 | 1.37815 | −114.41 | 1.4095 | 0.52782 | 1.12402 | 815.37 |
|  |  | 0.0074416 | 44.52 | 3.1755 | 0.33284 | 0.57812 | 172.67 |
| 92 | 0.162078 | 1.36549 | −112.15 | 1.4342 | 0.52164 | 1.12613 | 802.28 |

**Table 12.** *Cont.*

| Temperature (K) | Pressure (MPa) | Density (g cm$^{-3}$) | Enthalpy (kJ kg$^{-1}$) | Entropy (kJ kg$^{-1}$ K$^{-1}$) | $C_V$ (kJ kg$^{-1}$ K$^{-1}$) | $C_P$ (kJ kg$^{-1}$ K$^{-1}$) | $c$ (m s$^{-1}$) |
|---|---|---|---|---|---|---|---|
|  |  | 0.0088882 | 45.18 | 3.1443 | 0.33519 | 0.58629 | 174.02 |
| 94 | 0.194930 | 1.35271 | −109.88 | 1.4583 | 0.51569 | 1.12944 | 788.92 |
|  |  | 0.0105356 | 45.80 | 3.1145 | 0.33768 | 0.59531 | 175.30 |
| 96 | 0.232558 | 1.33979 | −107.60 | 1.4820 | 0.50996 | 1.13400 | 775.28 |
|  |  | 0.0124010 | 46.37 | 3.0859 | 0.34032 | 0.60523 | 176.51 |
| 98 | 0.275373 | 1.32672 | −105.31 | 1.5053 | 0.50445 | 1.13988 | 761.35 |
|  |  | 0.0145031 | 46.89 | 3.0584 | 0.34311 | 0.61617 | 177.64 |
| 100 | 0.323796 | 1.31346 | −103.00 | 1.5282 | 0.49915 | 1.14718 | 747.12 |
|  |  | 0.0168613 | 47.36 | 3.0319 | 0.34607 | 0.62824 | 178.69 |
| 102 | 0.378255 | 1.29999 | −100.68 | 1.5508 | 0.49405 | 1.15597 | 732.59 |
|  |  | 0.0194966 | 47.78 | 3.0063 | 0.34919 | 0.64156 | 179.68 |
| 104 | 0.439185 | 1.28630 | −98.34 | 1.5730 | 0.48916 | 1.16640 | 717.75 |
|  |  | 0.0224312 | 48.14 | 2.9816 | 0.35251 | 0.65630 | 180.58 |
| 106 | 0.507023 | 1.27233 | −95.97 | 1.5951 | 0.48446 | 1.17859 | 702.59 |
|  |  | 0.0256892 | 48.43 | 2.9575 | 0.35603 | 0.67264 | 181.41 |
| 108 | 0.582216 | 1.25808 | −93.58 | 1.6169 | 0.47997 | 1.19271 | 687.10 |
|  |  | 0.0292969 | 48.67 | 2.9341 | 0.35977 | 0.69081 | 182.17 |
| 110 | 0.665211 | 1.24349 | −91.15 | 1.6385 | 0.47568 | 1.20896 | 671.25 |
|  |  | 0.0332828 | 48.83 | 2.9112 | 0.36375 | 0.71110 | 182.84 |
| 112 | 0.756461 | 1.22855 | −88.70 | 1.6600 | 0.47160 | 1.22758 | 655.04 |
|  |  | 0.0376785 | 48.92 | 2.8888 | 0.36801 | 0.73383 | 183.44 |
| 114 | 0.856424 | 1.21319 | −86.20 | 1.6814 | 0.46775 | 1.24886 | 638.45 |
|  |  | 0.0425194 | 48.94 | 2.8669 | 0.37257 | 0.75942 | 183.97 |
| 116 | 0.965560 | 1.19739 | −83.66 | 1.7027 | 0.46415 | 1.27315 | 621.45 |
|  |  | 0.0478449 | 48.86 | 2.8452 | 0.37747 | 0.78839 | 184.41 |
| 118 | 1.0843 | 1.18108 | −81.07 | 1.7240 | 0.46080 | 1.30090 | 604.01 |
|  |  | 0.0537001 | 48.70 | 2.8238 | 0.38276 | 0.82138 | 184.77 |
| 120 | 1.2132 | 1.16422 | −78.43 | 1.7452 | 0.45774 | 1.33265 | 586.10 |
|  |  | 0.0601365 | 48.44 | 2.8026 | 0.38849 | 0.85921 | 185.05 |
| 122 | 1.3526 | 1.14673 | −75.72 | 1.7666 | 0.45501 | 1.36910 | 567.70 |
|  |  | 0.0672141 | 48.08 | 2.7814 | 0.39473 | 0.90296 | 185.24 |
| 124 | 1.5032 | 1.12853 | −72.95 | 1.7880 | 0.45264 | 1.41114 | 548.75 |
|  |  | 0.0750035 | 47.60 | 2.7603 | 0.40154 | 0.95400 | 185.33 |
| 126 | 1.6653 | 1.10955 | −70.11 | 1.8097 | 0.45070 | 1.45995 | 529.20 |
|  |  | 0.0835889 | 46.99 | 2.7390 | 0.40904 | 1.01421 | 185.34 |
| 128 | 1.8394 | 1.08966 | −67.17 | 1.8315 | 0.44926 | 1.51708 | 509.01 |
|  |  | 0.0930725 | 46.24 | 2.7176 | 0.41733 | 1.08612 | 185.24 |
| 130 | 2.0261 | 1.06875 | −64.14 | 1.8536 | 0.44841 | 1.58468 | 488.11 |
|  |  | 0.103580 | 45.33 | 2.6958 | 0.42658 | 1.17327 | 185.03 |
| 132 | 2.2260 | 1.04666 | −61.00 | 1.8762 | 0.44830 | 1.66578 | 466.42 |
|  |  | 0.115272 | 44.23 | 2.6735 | 0.43697 | 1.28078 | 184.69 |
| 134 | 2.4395 | 1.02319 | −57.73 | 1.8992 | 0.44909 | 1.76485 | 443.85 |
|  |  | 0.128351 | 42.94 | 2.6505 | 0.44877 | 1.41628 | 184.21 |
| 136 | 2.6672 | 0.998082 | −54.31 | 1.9229 | 0.45102 | 1.88882 | 420.29 |
|  |  | 0.143090 | 41.39 | 2.6266 | 0.46233 | 1.59167 | 183.57 |
| 138 | 2.9099 | 0.970997 | −50.71 | 1.9474 | 0.45442 | 2.04902 | 395.56 |
|  |  | 0.159864 | 39.55 | 2.6015 | 0.47818 | 1.82657 | 182.73 |
| 140 | 3.1682 | 0.941443 | −46.88 | 1.9730 | 0.45984 | 2.26491 | 369.39 |
|  |  | 0.179210 | 37.35 | 2.5747 | 0.49706 | 2.15574 | 181.63 |
| 142 | 3.4430 | 0.908698 | −42.77 | 2.0000 | 0.46824 | 2.57120 | 341.35 |
|  |  | 0.201946 | 34.68 | 2.5455 | 0.52030 | 2.64684 | 180.19 |
| 144 | 3.7351 | 0.871668 | −38.28 | 2.0292 | 0.48183 | 3.03858 | 310.73 |
|  |  | 0.229420 | 31.38 | 2.5129 | 0.55034 | 3.45008 | 178.23 |
| 146 | 4.0460 | 0.828488 | −33.24 | 2.0614 | 0.50566 | 3.85590 | 276.57 |
|  |  | 0.264135 | 27.13 | 2.4749 | 0.59279 | 4.97016 | 175.27 |
| 148 | 4.3773 | 0.775089 | −27.30 | 2.0989 | 0.55048 | 5.71750 | 237.51 |
|  |  | 0.311718 | 21.29 | 2.4273 | 0.66271 | 8.72009 | 169.94 |
| 150 | 4.7323 | 0.698343 | −19.28 | 2.1495 | 0.64763 | 13.9176 | 190.54 |
|  |  | 0.390945 | 11.82 | 2.3569 | 0.80939 | 26.4188 | 157.52 |
| 150.687 [b] | 4.8607 | 0.656707 | −15.14 | 2.1758 | 0.71568 | 29.7562 | 171.24 |
|  |  | 0.439127 | 6.406 | 2.3188 | 0.89167 | 56.4503 | 149.66 |
| 151.396 [c] | 4.99684 | 0.543786 | −4.184 | 2.2468 | 0.87926 | 3580 | 146.56 |

[a] Triple-point temperature.  [b] Critical temperature from NIST.  [c] Critical temperature from Maxwell relations Equations (50)–(52).

Figure 33 shows that the relative error between the TSW model and the present one for the saturated liquid density is less than $\pm 0.2\%$ in the range $T_\text{t}$ to 139 K, which is within the uncertainty of the data selected by Tegeler et al. (see Figure 5 of [4]). Therefore, on the liquid side, the network of isochors induced by the present model below 139 K is clearly more realistic.

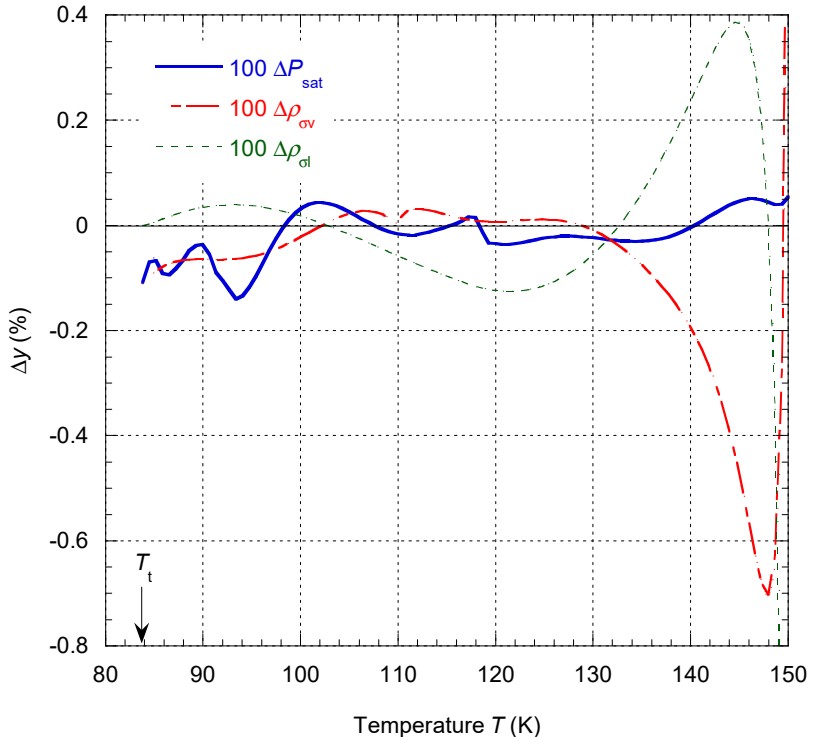

**Figure 33.** Percentage deviations $\Delta y = (y_\text{NIST} - y_\text{calc})/y_\text{NIST}$ of the selected thermal data at saturation from values calculated from Equation (39) in the temperature range of $T_\text{t}$ to 149.5 K.

Also, Figure 33 shows that the relative error between the TSW model and the present one for the saturated vapor density is within the uncertainty of the data selected by Tegeler et al. (see Figure 6 of [4]) in the range 100 K to 149.5 K. Below 100 K, the error of the present model is 3 to 4 times larger than the claimed experimental uncertainties of Gilgen et al. [31]. But, as mentioned by Tegeler et al., the densities on the saturated vapor curve were extrapolated from measurement in the homogeneous region close to the phase boundary. Such density values are obviously depending on the method used for doing the extrapolation. Although the error of the present model in this region is relatively high, the calculated density values are compatible with the experimental data.

Finally, Figure 33 also shows that the relative error between the TSW model and the present one for the saturated vapor pressure is within the uncertainty of the data selected by Tegeler et al. in the range of 97 to 144 K. Below 97 K, the error of the present model oscillates slightly around $-0.1\%$, which is within the uncertainty of the data assigned to Group 2 by Tegeler et al.; therefore, it can be said that these results are also in agreement with the experimental data.

However, in the vicinity of the critical state, the two models lead to very different values. This is due to the different approaches used for the two models. For the TSW model, the parameters of the critical point are imposed, whereas they are calculated in the present model. Apart from the numerical values, Figure 34 shows that the shape of the saturated vapor pressure curve around the critical point depends on the model. The TSW model generates an extremely "flat" variation on a wide range of densities around the critical point, whereas the present model produces a more rounded variation in the same range of density. This last variation is closer to the experimental saturated vapor pressure curve from L'Air Liquide [5]

than the one given by the TSW model. The fact that the critical state is imposed in the TSW model seems to lead to a forced flattening out of the saturated vapor pressure curve at the critical point.

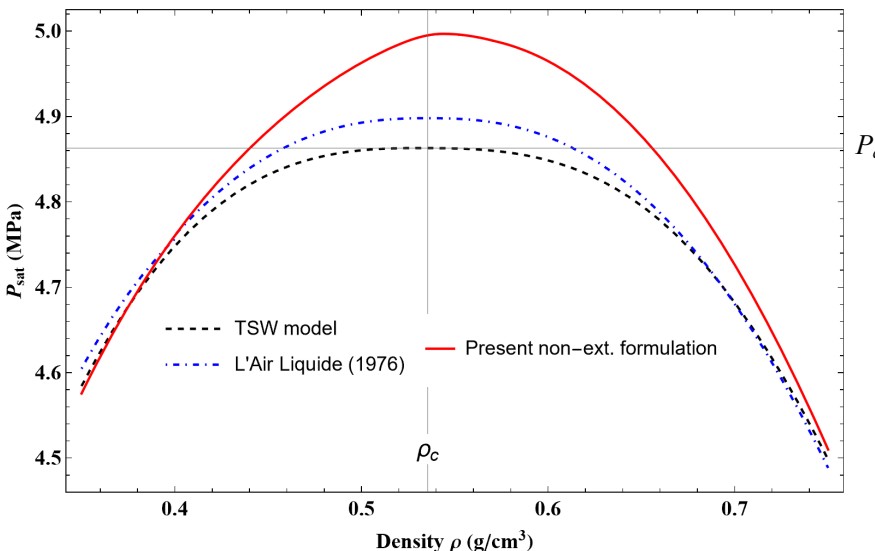

**Figure 34.** Variations with density of the saturated pressure in the vicinity of the critical point (i.e., from 0.35 to 0.75 g/cm$^3$). The plotted curves correspond to values calculated from Maxwell relations Equations (50)–(52) (red curve), the TSW model (black dashed curve), and the data of L'Air Liquide (blue dot-dashed curve, [5]).

If we use $T_{\mathrm{sat}}(\rho)$ on the phase boundary derived from NIST data and calculate the saturated pressure curve using Equation (45), then Figure 35 shows that the relative error on the saturated vapor pressure curve $P_{\mathrm{sat}}(\rho)$ is less than $\pm 0.2\%$ below $\rho_c$. This uncertainty is compatible with the uncertainty of the data assigned to Group 2 by Tegeler et al. On the other hand, the relative error on the saturated liquid pressure curve $P_{\mathrm{sat}}(\rho)$ in the range $\rho_c$ to 0.85 g/cm$^3$ is compatible with the uncertainty of the data assigned to Group 3 by Tegeler et al. From this, it can be concluded that the present thermal equation of state is probably not accurate enough in the range of 145 K to $T_c$.

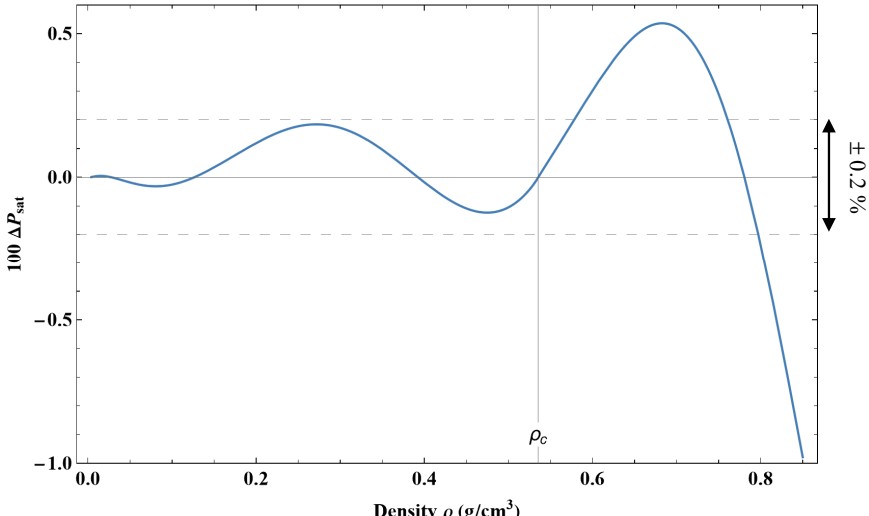

**Figure 35.** Percentage deviations of saturated pressure $\Delta P_{\mathrm{sat}} = (P_{\mathrm{TSW\,mod.}}(T_{\mathrm{sat}}(\rho), \rho) - P_{\mathrm{calc}}(T_{\mathrm{sat}}(\rho), \rho))/P_{\mathrm{TSW\,mod.}}(T_{\mathrm{sat}}(\rho), \rho)$ between the TSW model and Equation (45) in the density range of $\rho_{\mathrm{t,Gas}}$ to 0.85 g/cm$^3$. The curve $T_{\mathrm{sat}}(\rho)$ used is an interpolation of the data from NIST [6].

From the analysis of the latent heat of vaporization $L_v = H_{\sigma v} - H_{\sigma l}$, the effect of cumulative errors between the properties on the saturated vapor and saturated liquid sides can be determined. Figure 36 shows that, until 149.5 K, the relative error between the TSW model and ours is far greater than the experimental data uncertainties shown in Figure 15 of [4], and more particularly, from $T_t$ to 134 K, the relative error is less than $\pm0.1\%$.

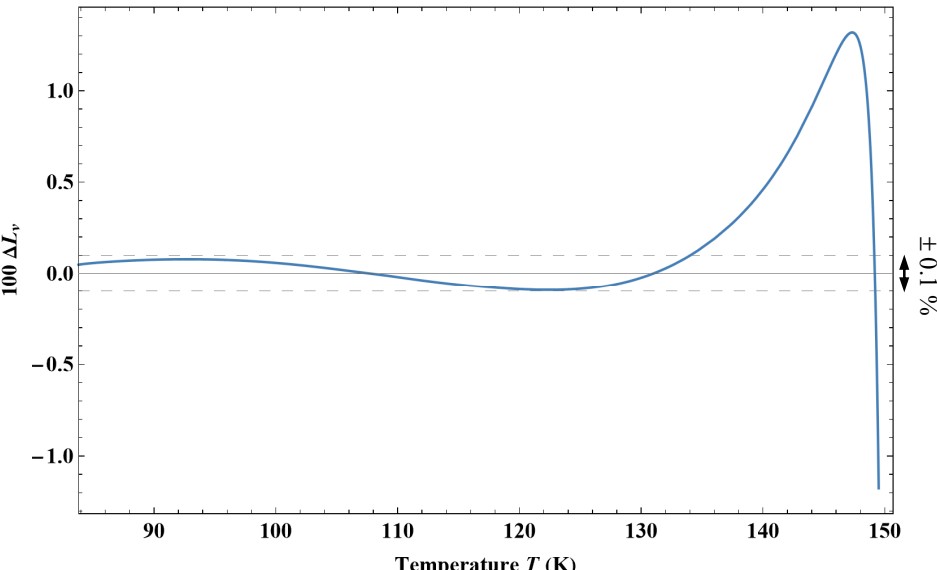

**Figure 36.** Percentage deviations of latent heat of vaporization $\Delta L_v = \left( L_{v,\text{TSW model}} - L_{v,\text{Eq. (39)}} \right) / L_{v,\text{TSW model}}$ between the TSW model and Equation (39) in the temperature range of $T_t$ to 149.5 K.

Owing to the fact that the "ideal curves" are correctly described, in particular near the critical point, but that the saturated pressure is not correctly reproduced in the vicinity of the critical point, it can be concluded that the problem comes from the fact that the experimental data in this region are not correctly described and, moreover, are not coherent enough between themselves. These arguments can be easily observed with the spinodal properties.

### 4.3.3. The Spinodal Properties

The spinodal properties correspond to the metastable states of the fluid system. The knowledge of these metastable states is important for industrial processes that involve ever-increasing heat fluxes and rapid transients but also for testing the validity of a new equation of state formulation.

Most of the available experimental data pertain to states much closer to the saturated liquid state than the spinodal limit, except very close to the critical point. The experimental data of Voronel et al. [28,29] crossed the spinodal limit in a very narrow range of density around the critical point, and the divergence states are shown in Figure 37. This figure also shows the liquid spinodal data points from Baıdakov et al. [32]. These data were determined from experimental $P\rho T$ data combined with a simple theoretical equation of $C_V(\rho, T)$. Therefore, these data points are dependent on the theoretical variations in $C_V$ chosen by Baıdakov et al. [32]. Figure 37 shows that these data decrease rapidly as the density increase, but they are compatible with the spinodal states determined from the present approach or from the TSW model, except for the TSW model around the density of 0.8 g/cm$^3$, where a strong unphysical hole appears due to uncontrollable strong oscillations of the polynomial terms.

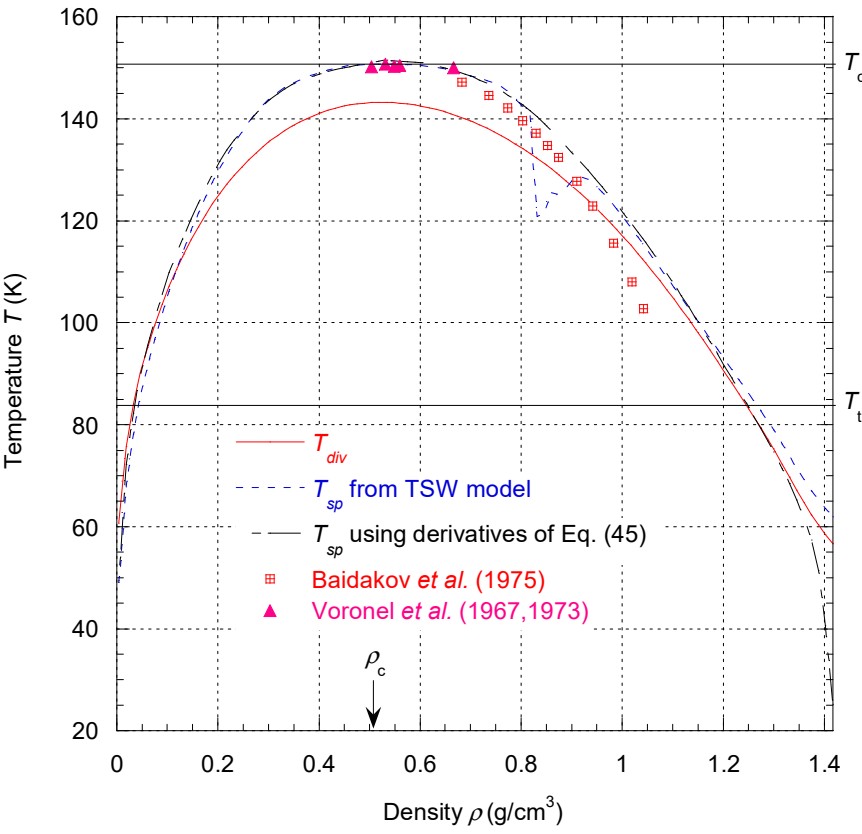

**Figure 37.** Variations with the density of the spinodal temperature. The plotted curves correspond to the divergence curve of $C_V$, the calculated values using the derivation of Equation (45), and the TSW model. The data points are from Voronel et al. [28,29] and Baidakov et al. [32].

Figure 37 shows that, globally, the two spinodal curves determined from the present approach and the TSW model are very close, except for the liquid triple point, where our spinodal curve shows a strong decrease vs. density. This is due to the fact that in this region, the $P\rho T$ data are not represented with enough accuracy by the TSW model. It has already been seen in previous sections that the present model has better accuracy in this region, which was obtained by "twisting" the isochors network. The strong decrease in the spinodal curve near the liquid triple point is simply the result of this local network deformation.

From Figure 37, it is possible to compare our divergence curve of $C_V$ with the spinodal curve. If the data sets of $C_V$ and $P\rho T$ used for the theoretical developments were sufficiently coherent, both curves would be identical. This is approximately true only on the liquid side for densities higher than 0.9 g/cm$^3$ and on the gaseous side in the density range of 0.025 to 0.15 g/cm$^3$. Elsewhere, this is not the case, showing that the variations in $C_V$ close to the saturation curve are not correctly represented. Because high accuracy is obtained with the TSW model, it means that the variations in $C_V$ calculated from the TSW model do not have a good shape. This shows that it is not enough to have great precision with a priori selected set of data to ensure a coherent representation. Local variations in some measured quantities have physical, non-negligible importance.

## 5. Uncertainty of the New Equation of State

Mainly guided by comparison with the TSW model, estimates for the uncertainty of calculated densities $\rho$, speeds of sound $c$, and isobaric heat capacities $C_P$ calculated from Equation (39) were made. These uncertainties are illustrated in the following tolerance diagrams: Figures 17, 38 and 39. For all these tolerance diagrams, the variables are the pressure $P$ and the temperature $T$. Because the quantities $c$ and $C_P$ depend on $\rho$ and $T$, the pressure was converted to density by inversion of Equation (45). In order to make an easier comparison with

the tolerance diagrams given in [4], we used the same tolerance ranges ($\pm 0.03$ to $\pm 5\%$) and identical notations (A, B, C, D, E, and F) with their corresponding meanings.

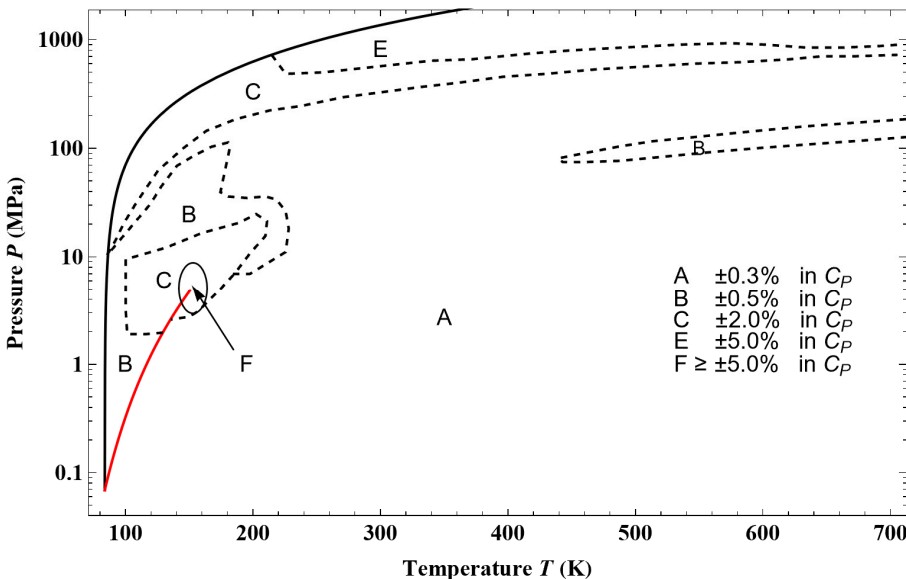

**Figure 38.** Tolerance diagram for isobaric heat capacities calculated from Equation (68) with the use of Equation (45) for determining densities as a function of pressure. The red curve corresponds to the saturated vapor pressure curve and the black one to the melting line.

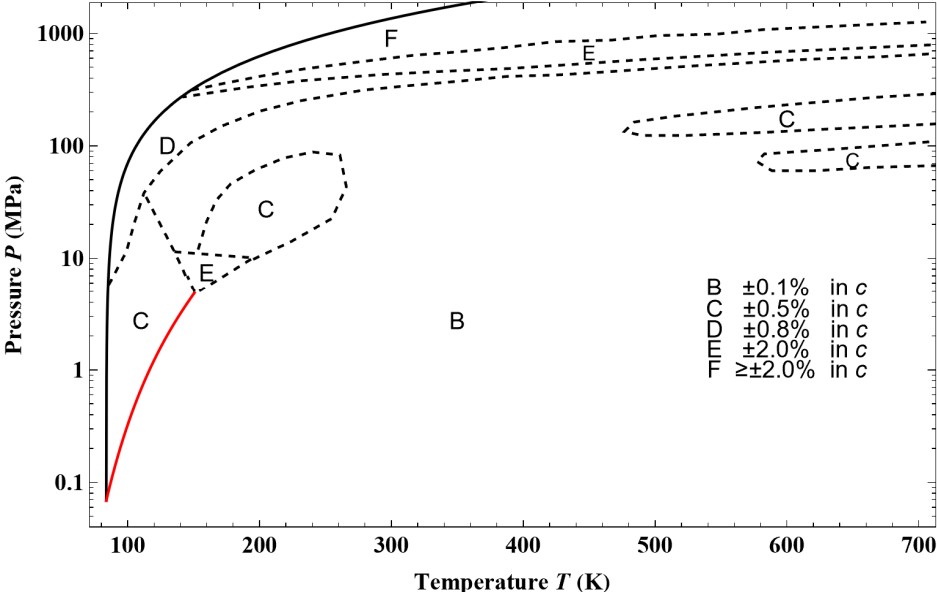

**Figure 39.** Tolerance diagram for sound speed calculated from Equation (69) with the use of Equation (45) for determining densities as a function of pressure. The red curve corresponds to the saturated vapor pressure curve and the black one to the melting line.

We did not plot a tolerance diagram for $C_V$ as it will not produce different information from that contained in Figure 11. Moreover, the relative error on $C_V$ between the TSW model and the present one is everywhere far inside the errors shown in Figure 44 of [4].

Comparisons with the data of L'Air Liquide (1710 data points, [5]) allow completing the tolerance diagrams in the temperature range of 700 to 1100 K. These uncertainties for calculated densities $\rho$, isobaric heat capacities $C_P$, and isochoric heat capacities $C_V$ are illustrated in the tolerance diagrams (Figures 40–42). Here again, in order to extend the comparison with

the tolerance diagrams in [4], we have retained the same notations with their corresponding meanings. The new tolerance intervals are entered directly without using new letters of the alphabet.

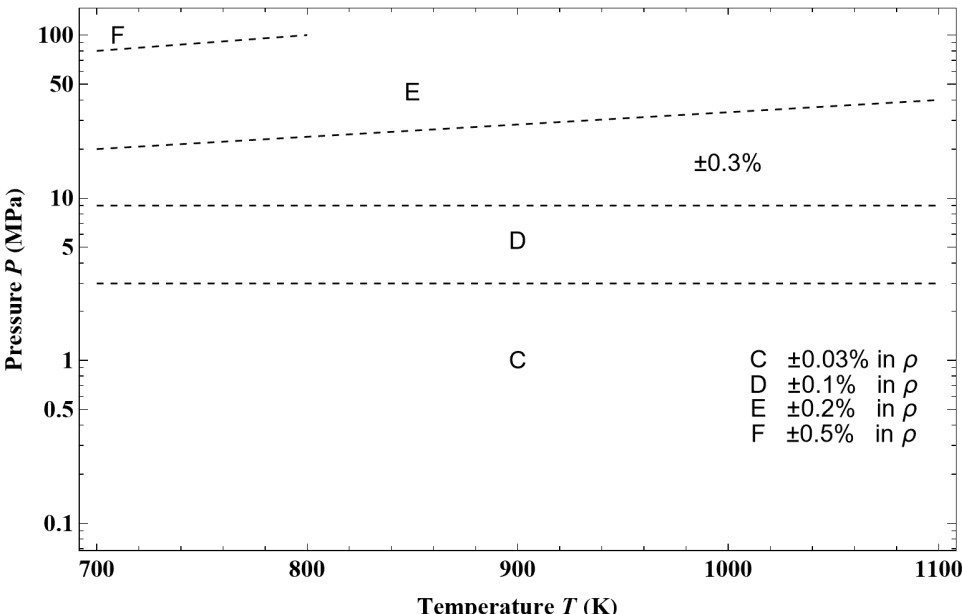

**Figure 40.** Tolerance diagram for densities calculated from inversion of Equation (45) in the temperature range of 700 to 1100 K.

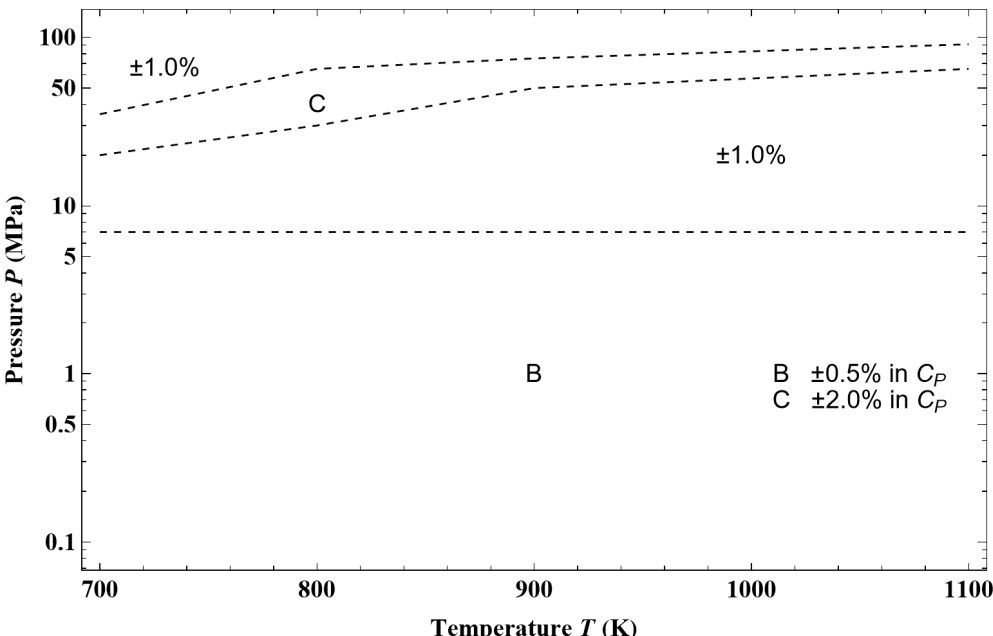

**Figure 41.** Tolerance diagram for isobaric heat capacities calculated from Equation (68) with the use of Equation (45) for determining densities as a function of pressure in the temperature range of 700 to 1100 K.

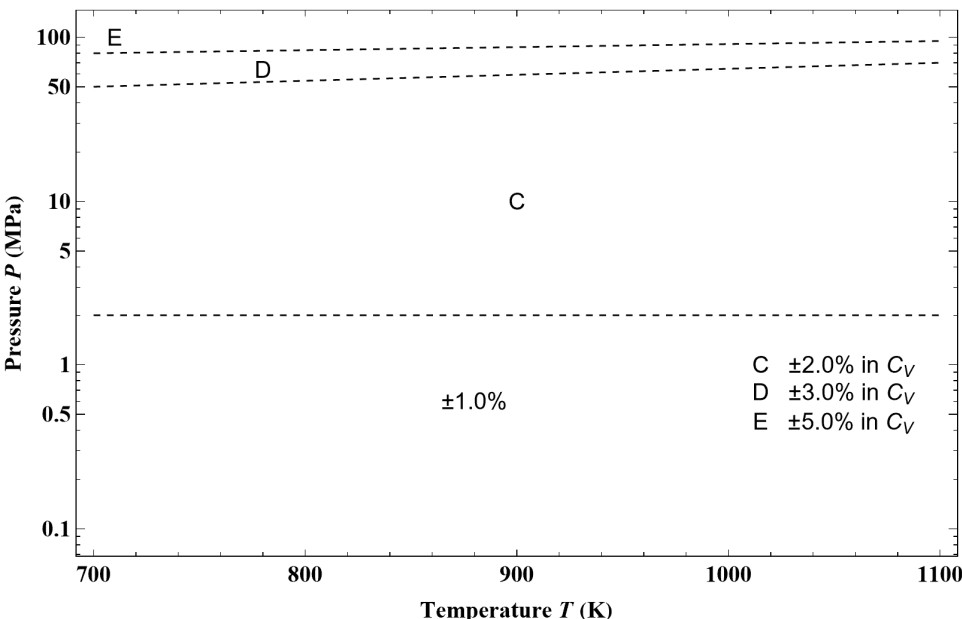

**Figure 42.** Tolerance diagram for isochoric heat capacities calculated from Equation (1) with the use of Equation (45) for determining densities as a function of pressure in the temperature range of 700 to 1100 K.

## 6. Tait's Equation of State to Describe the Liquid Phase

As a result of Tait's work to understand and analyze the ocean temperature measurements [33] from the first global oceanographic campaign of the H.M.S. Challenger [34], it was shown that most liquids can be described empirically by a so-called Tait equation of state [35]. The most commonly used form of equation is the so-called Tait–Tammann equation, defined from the isothermal mixed elasticity modulus in volume, which we will write here using the notations of [35] (in Chapter 3):

$$V(T, P) = V_{\sigma l}(T) - \widetilde{J}(T) \ln\left(\frac{P + \widetilde{\Pi}(T)}{P_{\text{sat}}(T) + \widetilde{\Pi}(T)}\right) \tag{71}$$

where $V$ represents the specific volume of the liquid, $V_{\sigma l}$ the specific volume of liquid along the coexistence curve deduced from Equation (A24), and $P_{\text{sat}}$ is given by Equation (A23). The two Tait–Tammann parameters $\widetilde{\Pi}$ and $\widetilde{J}$ have the following expression in the case of liquid argon:

$$\begin{aligned}
\widetilde{\Pi}(T) = -P_c + 7.5538(1 - T_r)^{1.0728} \quad & \exp\left(-\left|\frac{T_r - 0.77453}{0.22816}\right|^{39}\right) \\
& \times \left(1 + 176\exp\left(-|T_r - 0.5614|^{1.1276}\right)^{2.3856}\right)
\end{aligned} \tag{72}$$

$$\begin{aligned}
\widetilde{J}(T) = 1.7549\, T_r^6\, \exp\left(-\left|\frac{T_r - 0.56277}{0.1147}\right|^{1.0685}\right) + 0.064031\, \exp\left(-\left|\frac{T_r - 0.98098}{0.17611}\right|^{1.2217}\right) \\
+ 0.01\exp\left(-\left|\frac{T_r - 0.577}{0.061826}\right|^{1.2}\right)
\end{aligned} \tag{73}$$

where $T_r = T/T_c$ with $T_c = T_{c,\text{non-ext. formulation}}$ = 151.396 K and $P_c = P_{c,\text{non-ext. formulation}}$ = 49.9684 bar. Equation (72) gives $\widetilde{\Pi}$ in bar, and Equation (73) gives $\widetilde{J}$ in cm$^3$/g.

It can be immediately noticed that Equations (72) and (73) verify the necessary conditions for thermodynamic stability such that $\widetilde{\Pi}(T_c) = -P_c$ and $\widetilde{J} \geq 0$ (see Section 3.3 of [35]). Equation (72) implies that the corresponding temperature for which $\widetilde{\Pi} = 0$ for argon is $T_\infty$ = 137.584 K. This temperature represents the transition between liquid- and gas-like behavior. Therefore, for temperatures $T_\infty < T < T_c$, Equation (71) shows an asymptote that lies in the biphasic

region for $T < 150.5$ K, but beyond that, it appears in the liquid phase. In other words, it is not advisable to use the Tait–Tammann approximation for temperatures above 150.5 K.

Figure 43 shows that the variations in Tait–Tammann parameters for argon are very similar to those of water (see Figure 3.4 of [35]), except for $\widetilde{J}$ near the critical point, which reaches a maximum here. For many liquids, it is generally assumed that $\widetilde{J}$ is constant. Figure 43b shows that this is approximately the case between 90 and 110 K for liquid argon, so the average value is $\widetilde{J}_{\text{average}} \approx 0.07305$ cm$^3$/g. The closeness of the variations in the Tait–Tammann parameters to those of water still implies that the Ginell parameters [36] will have the same variations along an isotherm or isobar as those shown in Figures 3.7 and 3.8 of [35], and consequently, the same picture of the structure of liquid argon in terms of aggregates can be drawn.

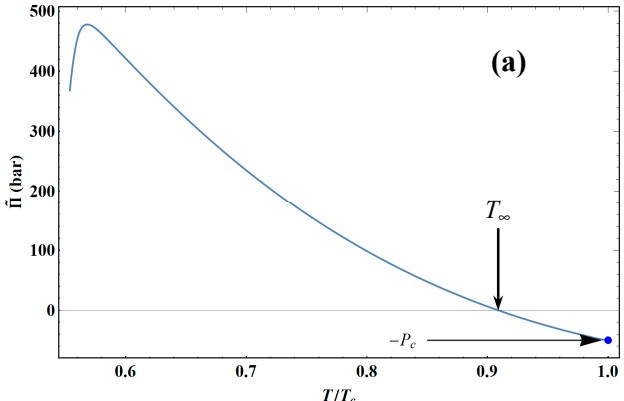
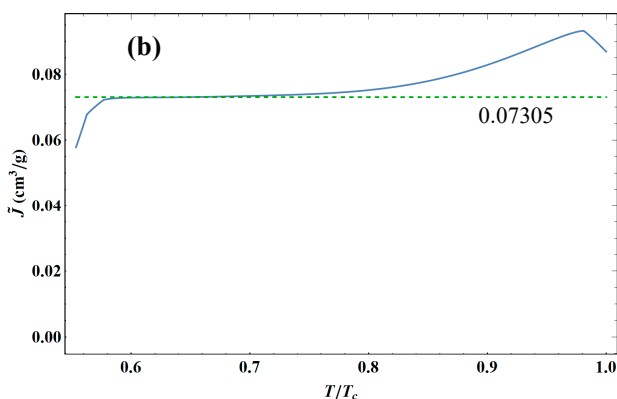

**Figure 43.** Variations in Tait–Tammann parameters for liquid argon between $T_t$ and $T_c = 151.396$ K. (**a**) Equation (72); (**b**) Equation (73). The green dotted curve represents the average value of $\widetilde{J}$ between 90 and 110 K.

In Figure 44, from an example of a given isotherm, it can be observed that the isothermal mixed elasticity modulus of liquid argon has an appreciable curvature over a pressure range of 1000 bar. Therefore, the Tait–Tammann equation over this pressure range can only be a rough approximation. If one wants to obtain a representation that fits into the tolerance diagram in Figure 17, the pressure range must be reduced. In the case of liquid argon, the variation of the isothermal mixed elasticity modulus as a function of pressure can be considered linear for a pressure range globally below 200 bar. Above 200 bar, it was shown in Chapter 4 of [35] that it is the adiabatic modulus of elasticity that satisfies the linearity with pressure, and therefore, a modified Tait equation must be used to describe the liquid and supercritical states of argon.

Figure 45a shows the deviation obtained between the Tait–Tammann equation of state given by Equations (71)–(73) and the present non-extensive formulation given by the inversion of Equation (45). It can be observed that the deviation essentially increases for temperatures above 140 K, so it diverges from the prescription given by the tolerance diagram in Figure 17, which provides a tolerance of $\pm 0.2\%$ for subregion E. Figure 45b shows that to achieve the tolerance of $\pm 0.2\%$, the pressure range must be reduced between $P_{\text{sat}}$ and 100 bar. However, the tolerance corresponding to subregion E is still slightly exceeded between 146 and 148 K.

Figure 45a shows a larger negative deviation for pressures above 100 bar and temperatures above 130 K. In other words, it seems that the tolerance diagram prescription of $\pm 0.1\%$ for subregion D is not held at high pressures. This is confirmed by Figure 46a, which shows that the tolerance of $\pm 0.1\%$ for subregion D is slightly exceeded for temperatures above 130 K and pressures above 160 bar. On the other hand, Figure 46b shows that the tolerance diagram prescription of $\pm 0.03\%$ for subregion C is well achieved for the pressure range of $P_{\text{sat}}$ to 200 bar. Moreover, Figure 47 shows that the application of the Tait–Tammann equation of state up to 300 bar in subregion C leads to a deviation of $\pm 0.05\%$ between $T_t$ and 110 K and then becomes very slightly higher between 110 and 115 K for pressures above 260 bar. Even if the deviation is slightly larger than the tolerance in Figure 17, the use of the Tait–Tammann

equation of state between $T_t$ and 115 K can be considered a "good" approximation between $P_{sat}$ and 300 bar.

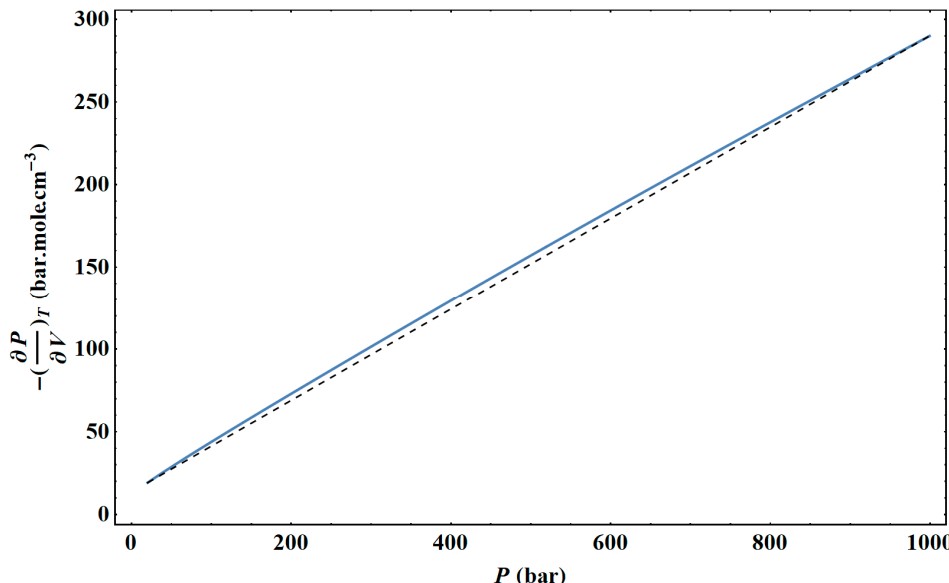

**Figure 44.** Evolution of the isothermal mixed elasticity modulus along the isotherm at $T = 130$ K, calculated from the present non-extensive formulation (i.e., blue curve). The black dotted line simply represents a straight line connecting the first and last points of the isotherm.

Jain et al. [37] also developed a Tait–Tammann equation to describe the specific volume of liquid argon between 90 and 148 K for a pressure range of 300 bar. It is therefore instructive to compare the evolution of the Tait–Tammann parameters with Equations (71)–(73). The form proposed by Jain et al. is such that their reference volume is no longer that of the liquid on the coexistence curve $V_{\sigma l}$ but rather the volume corresponding to zero pressure, noted as $V_0$. Under this condition, the pressure $P_{sat}$ is now replaced by the zero value in Equation (71). With the notations of Jain et al., $\widetilde{J}$ is equivalent to $cV_0$, where $c$ is a constant and $\widetilde{\Pi}$ to $B$.

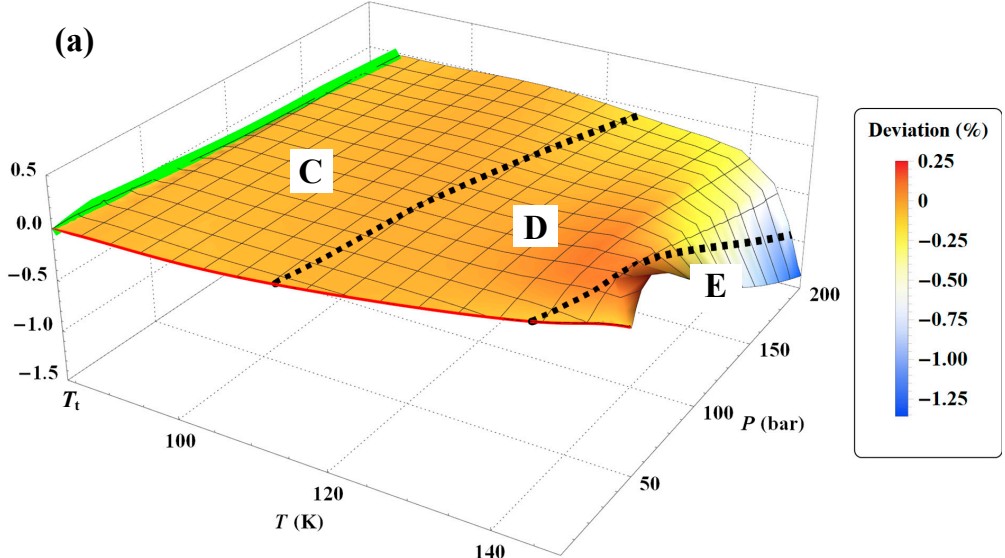

**Figure 45.** *Cont.*

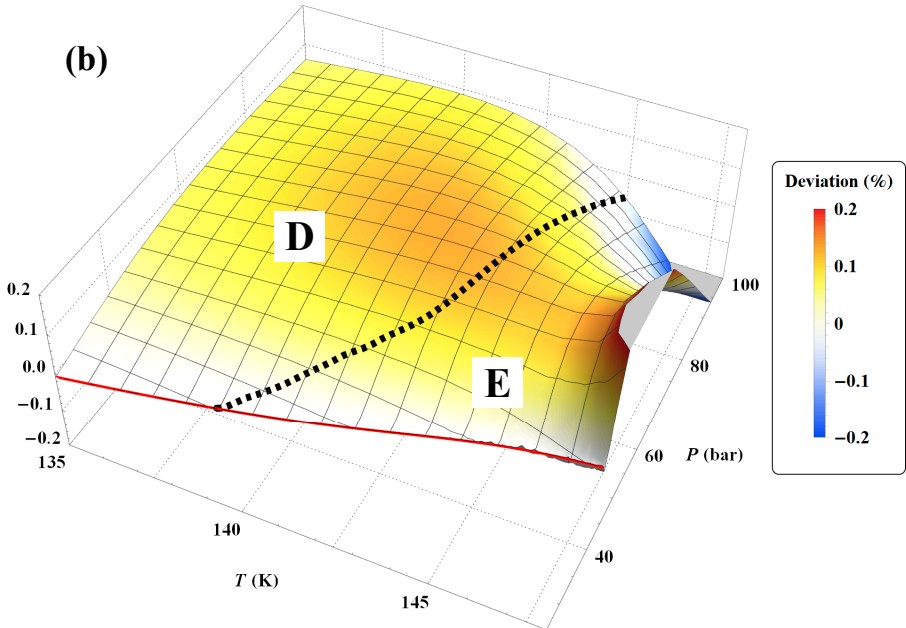

**Figure 45.** Percentage deviations of the specific volume $100(V_{\text{non-ext. formulation}} - V_{\text{Eq. (71)}})/V_{\text{non-ext. formulation}}$ between the Tait–Tammann equation of state Equation (71) and the present non-extensive formulation from the inversion of Equation (45). The thick green curve represents the melting line, while the red curve represents the saturated vapor pressure curve. The black dashed curves are those in Figure 17 that represent the separation between subregions C, D, and E in the liquid phase. (**a**) Pressure range of $P_{\text{sat}}$ to 200 bar and temperature range of $T_t$ to 148 K; (**b**) pressure range of $P_{\text{sat}}$ to 100 bar and temperature range of 135 to 148 K.

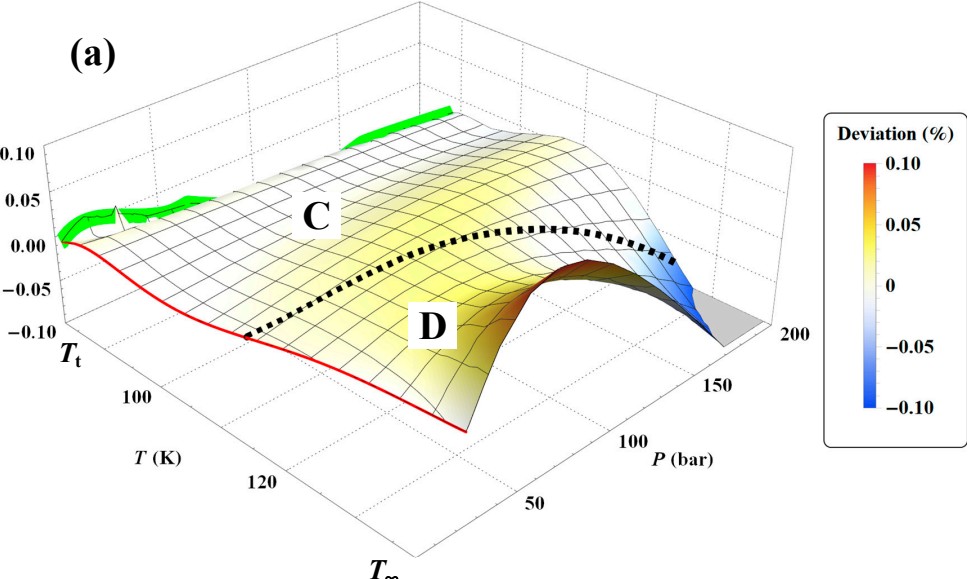

**Figure 46.** *Cont.*

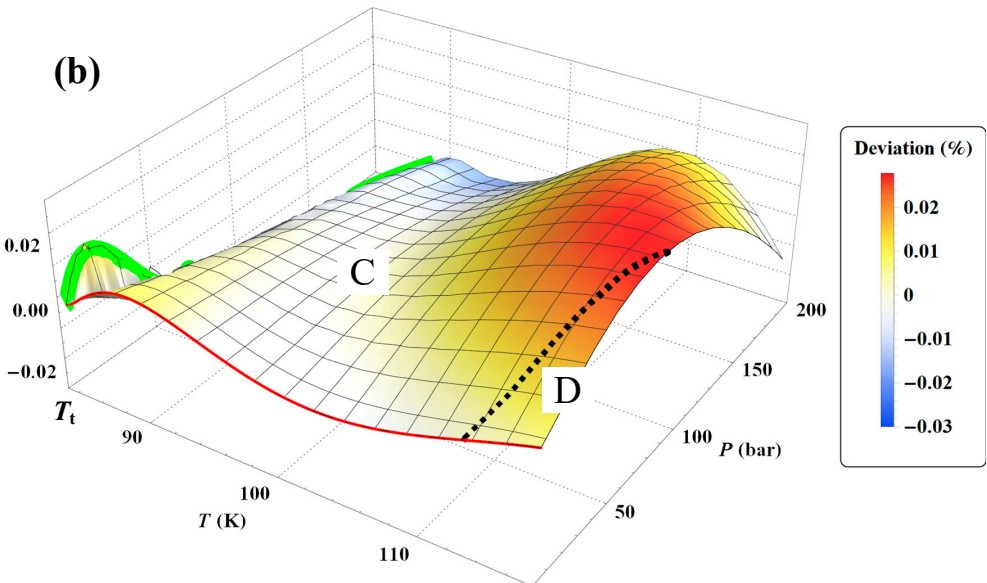

**Figure 46.** Percentage deviations of the specific volume $100(V_{\text{non-ext. formulation}} - V_{\text{Eq. (71)}})/V_{\text{non-ext. formulation}}$ between the Tait–Tammann equation of state Equation (71) and the present non-extensive formulation from the inversion of Equation (45), in the pressure range of $P_{\text{sat}}$ to 200 bar. The thick green curve represents the melting line, while the red curve represents the saturated vapor pressure curve. The black dashed curves are those in Figure 17 that represent the separation between subregions C and D in the liquid phase. (**a**) Temperature range $T_t$ to $T_\infty$; (**b**) temperature range of $T_t$ to 115 K.

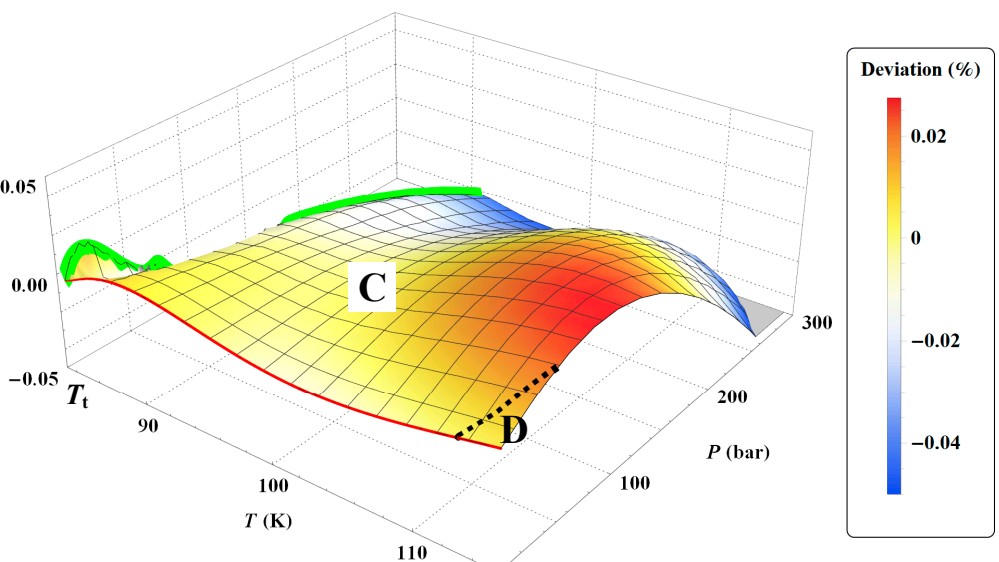

**Figure 47.** Percentage deviations of the specific volume $100(V_{\text{non-ext. formulation}} - V_{\text{Eq. (71)}})/V_{\text{non-ext. formulation}}$ between the Tait–Tammann equation of state Equation (71) and the present non-extensive formulation from the inversion of Equation (45), in the pressure range of $P_{\text{sat}}$ to 300 bar and temperature range of $T_t$ to 115 K. The thick green curve represents the melting line, while the red curve represents the saturated vapor pressure curve. The black dashed curves are those in Figure 17 that represent the separation between subregions C and D in the liquid phase.

To cover the entire temperature range between 90 and 148 K, Jain et al. [37] performed a piecewise determination of their equation of state as they explained:

"in two overlapping temperature ranges (i) 90 to 130 K and (ii) 127 to 148 K. [...] The differences in the calculated values of $V_0$, and $B$ in these overlapping ranges for both liquids are less than 1 in 4000".

Figure 48 shows the comparison between the two specific reference volumes of the Tait–Tammann equation of state for the two determinations of Jain et al. [37] and Equations (71)–(73). It can be observed that the specific volume at zero pressure $V_0$ is smaller and then becomes the same as the liquid coexistence specific volume when the temperature is lower than 100 K, which is not physically acceptable. However, the saturated vapor pressure $P_{sat}$ is not negligible for temperatures between 90 and 100 K; therefore, the equation of state of Jain et al. is expected to deviate strongly from the TSW model or the present non-extensive formulation in the vicinity of the saturated vapor pressure curve. However, this is a voluntary choice made by Jain et al., as they have explained below:

"At sufficiently low saturation pressures, the observed volume can be taken equal to $V_0$, and the problem of finding a suitable value of $B$ to represent the experimental results presents little difficulty".

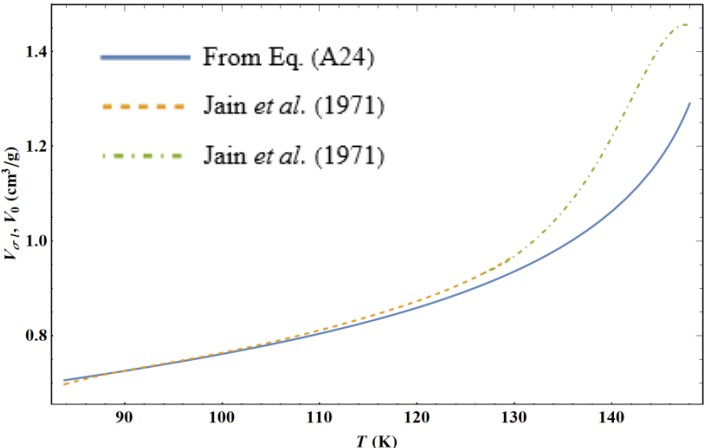

**Figure 48.** Evolution of the reference volume for the Tait–Tammann equation of state as a function of temperature in the range of $T_t$ to 148 K: the dashed curves correspond to the two determinations of Jain et al. [37] and the solid blue curve to $V_{\sigma l}$ deduced from Equation (A24).

Figure 49 shows significant differences between the parameters $B$ and $\widetilde{\Pi}$, and $cV_0$ and $\widetilde{J}$. However, the variations show some similarities, like the maximum of $cV_0$. Indeed, Figure 49b shows that Equation (73) also has a maximum but for a slightly higher temperature than for Jain et al. [37].

Jain et al. [37] developed their Tait–Tammann equation of state to represent the experimental data of van Itterbeek et al. (11 isotherms that range from 90.15 to 148.25 K, TABLE I of [38]). These experimental data were assigned by Tegeler et al. [4] to Group 3 and therefore do not fit into the tolerance diagram of Figure 17. Thus, the equation of state of Jain et al. is expected to represent the raw data of van Itterbeek et al. much more accurately than the non-extensive formulation of the present model or the Tait–Tammann equation of state, i.e., Equations (71)–(73).

Figure 50 shows a comparison of the different models to represent the raw van Itterbeek et al. data corresponding to subregion C in Figure 17. It can be first observed that Equation (71) and the non-extensive formulation of the present model are not much different according to the deviation in Figure 47. It appears that the deviation of the model of Jain et al. [37] is comparable to the other models for pressures above 100 bar, which is quite surprising. It is even more surprising to note that the model of Jain et al. appears entirely shifted by $-0.15\%$ on the lowest isotherm at 90.15 K. However, the model of Jain et al. better represents data that is close to the saturated vapor pressure curve. This is consistent with the fact that these data were not considered by Tegeler et al. to determine the liquid density on the coexistence curve.

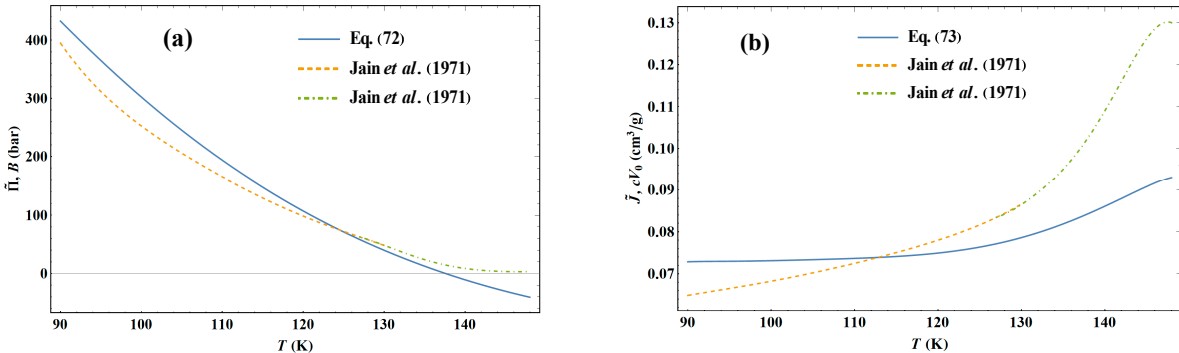

**Figure 49.** Variations in Tait–Tammann parameters for liquid argon between 90 and 148 K. (**a**) $B$ from the two determinations of Jain et al. [37] and $\widetilde{\Pi}$ from Equation (72); (**b**) $cV_0$ from the two determinations of Jain et al. [37] and $\widetilde{J}$ from Equation (73).

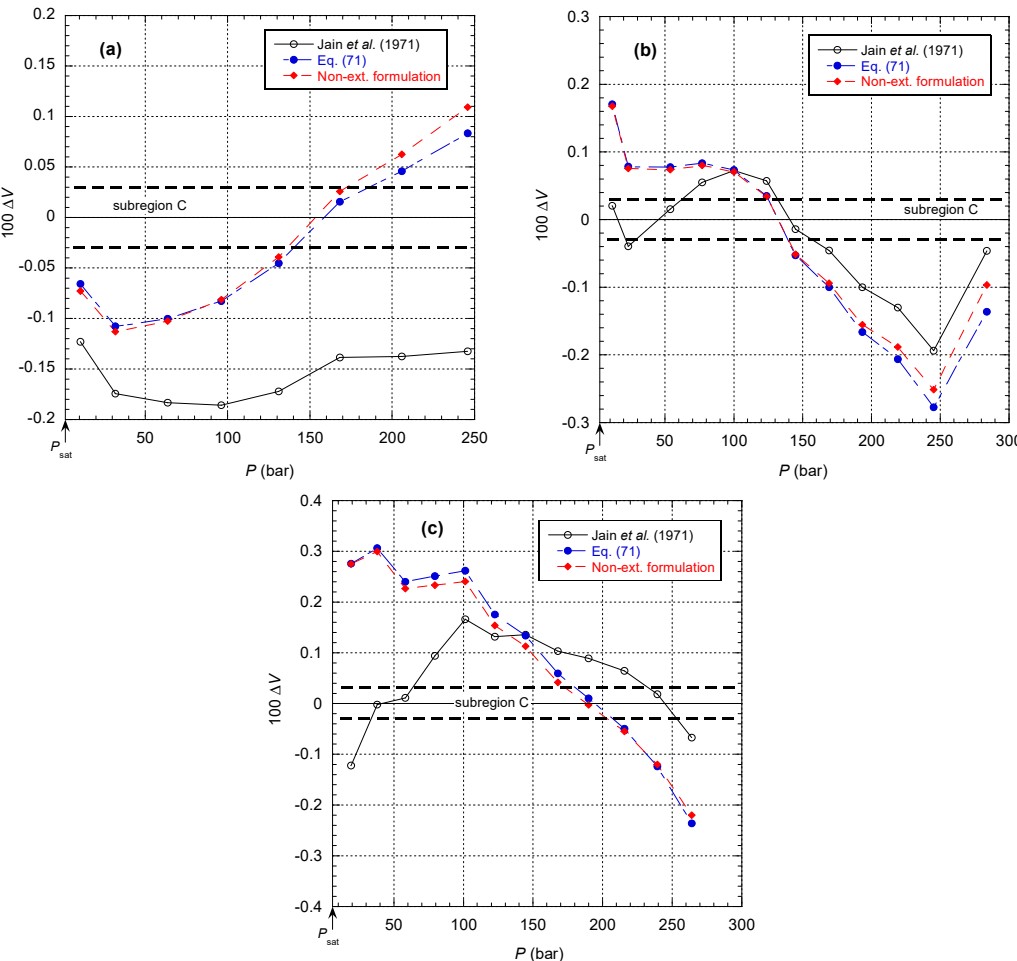

**Figure 50.** Percentage deviations of the specific volume $\Delta V = (V_{\text{van Itterbeek}} - V_{\text{calc}})/V_{\text{van Itterbeek}}$ between the raw data of van Itterbeek et al. (TABLE I of [38]) and the different models (i.e., Jain et al. [37], equation 71 and the present non-extensive formulation) for the three isotherms corresponding to subregion C in Figure 17: (**a**) 90.15 K; (**b**) 96.99 K; (**c**) 108.18 K.

Figure 51 shows the comparison of the different models for the isotherms corresponding to subregion D in Figure 17. It can be first observed that there is a larger deviation between the non-extensive formulation and Equation (71) beyond 150 bar, which is consistent with Figure 46a. Then, it can be seen that the model of Jain et al. [37] again has difficulties reproducing

the data near the saturated vapor pressure curve. This can only be explained by an incorrect extrapolation to determine their function $V_0(T)$, if we refer to their explanation below:

> "However, at high saturation pressures, both $V_0$, and $B$ have to be suitably chosen. The trial value of $V_0$ is first obtained by extrapolation of the $V_0$, against $T$ graph from the low temperature side".

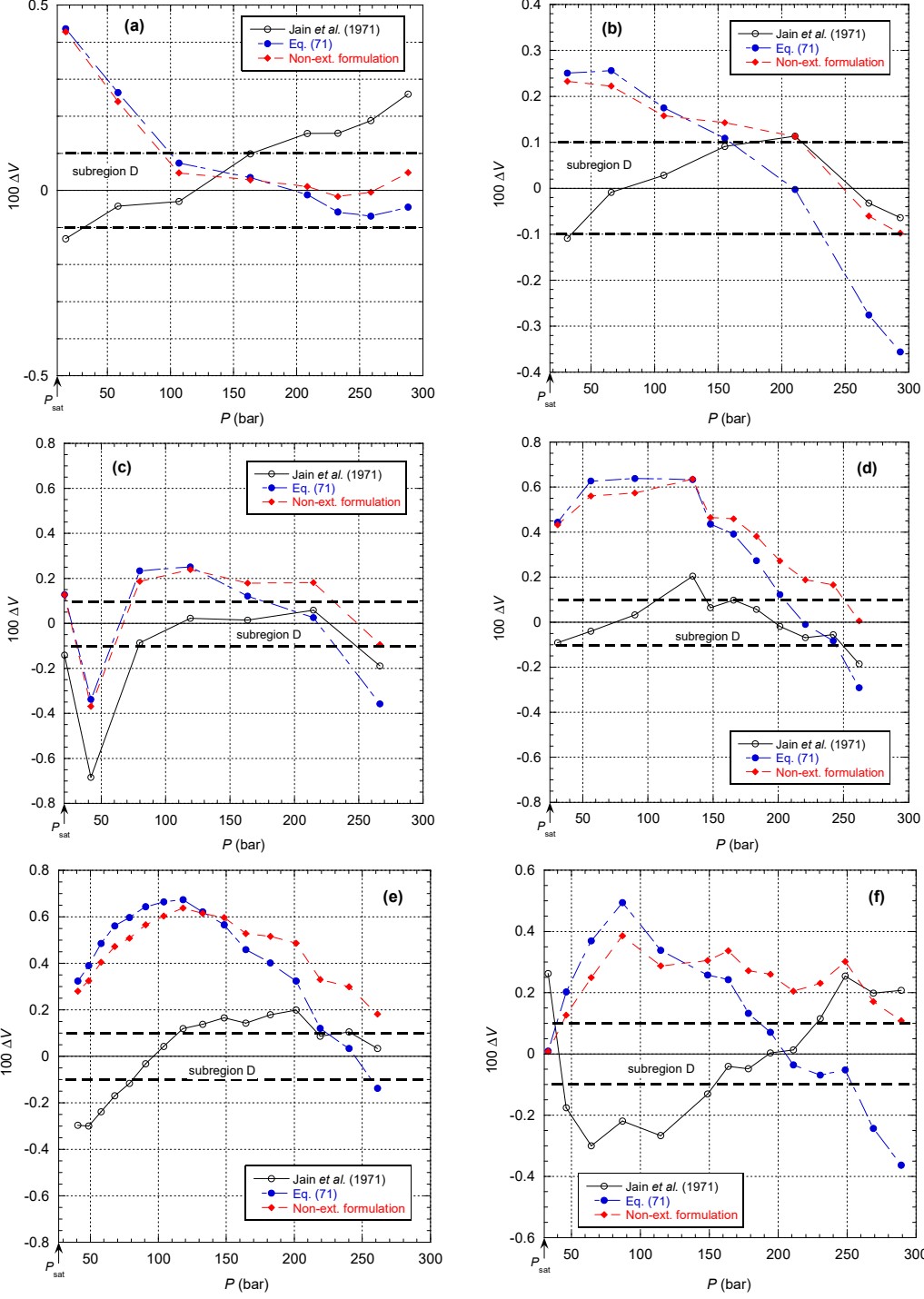

**Figure 51.** Percentage deviations of the specific volume $\Delta V = (V_{\text{van Itterbeek}} - V_{\text{calc}})/V_{\text{van Itterbeek}}$ between the raw data of van Itterbeek et al. (TABLE I of [38]) and the different models (i.e., Jain et al. [37], Equation (71) and the present non-extensive formulation) for the six isotherms corresponding to subregion D in Figure 17: (**a**) 117.10 K; (**b**) 127.05 K; (**c**) 130.85 K; (**d**) 134.40 K; (**e**) 136.02 K; (**f**) 138.98 K.

Figure 52 shows the comparison of the different models for the isotherms corresponding to subregion E in Figure 17. It can be observed here that the deviation of the Jain et al. model from the data is greatest in the vicinity of the saturated vapor pressure curve, and the deviation is systematical in the same direction. On the other hand, this model allows data to be taken into account with the tolerance prescribed for subregion E for pressures higher than 120 bar. It can be seen that the opposite process occurs to a smaller extent for Equation (71). This is due to the curvature of the mixed elastic modulus becoming stronger and stronger, and therefore the approximation of the Tait–Tammann equation only makes sense for a smaller and smaller range of pressures. Equation (71) allows a satisfactory representation of data up to 150 bar, while the model of Jain et al. is able to reproduce data between 150 and 300 bar. The two descriptions cannot be connected except by making the Tait–Tammann parameters depend on the pressure, but under these conditions, it is preferable to use the non-extensive formulation.

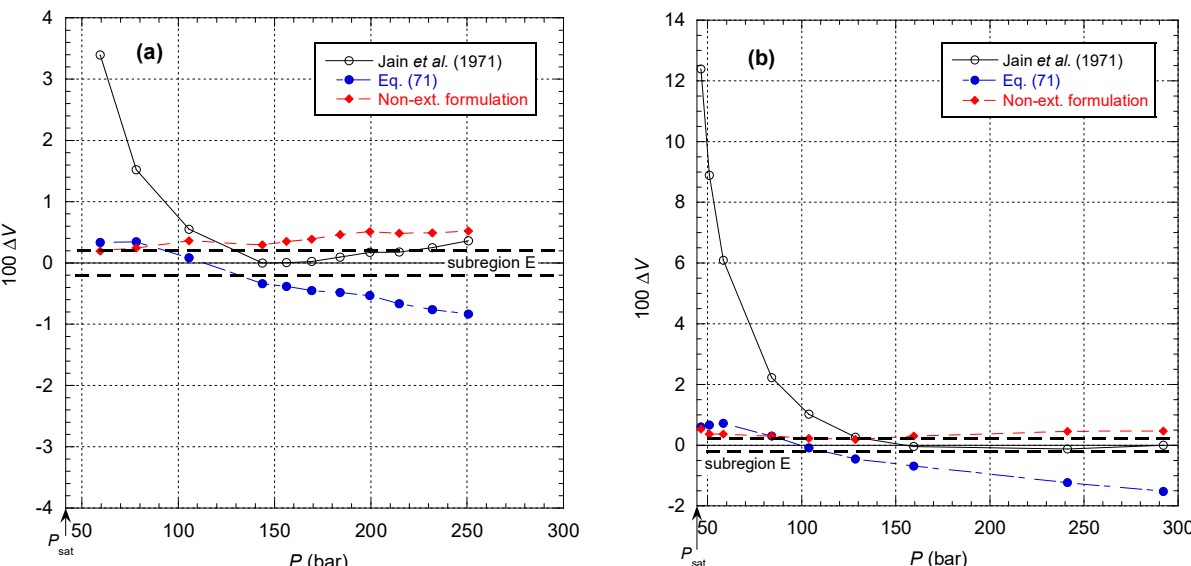

**Figure 52.** Percentage deviations of the specific volume $\Delta V = (V_{\text{van Itterbeek}} - V_{\text{calc}})/V_{\text{van Itterbeek}}$ between the raw data of van Itterbeek et al. (TABLE I of [38]) and the different models (i.e., Jain et al. [37], equation 71 and the present non-extensive formulation) for the two isotherms corresponding to subregion E in Figure 17: (**a**) 146.63 K; (**b**) 148.25 K.

In this section, it has been shown that the Tait–Tammann equation of state can be a simple alternative to describe the specific volume of liquid argon between $T_t$ and 148 K, for pressures varying between $P_{\text{sat}}$ and 300 bar in subregion C, then between $P_{\text{sat}}$ and 200 bar in subregion D, and finally between $P_{\text{sat}}$ and 100 bar for subregion E. These pressure ranges are sufficient for a very large number of applications.

## 7. Conclusions

A new equation of state for argon was developed, which can be written in the form of a fundamental equation explicit in the reduced Helmholtz free energy. This equation was derived from the measured quantities $C_V(\rho, T)$ and $P(\rho, T)$. It is valid for the whole fluid region (single-phase and coexistence states) from the melting line to 2300 K and for pressures up to 50 GPa. The formulation is based on data from NIST (or equivalently on the calculated values from the TSW model) and calculated values from the model of Ronchi [7].

This new approach, using mainly power laws with density-dependent exponents, involves much fewer coefficients than the TSW model and, more importantly, eliminates the very small oscillations introduced by a polynomial description. This leads to a more physical description of the thermodynamic properties. On the other hand, the reduction in the number of terms and parameters does not modify the uncertainties of the calculated data in an appreciable way (as shown in the different diagrams of tolerance). However, in an un-

expected way, the present approach, which generates more regular and monotonous expressions, raises greater difficulty for the reversal of certain equations of state due to the highly nonlinear behavior of these expressions.

The new equation of state also shows more physical behavior along isochors when $T$ tends to zero for basic properties, such as the isochoric heat capacity and the compressibility factor. It also shows a more reasonable behavior for the crossing of the coexistence phase. However, it does not correctly describe the properties in the vicinity of the critical point, in the same way as the TSW model does not properly describe the properties in the vicinity of the critical point, with the exception of the saturation curve. However, variations in the isochoric heat capacity in the coexistence phase with the present model show peaks that are qualitatively in agreement with experimental observations, unlike the TSW model, which produces unphysical variations. Comparison of the present model with the data of L'Air Liquide [5], which had not previously been taken into account, shows that this model is consistent with data up to 1100 K and 100 MPa, which allows, regardless of the data of Ronchi [7], the range of NIST data to be extended [6].

In Section 6 and Appendix C, simple expressions are also provided to describe the specific volume of the liquid states of argon between $T_t$ and 148 K in the form of a Tait–Tammann equation of state and some properties of the liquid–vapor coexistence curve. These approximate formulas can advantageously replace the complex, non-extensive formulation of the present model for a large number of applications.

The non-extensive approach developed here shows that metastable states are, by construction, included as an extension of single-phase isochoric heat capacity modeling. As $C_V$ data are generally known for the vast majority of fluids, this new approach can be easily extended to all of them.

**Supplementary Materials:** The following supporting information can be downloaded at: <https://notebookarchive.org/2024-01-8ag2wbe>. Aitken F. and Volino F. "NewEoSArgon" from the Notebook Archive (2024).

**Author Contributions:** Writing—original draft preparation, F.A., A.D. and F.V. All authors have read and agreed to the published version of the manuscript.

**Funding:** This work benefited from the support of the project ZEROUATE under Grant ANR-19-CE24-0013, operated by the French National Research Agency (ANR).

**Data Availability Statement:** Data are contained within the article.

**Acknowledgments:** The authors are grateful to J.-L. Garden (Institut Néel Laboratory, France) for his valuable support.

**Conflicts of Interest:** The authors declare no conflict of interest.

## Appendix A. Expression of the Regular Term of Pressure $P_{reg}$

The regular term of pressure is formed by the difference of two terms that come from the derivative of energy and entropy, respectively:

$$P_{reg}(\rho, T) = P_{Ureg}(\rho, T) - P_{Sreg}(\rho, T) \tag{A1}$$

and

$$Z_{reg}(\rho, T) = \frac{P_{reg}(\rho, T)}{\rho R_A T} = Z_{Ureg}(\rho, T) - Z_{Sreg}(\rho, T) \tag{A2}$$

with

$$
\begin{aligned}
Z_{\text{Ureg}} &= \frac{P_{\text{Ureg}}(\rho,T)}{\rho R_A T} = \rho \left( \frac{\partial \tilde{u}_{\text{reg}}}{\partial \rho} \right)_T = \frac{3}{2}\rho\, n'_{\text{reg}}(\rho) \frac{T_c}{T} \left[ \frac{(T/T_c)^{m(\rho)}-1}{m(\rho)} + \frac{\lambda^{-m(\rho)}}{2-m(\rho)} \Gamma\left( \frac{m(\rho)}{2-m(\rho)}, \left(\frac{\lambda T}{T_c}\right)^{2-m(\rho)} \right) \right] \\
&+ \frac{3}{2} n_{\text{reg}}(\rho) \frac{T_c}{T}\rho\, m'(\rho) \left\{ \frac{1-(T/T_c)^{m(\rho)}}{m(\rho)^2} + \frac{(T/T_c)^{m(\rho)}\ln(T/T_c)}{m(\rho)} + \frac{\lambda^{-m(\rho)}}{2-m(\rho)} \left(\frac{\lambda T}{T_c}\right)^{m(\rho)} \ln\left(\frac{\lambda T}{T_c}\right) \exp\left( -\left(\frac{\lambda T}{T_c}\right)^{\frac{5}{2}-m(\rho)} \right) \right\} \\
&+ \frac{3}{2} n_{\text{reg}}(\rho) \frac{T_c}{T}\rho\, m'(\rho) \frac{\lambda^{-m(\rho)}}{[2-m(\rho)]^2} \Gamma\left( \frac{m(\rho)}{2-m(\rho)}, \left(\frac{\lambda T}{T_c}\right)^{2-m(\rho)} \right) \left\{ 1 - \ln\left(\lambda^{2-m(\rho)}\right) + \frac{2}{2-m(\rho)} \ln\left( \left(\frac{\lambda T}{T_c}\right)^{2-m(\rho)} \right) \right\} \\
&+ 3 n_{\text{reg}}(\rho) \frac{T_c}{T}\rho\, m'(\rho) \frac{\lambda^{-m(\rho)}}{[2-m(\rho)]^3} G_{2\ \ 3}^{3\ \ 0} \left( \left(\frac{\lambda T}{T_c}\right)^{2-m(\rho)} \middle| \begin{array}{ccc} 1, & 1 & \\ 0, & 0, & \frac{m(\rho)}{2-m(\rho)} \end{array} \right)
\end{aligned} \tag{A3}
$$

and

$$
\begin{aligned}
Z_{\text{Sreg}} &= \frac{P_{\text{Sreg}}(\rho,T)}{\rho R_A T} = \rho \left( \frac{\partial \tilde{s}_{\text{reg}}}{\partial \rho} \right)_T = \frac{3}{2}\rho\, n'_{\text{reg}}(\rho) \left[ \frac{(T/T_c)^{m(\rho)-1}-1}{m(\rho)-1} + \frac{\lambda^{1-m(\rho)}}{2-m(\rho)} \Gamma\left( \frac{m(\rho)-1}{2-m(\rho)}, \left(\frac{\lambda T}{T_c}\right)^{2-m(\rho)} \right) \right] \\
&+ \frac{3}{2} n_{\text{reg}}(\rho)\rho\, m'(\rho) \left\{ \frac{1-(T/T_c)^{m(\rho)-1}}{[m(\rho)-1]^2} + \frac{(T/T_c)^{m(\rho)-1}\ln(T/T_c)}{m(\rho)-1} + \frac{\lambda^{1-m(\rho)}}{2-m(\rho)} \left(\frac{\lambda T}{T_c}\right)^{m(\rho)-1} \ln\left(\frac{\lambda T}{T_c}\right) \exp\left( -\left(\frac{\lambda T}{T_c}\right)^{2-m(\rho)} \right) \right\} \\
&+ \frac{3}{2} n_{\text{reg}}(\rho)\rho\, m'(\rho) \frac{\lambda^{1-m(\rho)}}{[2-m(\rho)]^2} \Gamma\left( \frac{m(\rho)-1}{2-m(\rho)}, \left(\frac{\lambda T}{T_c}\right)^{2-m(\rho)} \right) \left\{ 1 - \ln\left(\lambda^{2-m(\rho)}\right) + \frac{1}{2-m(\rho)} \ln\left( \left(\frac{\lambda T}{T_c}\right)^{2-m(\rho)} \right) \right\} \\
&+ \frac{3}{2} n_{\text{reg}}(\rho)\rho\, m'(\rho) \frac{\lambda^{1-m(\rho)}}{[2-m(\rho)]^3} G_{2\ \ 3}^{3\ \ 0} \left( \left(\frac{\lambda T}{T_c}\right)^{2-m(\rho)} \middle| \begin{array}{ccc} 1, & 1 & \\ 0, & 0, & \frac{m(\rho)-1}{2-m(\rho)} \end{array} \right)
\end{aligned} \tag{A4}
$$

where $G_{p\ q}^{m\ n}\left( z \middle| \begin{array}{c} a_1, \cdots, a_p \\ b_1, \ldots, b_q \end{array} \right)$ represents the Meijer G function. It is worth noting that the Meijer functions in $Z_{\text{Ureg}}$ and $Z_{\text{Sreg}}$ are equal to zero when $T \geq T_t$ whatever the value of density.

For calculating some thermodynamics parameters, the first partial derivatives of this pressure term are needed.

The first partial derivatives of $Z_{\text{Ureg}}$ and $Z_{\text{Sreg}}$ with temperature are written below:

$$
\begin{aligned}
\left( \frac{\partial Z_{\text{Ureg}}}{\partial T} \right)_\rho &= \frac{3}{2} \frac{T_c}{T} \frac{\rho}{T} n'_{\text{reg}} \lambda^{-m} \left[ \left(\frac{\lambda T}{T_c}\right)^{m} \left( \frac{m-1}{m} - \exp\left( -\left(\frac{\lambda T}{T_c}\right)^{2-m} \right) \right) + \frac{\lambda^m}{m} + \frac{1}{m-2} \Gamma\left( \frac{m}{2-m}, \left(\frac{\lambda T}{T_c}\right)^{2-m} \right) \right] \\
&+ \frac{3}{2} \frac{T_c}{T} \frac{\rho}{T} n_{\text{reg}} \frac{m'}{m^2} \left\{ -1 + \left(\frac{T}{T_c}\right)^{m} \left[ 1 + m(m-1)\ln\left(\frac{T}{T_c}\right) \right] \right\} \\
&- 3 \frac{T_c}{T} \frac{\rho}{T} n_{\text{reg}} m' \frac{\lambda^{-m}}{(2-m)^3} \Gamma\left( \frac{m}{2-m}, \left(\frac{\lambda T}{T_c}\right)^{2-m} \right) \left\{ \ln\left( \left(\frac{\lambda T}{T_c}\right)^{2-m(\rho)} \right) + \frac{1}{2}\left( 1 + \ln(\lambda^{2-m}) \right)(m-2) \right\} \\
&+ \frac{3}{2} \frac{T_c}{T} \frac{\rho}{T} n_{\text{reg}} m' \frac{\lambda^{-m}}{(2-m)} \left(\frac{\lambda T}{T_c}\right)^2 \exp\left( -\left(\frac{\lambda T}{T_c}\right)^{2-m} \right) \\
&\qquad \times \left\{ \left(\frac{\lambda T}{T_c}\right)^{m-2} \left[ \ln(\lambda^{2-m}) - \ln\left( \left(\frac{\lambda T}{T_c}\right)^2 \right) \right] - \ln\left( \left(\frac{\lambda T}{T_c}\right)^{2-m} \right) \left( 1 + \left(\frac{\lambda T}{T_c}\right)^{m-2} \frac{m-1}{m-2} \right) \right\} \\
&+ 3 \frac{T_c}{T} \frac{\rho}{T} n_{\text{reg}} m' \frac{\lambda^{-m}}{(2-m)^3} \left\{ (m-2) G_{1\ \ 2}^{2\ \ 0} \left( \left(\frac{\lambda T}{T_c}\right)^{2-m} \middle| \begin{array}{cc} 1 & \\ 0, & \frac{m}{2-m} \end{array} \right) - G_{2\ \ 3}^{3\ \ 0} \left( \left(\frac{\lambda T}{T_c}\right)^{2-m} \middle| \begin{array}{ccc} 1, & 1 & \\ 0, & 0, & \frac{m}{2-m} \end{array} \right) \right\}
\end{aligned} \tag{A5}
$$

and

$$
\begin{aligned}
\left( \frac{\partial Z_{\text{Sreg}}}{\partial T} \right)_\rho &= \frac{3}{2} \frac{\rho}{T} n'_{\text{reg}} \left(\frac{T}{T_c}\right)^{m-1} \left[ 1 - \exp\left( -\left(\frac{\lambda T}{T_c}\right)^{2-m} \right) \right] + \frac{3}{2} \frac{\rho}{T} n_{\text{reg}} m' \left(\frac{T}{T_c}\right)^{m-1} \ln\left(\frac{T}{T_c}\right) \\
&+ \frac{3}{2} \frac{\rho}{T} n_{\text{reg}} m' \frac{\lambda^{1-m}}{(m-2)^2} \left\{ \Gamma\left( \frac{1-m}{m-2}, \left(\frac{\lambda T}{T_c}\right)^{2-m} \right) - G_{1\ \ 2}^{2\ \ 0} \left( \left(\frac{\lambda T}{T_c}\right)^{2-m} \middle| \begin{array}{cc} 1 & \\ 0, & \frac{1-m}{m-2} \end{array} \right) \right\} \\
&+ \frac{3}{2} \frac{\rho}{T} n_{\text{reg}} m' \frac{\lambda^{1-m}}{(m-2)} \left(\frac{\lambda T}{T_c}\right)^{2-m} \exp\left( -\left(\frac{\lambda T}{T_c}\right)^{2-m} \right) \left\{ \left( 1 + \left(\frac{\lambda T}{T_c}\right)^{2-m} \right) \ln\left( \left(\frac{\lambda T}{T_c}\right)^{2-m} \right) - \ln(\lambda^{2-m}) \right\}
\end{aligned} \tag{A6}
$$

The first partial derivatives of $Z_{\text{Ureg}}$ and $Z_{\text{Sreg}}$ with density are written hereafter:

$$\left(\frac{\partial Z_{\text{Ureg}}}{\partial \rho}\right)_T = \frac{3}{2}\frac{T_c}{T}\left(n'_{\text{reg}} + \rho\, n''_{\text{reg}}\right)\left[\frac{\lambda^{-m}}{2-m}\Gamma\left(\frac{m}{2-m}, \left(\frac{\lambda T}{T_c}\right)^{2-m}\right) - m^{-1}\left(1 - \left(\frac{T}{T_c}\right)^m\right)\right]$$

$$+ \frac{3}{2}\frac{T_c}{T}\rho\, n_{\text{reg}}\frac{m'^2}{m^3}\left\{-2 + 2\left(\frac{T}{T_c}\right)^m\left[1 - \ln\left(\left(\frac{T}{T_c}\right)^m\right) + \frac{1}{2}\ln\left(\left(\frac{T}{T_c}\right)^m\right)^2\right]\right\}$$

$$+ \frac{3}{2}\frac{T_c}{T}\rho\, n'_{\text{reg}}\, m'\frac{\lambda^{-m}}{(m-2)^2}\Gamma\left(\frac{m}{2-m}, \left(\frac{\lambda T}{T_c}\right)^{2-m}\right)\left(1 + \ln(\lambda^{m-2})\right)$$

$$+ \frac{3}{2}\frac{T_c}{T}\left(n_{\text{reg}}\, m' + \rho\, n_{\text{reg}}\, m'' + 2\rho\, n'_{\text{reg}}\, m'\right)\left[m^{-2}\left(1 - \left(\frac{T}{T_c}\right)^m\right) + m^{-1}\left(\frac{T}{T_c}\right)^m\ln\left(\frac{T}{T_c}\right)\right]$$

$$- \frac{3}{2}\frac{T_c}{T}\left(n_{\text{reg}}\, m' + \rho\, n_{\text{reg}}\, m'' + \rho\, n'_{\text{reg}}\, m'\right)\frac{\lambda^{-m}}{(m-2)^3}\left\{\Gamma\left(\frac{m}{2-m}, \left(\frac{\lambda T}{T_c}\right)^{2-m}\right)\left[2\ln\left(\left(\frac{\lambda T}{T_c}\right)^{2-m}\right) - (m-2)\left(1 + \ln(\lambda^{m-2})\right)\right]\right.$$

$$\left. + (m-2)^2\left(\frac{\lambda T}{T_c}\right)^m\ln\left(\frac{\lambda T}{T_c}\right)\exp\left(-\left(\frac{\lambda T}{T_c}\right)^{2-m}\right) + 2G\begin{smallmatrix}3&&0\\2&&3\end{smallmatrix}\left(\left(\frac{\lambda T}{T_c}\right)^{2-m}\middle|\begin{smallmatrix}1,&&1\\0,&&0,&\frac{m}{2-m}\end{smallmatrix}\right)\right\}$$

$$+ \frac{3}{2}\frac{T_c}{T}\rho\, m'\left[(2-m)\, n'_{\text{reg}} + n_{\text{reg}}\, m'\left(1 + 2\ln\left(\frac{T}{T_c}\right) + \ln(\lambda^m)\right)\right]\frac{\lambda^{-m}}{(m-2)^4}$$

$$\times\left\{(2-m)\left(\frac{\lambda T}{T_c}\right)^m\exp\left(-\left(\frac{\lambda T}{T_c}\right)^{2-m}\right)\ln\left(\left(\frac{\lambda T}{T_c}\right)^{2-m}\right)\right.$$

$$\left. + 2\Gamma\left(\frac{m}{2-m}, \left(\frac{\lambda T}{T_c}\right)^{2-m}\right)\ln\left(\left(\frac{\lambda T}{T_c}\right)^{2-m}\right) + 2G\begin{smallmatrix}3&&0\\2&&3\end{smallmatrix}\left(\left(\frac{\lambda T}{T_c}\right)^{2-m}\middle|\begin{smallmatrix}1,&&1\\0,&&0,&\frac{m}{2-m}\end{smallmatrix}\right)\right\} \tag{A7}$$

$$- \frac{3}{2}\frac{T_c}{T}\rho\, n_{\text{reg}}\, m'^2\frac{\lambda^{-m}}{(m-2)^3}\left\{\Gamma\left(\frac{m}{2-m}, \left(\frac{\lambda T}{T_c}\right)^{2-m}\right)\left[2 + (2 + \ln(\lambda^{m-2}))\left(2\ln\left(\frac{T}{T_c}\right) + \ln(\lambda^m)\right)\right]\right.$$

$$\left. + \left(\frac{\lambda T}{T_c}\right)^m\exp\left(-\left(\frac{\lambda T}{T_c}\right)^{2-m}\right)\ln\left(\left(\frac{\lambda T}{T_c}\right)^{2-m}\right)\left[1 + \ln\left(\left(\frac{\lambda T}{T_c}\right)^{2-m}\right) + \left(\frac{\lambda T}{T_c}\right)^{2-m}\ln\left(\left(\frac{\lambda T}{T_c}\right)^{2-m}\right)\right]\right\}$$

$$- 3\frac{T_c}{T}\rho\, n_{\text{reg}}\, m'^2\frac{\lambda^{-m}}{(m-2)^5}\left\{(2-m)\left(\frac{\lambda T}{T_c}\right)^{m-2}\ln\left(\left(\frac{\lambda T}{T_c}\right)^{2-m}\right)G\begin{smallmatrix}2&&0\\1&&2\end{smallmatrix}\left(\left(\frac{\lambda T}{T_c}\right)^{2-m}\middle|\begin{smallmatrix}&2\\1,&\frac{m}{2-m}\end{smallmatrix}\right)\right.$$

$$\left. + (2-m)\left(3 + 2\ln\left(\frac{T}{T_c}\right) + \ln(\lambda^m)\right)G\begin{smallmatrix}3&&0\\2&&3\end{smallmatrix}\left(\left(\frac{\lambda T}{T_c}\right)^{2-m}\middle|\begin{smallmatrix}1,&&1\\0,&&0,&\frac{m}{2-m}\end{smallmatrix}\right) + 4G\begin{smallmatrix}4&&0\\3&&4\end{smallmatrix}\left(\left(\frac{\lambda T}{T_c}\right)^{2-m}\middle|\begin{smallmatrix}1,&&1,&1\\0,&&0,&0,&\frac{m}{2-m}\end{smallmatrix}\right)\right\}$$

and

$$\left(\frac{\partial Z_{\text{Sreg}}}{\partial \rho}\right)_T = \frac{3}{2}\left(n'_{\text{reg}} + \rho\, n''_{\text{reg}}\right)\left[\frac{\lambda^{1-m}}{2-m}\Gamma\left(\frac{1-m}{m-2}, \left(\frac{\lambda T}{T_c}\right)^{2-m}\right) - (m-1)^{-1}\left(1 - \left(\frac{T}{T_c}\right)^{m-1}\right)\right]$$

$$+ \frac{3}{2}\left(n_{\text{reg}}\, m' + \rho\, n_{\text{reg}}\, m'' + 2\rho\, n'_{\text{reg}}\, m'\right)\frac{1}{(m-1)^2}\left[\left(1 - \left(\frac{T}{T_c}\right)^{m-1}\right) + (m-1)\left(\frac{T}{T_c}\right)^{m-1}\ln\left(\frac{T}{T_c}\right)\right]$$

$$+ \frac{3}{2}\left(n_{\text{reg}}\, m' + \rho\, n_{\text{reg}}\, m'' + \rho\, n'_{\text{reg}}\, m'\right)\frac{\lambda^{1-m}}{(2-m)^2}\left\{\Gamma\left(\frac{1-m}{m-2}, \left(\frac{\lambda T}{T_c}\right)^{2-m}\right)\left[1 + \ln\left(\frac{\lambda T}{T_c}\right) - \ln(\lambda^{2-m})\right]\right.$$

$$\left. + \left(\frac{\lambda T}{T_c}\right)^{m-1}\ln\left(\left(\frac{\lambda T}{T_c}\right)^{2-m}\right)\exp\left(-\left(\frac{\lambda T}{T_c}\right)^{2-m}\right)\right\}$$

$$+ \frac{3}{2}\rho\, n'_{\text{reg}}\, m'\frac{\lambda^{1-m}}{(m-2)^2}\Gamma\left(\frac{1-m}{m-2}, \left(\frac{\lambda T}{T_c}\right)^{2-m}\right)\left(1 + \ln(\lambda^{m-2})\right)$$

$$- \frac{3}{2}\rho\, n_{\text{reg}}\, m'^2\frac{1}{(m-1)^3}\left[2\left(1 - \left(\frac{T}{T_c}\right)^{m-1}\right) + 2\left(\frac{T}{T_c}\right)^{m-1}\ln\left(\left(\frac{T}{T_c}\right)^{m-1}\right) - \left(\frac{T}{T_c}\right)^{m-1}\ln\left(\left(\frac{T}{T_c}\right)^{m-1}\right)^2\right]$$

$$- \frac{3}{2}\rho\, n_{\text{reg}}\, m'^2\frac{\lambda^{1-m}}{(m-2)^3}\left(3 + \ln(\lambda^{m-2})\right)\left\{\Gamma\left(\frac{1-m}{m-2}, \left(\frac{\lambda T}{T_c}\right)^{2-m}\right)\left[1 + \ln\left(\frac{\lambda T}{T_c}\right) - \ln(\lambda^{2-m})\right]\right.$$

$$\left. + \left(\frac{\lambda T}{T_c}\right)^{m-1}\ln\left(\left(\frac{\lambda T}{T_c}\right)^{2-m}\right)\exp\left(-\left(\frac{\lambda T}{T_c}\right)^{2-m}\right)\right\} \tag{A8}$$

$$- \frac{3}{2}\rho\, n_{\text{reg}}\, m'^2\frac{\lambda^{1-m}}{(2-m)^3}\left\{\Gamma\left(\frac{1-m}{m-2}, \left(\frac{\lambda T}{T_c}\right)^{2-m}\right)\left[1 + \ln\left(\frac{\lambda T}{T_c}\right) - 2\ln(\lambda^{2-m})\right] - \left(\frac{\lambda T}{T_c}\right)\ln\left(\left(\frac{\lambda T}{T_c}\right)^{2-m}\right)^2\exp\left(-\left(\frac{\lambda T}{T_c}\right)^{2-m}\right)\right.$$

$$\left. + \left(\frac{\lambda T}{T_c}\right)^{m-1}\ln\left(\left(\frac{\lambda T}{T_c}\right)^{2-m}\right)\left(2 - \ln\left(\left(\frac{\lambda T}{T_c}\right)^{2-m}\right)\right)\exp\left(-\left(\frac{\lambda T}{T_c}\right)^{2-m}\right) - \left(\frac{\lambda T}{T_c}\right)^{m-2}\ln\left(\frac{\lambda T}{T_c}\right)G\begin{smallmatrix}2&&0\\1&&2\end{smallmatrix}\left(\left(\frac{\lambda T}{T_c}\right)^{2-m}\middle|\begin{smallmatrix}&2\\1,&\frac{1}{2-m}\end{smallmatrix}\right)\right\}$$

$$+ \frac{3}{2}\rho\, m'\left[(2-m)\, n'_{\text{reg}} + n_{\text{reg}}\, m'\left(1 + \ln\left(\frac{\lambda T}{T_c}\right) - \ln(\lambda^{2-m})\right)\right]\frac{\lambda^{1-m}}{(m-2)^4}\ln\left(\left(\frac{\lambda T}{T_c}\right)^{2-m}\right)\left\{(2-m)\left(\frac{\lambda T}{T_c}\right)^{m-1}\exp\left(-\left(\frac{\lambda T}{T_c}\right)^{2-m}\right)\right.$$

$$\left. + \Gamma\left(\frac{1-m}{m-2}, \left(\frac{\lambda T}{T_c}\right)^{2-m}\right)\right\}$$

$$- 3\rho\, n_{\text{reg}}\, m'^2\frac{\lambda^{1-m}}{(m-2)^5}\left\{\left(\ln\left(\left(\frac{\lambda T}{T_c}\right)^{2-m}\right) + (2-m)(2 - \ln(\lambda^{2-m}))\right)G\begin{smallmatrix}3&&0\\2&&3\end{smallmatrix}\left(\left(\frac{\lambda T}{T_c}\right)^{2-m}\middle|\begin{smallmatrix}1,&&1\\0,&&0,&\frac{1-m}{m-2}\end{smallmatrix}\right)\right.$$

$$\left. + G\begin{smallmatrix}4&&0\\3&&4\end{smallmatrix}\left(\left(\frac{\lambda T}{T_c}\right)^{2-m}\middle|\begin{smallmatrix}1,&&1,&1\\0,&&0,&0,&\frac{1-m}{m-2}\end{smallmatrix}\right)\right\}$$

$$+ \frac{3}{2}\frac{\lambda^{1-m}}{(2-m)^3}\left(n_{\text{reg}}\, m' + \rho\, n_{\text{reg}}\, m'' + 2\rho\, n'_{\text{reg}}\, m'\right)G\begin{smallmatrix}3&&0\\2&&3\end{smallmatrix}\left(\left(\frac{\lambda T}{T_c}\right)^{2-m}\middle|\begin{smallmatrix}1,&&1,\\0,&&0,&\frac{1-m}{m-2}\end{smallmatrix}\right)$$

## Appendix B. Expression of the First and Second Derivatives of the Coefficients in the $C_V$ Expression

In this appendix, the expressions of the first derivatives of coefficients that appear in the expression of $C_V$ (see Equation (2)) and those useful for calculating pressure are given below.

$$\rho\, n'_{\text{reg}}(\rho) = \alpha_{\text{reg},1}\left(\frac{\rho}{\rho + \rho_{\text{t,Liq}}}\right)^{\varepsilon_{\text{reg},1a}}\exp\left(-\left(\frac{\rho}{\rho_{\text{t,Gas}}}\right)^{\varepsilon_{\text{reg},1b}}\right)\left\{\varepsilon_{\text{reg},1a}\frac{\rho_{\text{t,Liq}}}{\rho + \rho_{\text{t,Liq}}} - \varepsilon_{\text{reg},1b}\left(\frac{\rho}{\rho_{\text{t,Gas}}}\right)^{\varepsilon_{\text{reg},1b}}\right\}$$

$$- \alpha_{\text{reg},2}\left(\frac{\rho}{\rho_{\text{t,Liq}}}\right)^{\varepsilon_{\text{reg},2a}}\left\{-\varepsilon_{\text{reg},2a}\left(1 - \exp\left(-\left(\frac{\rho}{\rho_{\text{reg,Ronc}}}\right)^{-\varepsilon_{\text{reg},2b}}\right)\right) + \varepsilon_{\text{reg},2b}\left(\frac{\rho}{\rho_{\text{reg,Ronc}}}\right)^{-\varepsilon_{\text{reg},2b}}\exp\left(-\left(\frac{\rho}{\rho_{\text{reg,Ronc}}}\right)^{-\varepsilon_{\text{reg},2b}}\right)\right\} \tag{A9}$$

$$\rho\, m'(\rho) = -\left\{ \alpha_{m,4} + \alpha_{m,3}\left(-\frac{3}{2} + \frac{\rho}{\rho_c}\right)\left(\frac{\rho}{\rho_c}\right)^{\frac{3}{2}} \exp\left(-\frac{\rho}{\rho_c}\right) + \frac{3}{2}\alpha_{m,2}\left(\frac{\rho}{\rho_{t,\mathrm{Liq}}}\right)^{\frac{3}{2}} \exp\left(-\left(\frac{\rho}{\rho_{t,\mathrm{Liq}}}\right)^{\frac{3}{2}}\right) \right\}$$
$$+ \rho\, m'_{\mathrm{Ronc}}(\rho) \tag{A10}$$

$$\rho\, m'_{\mathrm{Ronc}}\left(\rho \geq \frac{M}{12.9}\,\mathrm{g/cm^3}\right) = \frac{\rho}{\rho_{m,\mathrm{Ronc}}}\left(\frac{\rho + \rho_{m,\mathrm{Ronc}}}{\rho_{m,\mathrm{Ronc}}}\right)^{\varepsilon_{m,5a}-1} \exp\left(-\exp\left(\left(\frac{\rho_{m,\mathrm{Ronc}}}{\rho}\right)^{\varepsilon_{m,5b}}\right)\right)$$
$$\times \left\{ \alpha_{m,1}\,\varepsilon_{m,5a} + \alpha_{m,4}\left(1 + \frac{\rho_{m,\mathrm{Ronc}}}{\rho} + \varepsilon_{m,5a}\ln\left(\frac{\rho}{\rho_c}\right)\right) + \varepsilon_{m,5b}\left(\frac{\rho + \rho_{m,\mathrm{Ronc}}}{\rho_{m,\mathrm{Ronc}}}\right)\left(\frac{\rho_{m,\mathrm{Ronc}}}{\rho}\right)^{\varepsilon_{m,5b}} \right.$$
$$\left. \times \exp\left(\left(\frac{\rho_{m,\mathrm{Ronc}}}{\rho}\right)^{\varepsilon_{m,5b}}\right)\left(\alpha_{m,1} + \alpha_{m,4}\ln\left(\frac{\rho}{\rho_c}\right)\right) \right\} \tag{A11}$$

$$\rho\, n'_{\mathrm{nonreg}}\left(\rho \leq \rho_{t,\mathrm{Liq}}\right) = \alpha_{\mathrm{nonreg},1}\left(\frac{\rho}{\rho_{t,\mathrm{Gas}}}\right)^{\varepsilon_{\mathrm{nonreg},1a}}\left(\frac{\rho_{t,\mathrm{Liq}}}{\rho}\right)\exp\left(-\left(\frac{\rho_{t,\mathrm{Liq}}}{\rho_{t,\mathrm{Gas}}}\right)^{\varepsilon_{\mathrm{nonreg},1b}}\left(\frac{\rho}{\rho_{t,\mathrm{Liq}}-\rho}\right)^{\varepsilon_{\mathrm{nonreg},1b}}\right)$$
$$\times \left\{ \varepsilon_{\mathrm{nonreg},1a}\frac{\rho}{\rho_{t,\mathrm{Liq}}} - \varepsilon_{\mathrm{nonreg},1b}\left(\frac{\rho_{t,\mathrm{Liq}}}{\rho_{t,\mathrm{Gas}}}\right)^{\varepsilon_{\mathrm{nonreg},1b}}\left(\frac{\rho}{\rho_{t,\mathrm{Liq}}-\rho}\right)^{1+\varepsilon_{\mathrm{nonreg},1b}} \right\}$$
$$+ \alpha_{\mathrm{nonreg},2}\left(\frac{\rho}{\rho_{t,\mathrm{Gas}}}\right)^{\varepsilon_{\mathrm{nonreg},2a}}\left(\frac{\rho_{t,\mathrm{Liq}}}{\rho}\right)\exp\left(-\left(\frac{\rho_{t,\mathrm{Liq}}}{\rho_{t,\mathrm{Gas}}}\right)^{\varepsilon_{\mathrm{nonreg},2b}}\left(\frac{\rho}{\rho_{t,\mathrm{Liq}}-\rho}\right)^{\varepsilon_{\mathrm{nonreg},2b}}\right)$$
$$\times \left\{ \varepsilon_{\mathrm{nonreg},2a}\frac{\rho}{\rho_{t,\mathrm{Liq}}} - \varepsilon_{\mathrm{nonreg},2b}\left(\frac{\rho_{t,\mathrm{Liq}}}{\rho_{t,\mathrm{Gas}}}\right)^{\varepsilon_{\mathrm{nonreg},2b}}\left(\frac{\rho}{\rho_{t,\mathrm{Liq}}-\rho}\right)^{1+\varepsilon_{\mathrm{nonreg},2b}} \right\} \tag{A12}$$

$$\rho\, T'_{\mathrm{div}}(\rho) = \alpha_{\mathrm{div},1}\left(\frac{\rho}{\rho_c}\right)^{\varepsilon_{\mathrm{div},1a}}\exp\left(-\left(\frac{\rho}{\rho_c}\right)^{\varepsilon_{\mathrm{div},1b}}\right)\left\{ \varepsilon_{\mathrm{div},1a} - \varepsilon_{\mathrm{div},1b}\left(\frac{\rho}{\rho_c}\right)^{\varepsilon_{\mathrm{div},1b}} \right\}$$
$$+ \alpha_{\mathrm{div},2}\left(\frac{\rho}{\rho_{t,\mathrm{Liq}}}\right)^{\varepsilon_{\mathrm{div},2a}}\exp\left(-\left(\frac{\rho}{\rho_{t,\mathrm{Liq}}}\right)^{\varepsilon_{\mathrm{div},2b}}\right)\left\{ \varepsilon_{\mathrm{div},2a} - \varepsilon_{\mathrm{div},2b}\left(\frac{\rho}{\rho_{t,\mathrm{Liq}}}\right)^{\varepsilon_{\mathrm{div},2b}} \right\} \tag{A13}$$

$$\rho\, n'_{\mathrm{crit}}(\rho) = \alpha_{\mathrm{crit},a}\left(\frac{\rho}{\rho_c}\right)^{\varepsilon_{\mathrm{crit},a}} \times \exp\left[-\left(\left(\alpha_{\mathrm{crit},b}\frac{\rho-\rho_c}{\rho_c}\right)^2\right)^{\varepsilon_{\mathrm{crit},b}}\right]$$
$$\times \left\{ \varepsilon_{\mathrm{crit},a} - 2\,\varepsilon_{\mathrm{crit},b}\rho\left(\frac{\alpha_{\mathrm{crit},b}}{\rho_c}\right)^{2\,\varepsilon_{\mathrm{crit},b}}\left((\rho-\rho_c)^2\right)^{\varepsilon_{\mathrm{crit},b}-\frac{1}{2}}\mathrm{sign}(\rho-\rho_c) \right\} \tag{A14}$$

$$\rho_c\varepsilon'_{\mathrm{crit}}(\rho) = 2\varepsilon_{\mathrm{crit},d}\varepsilon_{\mathrm{crit},e}^2\frac{\rho_{\mathrm{crit},a}-\rho}{\rho_{\mathrm{crit},a}}\frac{\rho_c}{\rho_{\mathrm{crit},a}}\exp\left(-\left(\varepsilon_{\mathrm{crit},e}\frac{\rho-\rho_{\mathrm{crit},a}}{\rho_{\mathrm{crit},a}}\right)^2\right)$$
$$+ 2\varepsilon_{\mathrm{crit},f}\varepsilon_{\mathrm{crit},g}^2\frac{\rho_{\mathrm{crit},b}-\rho}{\rho_{\mathrm{crit},b}}\frac{\rho_c}{\rho_{\mathrm{crit},b}}\exp\left(-\left(\varepsilon_{\mathrm{crit},g}\frac{\rho-\rho_{\mathrm{crit},b}}{\rho_{\mathrm{crit},b}}\right)^2\right) \tag{A15}$$

It is interesting to note that the limits of $\rho\, n'_{\mathrm{crit}}(\rho)$, $\rho\, T'_{\mathrm{div}}(\rho)$, $\rho\, n'_{\mathrm{nonreg}}(\rho)$, and $\rho\, n'_{\mathrm{reg}}(\rho)$ are equal to zero when $\rho \to 0$. Moreover, the limit of $\rho\, m'(\rho)$ is equal to 0.240087 when $\rho \to 0$.

The expressions of the second derivatives of the coefficients, which appear in the expression of $C_V$ and are useful for calculating the compressibility factor, are expressed below.

$$\rho^2\, n''_{\mathrm{reg}}(\rho) = \alpha_{\mathrm{reg},1}\left(\frac{\rho}{\rho+\rho_{t,\mathrm{Liq}}}\right)^{\varepsilon_{\mathrm{reg},1a}}\exp\left(-\left(\frac{\rho}{\rho_{t,\mathrm{Gas}}}\right)^{\varepsilon_{\mathrm{reg},1b}}\right)\left\{ \varepsilon_{\mathrm{reg},1b}^2\left(\frac{\rho}{\rho_{t,\mathrm{Gas}}}\right)^{\varepsilon_{\mathrm{reg},1b}}\left(-1+\left(\frac{\rho}{\rho_{t,\mathrm{Gas}}}\right)^{\varepsilon_{\mathrm{reg},1b}}\right) \right.$$
$$\left. + \varepsilon_{\mathrm{reg},1a}\left(\frac{\rho_{t,\mathrm{Liq}}}{\rho+\rho_{t,\mathrm{Liq}}}\right)^2\left(-2\frac{\rho}{\rho_{t,\mathrm{Liq}}}-1+\varepsilon_{\mathrm{reg},1a}\right) + \varepsilon_{\mathrm{reg},1b}\left(\frac{\rho}{\rho_{t,\mathrm{Gas}}}\right)^{\varepsilon_{\mathrm{reg},1b}}\left(1-2\,\varepsilon_{\mathrm{reg},1a}\frac{\rho_{t,\mathrm{Liq}}}{\rho+\rho_{t,\mathrm{Liq}}}\right) \right\}$$
$$+ \alpha_{\mathrm{reg},2}\,\varepsilon_{\mathrm{reg},2b}\left(\frac{\rho}{\rho_{t,\mathrm{Liq}}}\right)^{\varepsilon_{\mathrm{reg},2a}}\left(\frac{\rho}{\rho_{\mathrm{reg},\mathrm{Ronc}}}\right)^{-2\,\varepsilon_{\mathrm{reg},2b}}\exp\left(-\left(\frac{\rho}{\rho_{\mathrm{reg},\mathrm{Ronc}}}\right)^{-\varepsilon_{\mathrm{reg},2b}}\right)$$
$$\times \left\{ -\varepsilon_{\mathrm{reg},2b} + \left(1 - 2\,\varepsilon_{\mathrm{reg},2a} + \varepsilon_{\mathrm{reg},2b}\right)\left(\frac{\rho}{\rho_{\mathrm{reg},\mathrm{Ronc}}}\right)^{\varepsilon_{\mathrm{reg},2b}} \right\}$$
$$+ \alpha_{\mathrm{reg},2}\,\varepsilon_{\mathrm{reg},2a}\left(\varepsilon_{\mathrm{reg},2a}-1\right)\left(\frac{\rho}{\rho_{t,\mathrm{Liq}}}\right)^{\varepsilon_{\mathrm{reg},2a}}\left(1-\exp\left(-\left(\frac{\rho}{\rho_{\mathrm{reg},\mathrm{Ronc}}}\right)^{\varepsilon_{\mathrm{reg},2b}}\right)\right) \tag{A16}$$

$$\rho^2\, m''(\rho) = \alpha_{m,4} + \alpha_{m,3}\left(\frac{\rho}{\rho_c}\right)^{\frac{3}{2}}\exp\left(-\frac{\rho}{\rho_c}\right)\left(\frac{3}{4} - 3\frac{\rho}{\rho_c} + \left(\frac{\rho}{\rho_c}\right)^2\right)$$
$$+ \frac{3}{2}\alpha_{m,2}\left(\frac{\rho}{\rho_{t,\mathrm{Liq}}}\right)^{\frac{3}{2}}\exp\left(-\left(\frac{\rho}{\rho_{t,\mathrm{Liq}}}\right)^{\frac{3}{2}}\right)\left(1 + \frac{3}{2}\left(-1+\left(\frac{\rho}{\rho_{t,\mathrm{Liq}}}\right)^{\frac{3}{2}}\right)\right) + \rho^2\, m''_{\mathrm{Ronc}}(\rho) \tag{A17}$$

$$\rho^2\, m''_{\text{Ronc}}\left(\rho \geq \tfrac{M}{12.9}\ \text{g/cm}^3\right) = \left(\frac{\rho + \rho_{\text{m,Ronc}}}{\rho_{\text{m,Ronc}}}\right)^{\varepsilon_{m,5a}} \exp\left(-\exp\left(\left(\frac{\rho_{\text{m,Ronc}}}{\rho}\right)^{\varepsilon_{m,5b}}\right)\right)$$
$$\times \left\{ \alpha_{m,4}\left[-1 + 2\,\varepsilon_{m,5a}\frac{\rho}{\rho + \rho_{\text{m,Ronc}}} + 2\,\varepsilon_{m,5b}\left(\frac{\rho_{\text{m,Ronc}}}{\rho}\right)^{\varepsilon_{m,5b}}\exp\left(\left(\frac{\rho_{\text{m,Ronc}}}{\rho}\right)^{\varepsilon_{m,5b}}\right)\right]\right.$$
$$+ \left(\alpha_{m,1} + \alpha_{m,4}\ln\left(\tfrac{\rho}{\rho_c}\right)\right)\left[\varepsilon_{m,5a}(\varepsilon_{m,5a} - 1)\left(\frac{\rho}{\rho + \rho_{\text{m,Ronc}}}\right)^2\right.$$
$$-\varepsilon_{m,5b}^2\left(\frac{\rho_{\text{m,Ronc}}}{\rho}\right)^{2\varepsilon_{m,5b}}\exp\left(\left(\frac{\rho_{\text{m,Ronc}}}{\rho}\right)^{\varepsilon_{m,5b}}\right)\left(1 - \exp\left(\left(\frac{\rho_{\text{m,Ronc}}}{\rho}\right)^{\varepsilon_{m,5b}}\right)\right)$$
$$\left.\left.-\varepsilon_{m,5b}\left(\frac{\rho_{\text{m,Ronc}}}{\rho}\right)^{\varepsilon_{m,5b}}\exp\left(\left(\frac{\rho_{\text{m,Ronc}}}{\rho}\right)^{\varepsilon_{m,5b}}\right)\left(1 + \varepsilon_{m,5b} - 2\,\varepsilon_{m,5a}\frac{\rho}{\rho + \rho_{\text{m,Ronc}}}\right)\right]\right\} \tag{A18}$$

$$\rho^2\, n''_{\text{nonreg}}(\rho \leq \rho_{\text{t,Liq}}) = \left(\frac{\rho_{\text{t,Liq}}}{\rho - \rho_{\text{t,Liq}}}\right)^2 \left\{\alpha_{\text{nonreg},1}\left(\frac{\rho}{\rho_{\text{t,Gas}}}\right)^{\varepsilon_{\text{nonreg},1a}}\exp\left(-\left(\frac{\rho_{\text{t,Liq}}}{\rho_{\text{t,Gas}}}\right)^{\varepsilon_{\text{nonreg},1b}}\left(\frac{\rho}{\rho_{\text{t,Liq}}-\rho}\right)^{\varepsilon_{\text{nonreg},1b}}\right)\right.$$
$$\times \left[\varepsilon_{\text{nonreg},1a}(\varepsilon_{\text{nonreg},1a} - 1)\left(\frac{\rho}{\rho_{\text{t,Liq}}} - 1\right)^2 + \varepsilon_{\text{nonreg},1b}^2\left(\frac{\rho_{\text{t,Liq}}}{\rho_{\text{t,Gas}}}\right)^{2\,\varepsilon_{\text{nonreg},1b}}\left(\frac{\rho}{\rho_{\text{t,Liq}}-\rho}\right)^{2\,\varepsilon_{\text{nonreg},1b}}\right.$$
$$\left.+\varepsilon_{\text{nonreg},1b}\left(\frac{\rho_{\text{t,Liq}}}{\rho_{\text{t,Gas}}}\right)^{\varepsilon_{\text{nonreg},1b}}\left(\frac{\rho}{\rho_{\text{t,Liq}}-\rho}\right)^{\varepsilon_{\text{nonreg},1b}}\left(2(\varepsilon_{\text{nonreg},1a} - 1)\frac{\rho}{\rho_{\text{t,Liq}}} + 1 - 2\,\varepsilon_{\text{nonreg},1a} - \varepsilon_{\text{nonreg},1b}\right)\right]$$
$$+\alpha_{\text{nonreg},2}\left(\frac{\rho}{\rho_{\text{t,Gas}}}\right)^{\varepsilon_{\text{nonreg},2a}}\exp\left(-\left(\frac{\rho_{\text{t,Liq}}}{\rho_{\text{t,Gas}}}\right)^{\varepsilon_{\text{nonreg},2b}}\left(\frac{\rho}{\rho_{\text{t,Liq}}-\rho}\right)^{\varepsilon_{\text{nonreg},2b}}\right)$$
$$\times \left[\varepsilon_{\text{nonreg},2a}(\varepsilon_{\text{nonreg},2a} - 1)\left(\frac{\rho}{\rho_{\text{t,Liq}}} - 1\right)^2 + \varepsilon_{\text{nonreg},2b}^2\left(\frac{\rho_{\text{t,Liq}}}{\rho_{\text{t,Gas}}}\right)^{2\,\varepsilon_{\text{nonreg},2b}}\left(\frac{\rho}{\rho_{\text{t,Liq}}-\rho}\right)^{2\,\varepsilon_{\text{nonreg},2b}}\right.$$
$$\left.\left.+\varepsilon_{\text{nonreg},2b}\left(\frac{\rho_{\text{t,Liq}}}{\rho_{\text{t,Gas}}}\right)^{\varepsilon_{\text{nonreg},2b}}\left(\frac{\rho}{\rho_{\text{t,Liq}}-\rho}\right)^{\varepsilon_{\text{nonreg},2b}}\left(2\,(\varepsilon_{\text{nonreg},2a} - 1)\frac{\rho}{\rho_{\text{t,Liq}}} + 1 - 2\,\varepsilon_{\text{nonreg},2a} - \varepsilon_{\text{nonreg},2b}\right)\right]\right\} \tag{A19}$$

$$\rho^2\, T''_{\text{div}}(\rho) = \alpha_{\text{div},1}\left(\frac{\rho}{\rho_c}\right)^{\varepsilon_{\text{div},1a}}\exp\left(-\left(\frac{\rho}{\rho_c}\right)^{\varepsilon_{\text{div},1b}}\right)$$
$$\times \left\{\varepsilon_{\text{div},1a}(\varepsilon_{\text{div},1a} - 1) - \varepsilon_{\text{div},1b}(\varepsilon_{\text{div},1b} + 2\,\varepsilon_{\text{div},1a} - 1)\left(\frac{\rho}{\rho_c}\right)^{\varepsilon_{\text{div},1b}} + \varepsilon_{\text{div},1b}^2\left(\frac{\rho}{\rho_c}\right)^{2\,\varepsilon_{\text{div},1b}}\right\}$$
$$+\alpha_{\text{div},2}\left(\frac{\rho}{\rho_{\text{t,Liq}}}\right)^{\varepsilon_{\text{div},2a}}\exp\left(-\left(\frac{\rho}{\rho_{\text{t,Liq}}}\right)^{\varepsilon_{\text{div},2b}}\right)$$
$$\times \left\{\varepsilon_{\text{div},2a}(\varepsilon_{\text{div},2a} - 1) - \varepsilon_{\text{div},2b}(\varepsilon_{\text{div},2b} + 2\,\varepsilon_{\text{div},2a} - 1)\left(\frac{\rho}{\rho_{\text{t,Liq}}}\right)^{\varepsilon_{\text{div},2b}} + \varepsilon_{\text{div},2b}^2\left(\frac{\rho}{\rho_{\text{t,Liq}}}\right)^{2\,\varepsilon_{\text{div},2b}}\right\} \tag{A20}$$

$$\rho^2\, n''_{\text{crit}}(\rho) = \alpha_{\text{crit,a}}\left(\frac{\rho}{\rho - \rho_c}\right)^2\left(\frac{\rho}{\rho_c}\right)^{\varepsilon_{\text{crit,a}}}\exp\left[-\left(\left(\alpha_{\text{crit,b}}\frac{\rho - \rho_c}{\rho_c}\right)^2\right)^{\varepsilon_{\text{crit,b}}}\right]$$
$$\times \left\{\varepsilon_{\text{crit,a}}(\varepsilon_{\text{crit,a}} - 1)\left(\frac{\rho - \rho_c}{\rho}\right)^2 + 4\,\varepsilon_{\text{crit,b}}^2\left(\left(\alpha_{\text{crit,b}}\frac{\rho - \rho_c}{\rho_c}\right)^2\right)^{2\varepsilon_{\text{crit,b}}}\right.$$
$$\left.-2\varepsilon_{\text{crit,b}}\left(\left(\alpha_{\text{crit,b}}\frac{\rho - \rho_c}{\rho_c}\right)^2\right)^{\varepsilon_{\text{crit,b}}}\left(-1 + 2\,\varepsilon_{\text{crit,b}} + 2\,\varepsilon_{\text{crit,a}}\left(1 - \frac{\rho_c}{\rho}\right)\right)\right\} \tag{A21}$$

$$\rho_c^2\, \varepsilon''_{\text{crit}}(\rho) = 2\,\varepsilon_{\text{crit,d}}\,\varepsilon_{\text{crit,e}}^2\left(\frac{\rho_c}{\rho_{\text{crit,a}}}\right)^2\left(2\,\varepsilon_{\text{crit,e}}^2\left(\frac{\rho - \rho_{\text{crit,a}}}{\rho_{\text{crit,a}}}\right)^2 - 1\right)\exp\left(-\left(\varepsilon_{\text{crit,e}}\frac{\rho - \rho_{\text{crit,a}}}{\rho_{\text{crit,a}}}\right)^2\right)$$
$$+2\,\varepsilon_{\text{crit,f}}\,\varepsilon_{\text{crit,g}}^2\left(\frac{\rho_c}{\rho_{\text{crit,b}}}\right)^2\left(2\,\varepsilon_{\text{crit,g}}^2\left(\frac{\rho - \rho_{\text{crit,b}}}{\rho_{\text{crit,b}}}\right)^2 - 1\right)\exp\left(-\left(\varepsilon_{\text{crit,g}}\frac{\rho - \rho_{\text{crit,b}}}{\rho_{\text{crit,b}}}\right)^2\right) \tag{A22}$$

It is interesting to note that the limits of $\rho^2\, n''_{\text{crit}}(\rho)$, $\rho^2\, T''_{\text{div}}(\rho)$, $\rho^2\, n''_{\text{nonreg}}(\rho)$ and $\rho^2\, n''_{\text{reg}}(\rho)$ are also equal to zero when $\rho \to 0$, and the limit of $\rho^2\, m''(\rho) = -0.240087$ is same absolute value as that of the limit of $\rho\, m'(\rho)$ but with opposite sign.

## Appendix C. Approximate Formulas to Describe Some Properties along the Liquid–Vapor Coexistence Curve

In this appendix, we propose simple formulas, valid between $T_t$ and $T_c$, to approximate the pressure and densities of liquid and vapor deduced from Maxwell's relations. Thus, a simple formula to describe the variation of pressure with temperature along the saturation vapor pressure curve (SVP) can be written as follows:

$$P_{\text{sat}}(T) = P_c\,\exp\left(\frac{-5.9887\,\theta + 1.7151\,\theta^{3/2} + 3.344\,\theta^2}{T_r^{1.7791}}\right) \tag{A23}$$

where $T_r = T/T_c$ and $\theta = 1 - T_r$ with $T_c = T_{c,\text{non-ext formulation}} = 151.396$ K and $P_c = P_{c,\text{non-ext formulation}} = 49.9684$ bar (see Table 11).

The variation of the liquid density with temperature along the SVP can be described approximately by the following formula:

$$\rho_{\sigma l}(T) = \rho_c \frac{1 + 5.38842 \left(1 - T_r^{8.23084}\right)^{0.296407} - (5.38842 - 1.02985) \left(1 - T_r^{2.54783}\right)^{0.280344}}{T_r^{3.19766}} \tag{A24}$$

with $\rho_c = \rho_{c,\text{non-ext formulation}} = 0.543786$ g/cm$^3$ (see Table 11).

The variation of the vapor density with temperature along the SVP can be described approximately by the following formula:

$$\rho_{\sigma v}(T) = \rho_c \exp\left(-\left[5.5815\left(1 - T_r^{3.5104}\right)^{0.49565} - (5.5815 + 0.209)\,\theta^{0.44766}\right]\frac{\exp\left(1.1761\,\theta^2\right)}{T_r^2}\right) \\ \times \exp\left(-5.5815\,\theta^{1.1685}\exp\left(-\left|\frac{T_r - 0.99338}{0.019791}\right|\right)\right) \tag{A25}$$

with $\rho_c = \rho_{c,\text{non-ext formulation}} = 0.543786$ g/cm$^3$ (see Table 11). By construction, all formulas cross exactly the critical point.

Figure A1 shows the deviations obtained between the approximate formulas Equations (A23)–(A25) from the values calculated with Maxwell's equations for the non-extensive formulation of the present model. First of all, it can be observed that the maximum deviations occur systematically in a neighborhood very close to the critical point.

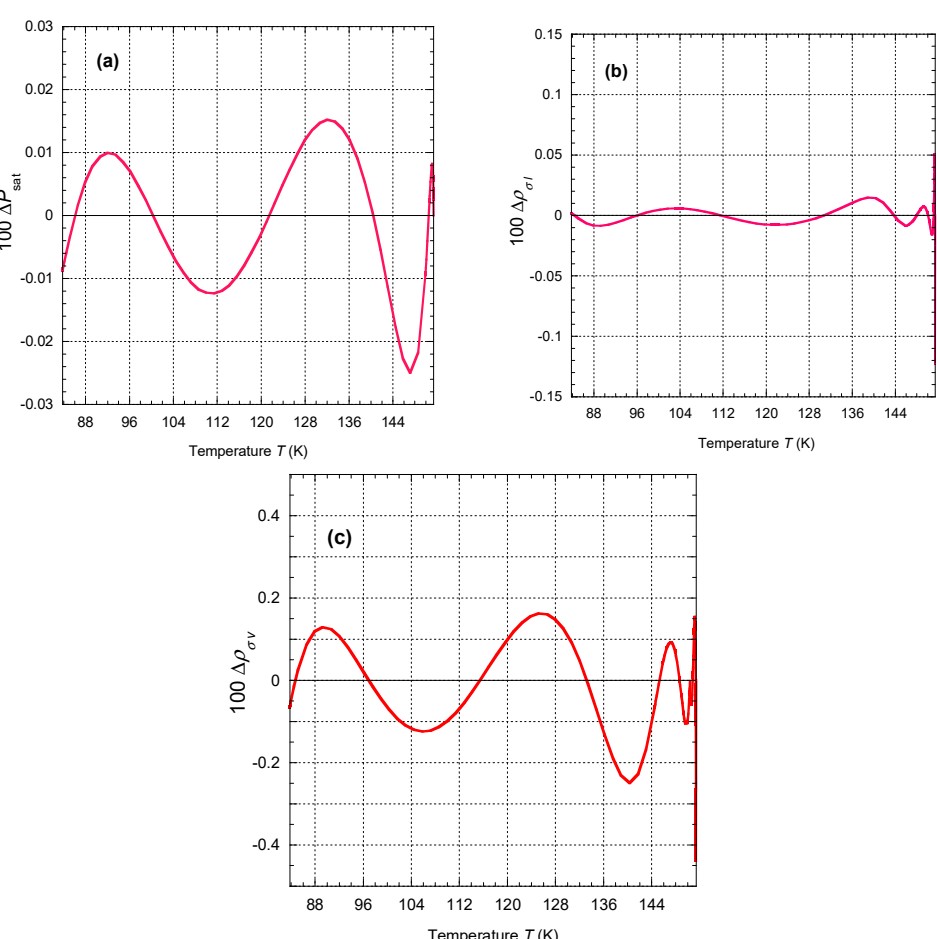

**Figure A1.** Percentage deviations $\Delta y = (y_{\text{Maxwell}} - y_{\text{calc}})/y_{\text{Maxwell}}$ of the approximate formulas Equations (A23)–(A25) from the values calculated from Maxwell's equations for the non-extensive formulation of the present model (i.e., Equations (50)–(52) with Equation (39)) in the temperature range of 83.8058 to 151.396 K.

Figure A1a shows that Equation (A23) reproduces the pressure very well because the deviation obtained is smaller than that of Figure 33, which compares the TSW model and the present model. Therefore, it can be said that Equation (A23) is both a good representation of the TSW model and the present model.

With respect to the deviation of the liquid density, Figure A1b appears to be consistent with the tolerance diagram of Figure 17 all along the coexistence curve up to the critical point. Again, the deviation is smaller than the corresponding one in Figure 33; therefore, Equation (A24) is a good representation of both the TSW model and the present model.

The greatest deviation is obtained for the vapor density, which does not conform to the tolerance diagram in Figure 17. Numerically, the deviation is comparable to the corresponding one in Figure 33. However, Figure A1c shows that the deviation is globally well centered on zero, indicating that the overall variation is correctly reproduced. This last remark is also valid for the deviations in Figure A1a,b.

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
