# Peer review of "A New Non-Extensive Equation of State for the Fluid Phases of Argon, Including the Metastable States, from the Melting Line to 2300 K and 50 GPa"

_fluids, doi:10.3390/fluids9050102_

Round 1
Reviewer 1 Report
Comments and Suggestions for Authors
The manuscript proposed a novel approach for developing an advanced equation of state (EoS) for pure argon. The method is far beyond the application to argon only. The advantage of the the approach is that it not only significantly reduces the number of the fitting coefficients of the EoS, but also takes much more physical consideration than other empirical EoSs that currently widely used. More importantly, the theoretical framework of the approach allows a reasonable extension of the thermodynamic properties prediction in the liquid-vapor coexistence region above the spinodal curve. By combining the data from Tegeler et al (NIST reference data) in low temperature and pressure data, and data from Ronchi et al at much higher temperature and pressure, the equation of state for argon proposed in this manuscript covers the most wide range of temperatures and pressures. The accuracy of the EoS is convincing. This 82-pages manuscript has original innovation and undoubtedly deserves a publication.
1. the authors mentioned both Tegeler et al.’s paper (Ref.4) (the primary one for comparison) and the NIST Refprop code (for generating some of the input data, such as fitting the data of Cv from NIST). As a fact, the Refprop code uses the EoS by Tegeler et al. Therefore, it is suggested to clarify that.
2. The authors claim that the TSW model gives negative values of Cv on some isotherms in the liquid-vapor coexistence. According to this reviewer’s knowledge, the Tegeler et al. never claimed that the TSW model can accurately predict Cv there. Instead, Refprop calculates the Cv value in the two-phase region by using Cv_x = x*Cv_sat_Vapor + (1-x)*Cv_sat_liq.
3. To fix the arbitrary functions produced in integration of Cv to get the internal energy, a comparison between calculated data and experimental data does solve this problem. However, what kind of experimental data should be chosen? Is it possible that choosing different experimental data leads to different values of the integration constants.
4. The authors claimed that the accuracy of the TSW model is very sensitive to the digits of the coefficients. Then how about the present coefficients listed in Table 1 if they are truncated?
5. Although at the very end of the document (in the appendix) the key properties like the triple point, critical point of argon are provided, it is suggested to mention or provided at the begging of the modeling, for better readability.
6. the lambda takes a value 6.8494 in eq.2. Why this value?
7. At the very beginning (page 6-7), table 2 is presented. The readers may have no idea about these values and this complete table, because by far most of the symbols listed in the table are not used/introduced yet.
8. lines 356 - 362, how about putting the coefficients into a table, like those for the other equations?
9. For better readability, it seems more appropriate to move the lines from 251 to 293 to somewhere right after eq.2.
10. in eq. 3 and 4 etc., does the lambda takes the same value as before, namely 6.8484?
11. Why choosing rho >= M/12.9 in eq.8? What about the consistency above and bellow this threshold value between the piece-wise function in eq.8, and eq.9 as well?
Some explanation regarding the fitting techniques are expected somewhere in the manuscript.
12. The authors provided another set of equation (7bis) other than equation (7) for the purpose of showing the flexibility of their method and checking the applicability of representing NIST data. A graph is actually expected to show how well the eq.7bis works for the NIST data (lower temperature and pressure); what is the consistency between the one with eq.7 and the one with eq.7bis, in particular, in the joint/overlapping region by the NIST data and the Ronchi data.
13. the last symbol in eq.18 is not correctly displayed on my computer. So does the one in the line after eq. 19.
14. The present model does predict the peak of Cv as shown in fig.8, which is a great improvement to the others. However, why two peaks on the two sides of the critical density line?
15. What does it mean that the present model (with those sub-equations and coefficients) well represents most of the Cv and P-rho-T data in most of the phase regions, however, it fails to produce the sharp peak of Cv in the vicinity of critical temperature as shown in fig. 32, although the authors emphasize that their model is capable of doing that by modifying the equation and doing a re-fitting?
Comments on the Quality of English Language1. The title is too long and contains some unnecessary words. e.g. “Based on an Original Approach” could be deleted. Reference citation is not suggested in the abstract.
2. Throughout the text, most of the citation of the figures and tables are not correctly displayed. Error message like “Error!bookmark not defined”, and “Error! Reference source not found” are everywhere.
3. Improve the numbering of the equations. The equation in line 207 is better to be numbered instead of as an inline equation.
4. The authors mentioned “section 0” a few times, but this section 0 does not exist.
Author Response
Our answers are written in red in the file.

Reviewer 2 Report
Comments and Suggestions for Authors
In this manuscript, the authors have developed a model base on a new equation of state for the fluid phase of argon from the melting to high temperatures and pressures. The model is based on the Helmholtz free energy equation which is derived from the measured quantities of Cv(rho,T) and P(rho,T) data which can be found in literature. The new model is able to fit some parts of the diagram phase of argon with high precision, sometimes better than other models, but also to fit other parts, as for example, when argon is in high-density metastable states, that other models cannot fit. Since the model is not based on polynomial expressions, oscillations in the fitting of data are not present which is another important point. This manuscript shows a very huge and impressive work. The theoretical approach is compared, at the same time, with other works in this field, and approved by confrontation to experimental data issued from different sources. All the thermodynamic calculations are given with details explanations on each part. This model is universal and can apply to other fluids, like water for example. This is why this work is of high scope and can be published in its current state in Fluids journal.
Some minor remarks can be made:
-There are numerous typographic errors in the manuscript, and reference to tables or figures are marked (see Error! Reference source not found.)
-Since Ref. 4 of Tegeler at al. is of importance, and already cited in the summary, it could have the number 1 and not 4.
-Since the model is universal, it could be more enlightened all along the paper that it can apply to other fluids.
Author Response

(The authors gave the same response as above.)
